# SGDA WITH SHUFFLING: FASTER CONVERGENCE FOR NONCONVEX-PŁ MINIMAX OPTIMIZATION

**Hanseul Cho, Chulhee Yun**
Kim Jaechul Graduate School of AI, KAIST
{jhs4015, chulhee.yun}@kaist.ac.kr

## ABSTRACT

Stochastic gradient descent-ascent (SGDA) is one of the main workhorses for solving finite-sum minimax optimization problems. Most practical implementations of SGDA randomly reshuffle components and sequentially use them (*i.e.*, without-replacement sampling); however, there are few theoretical results on this approach for minimax algorithms, especially outside the easier-to-analyze (strongly-)monotone setups. To narrow this gap, we study the convergence bounds of SGDA with random reshuffling (**SGDA-RR**) for smooth nonconvex-nonconcave objectives with Polyak-Łojasiewicz (PŁ) geometry. We analyze both simultaneous and alternating SGDA-RR for *nonconvex-PŁ* and *primal-PŁ-PŁ* objectives, and obtain convergence rates faster than with-replacement SGDA. Our rates extend to mini-batch SGDA-RR, recovering known rates for full-batch gradient descent-ascent (GDA). Lastly, we present a comprehensive lower bound for GDA with an arbitrary step-size ratio, which matches the full-batch upper bound for the primal-PŁ-PŁ case.

## 1 INTRODUCTION

A finite-sum minimax optimization problem aims to solve the following:

$$\min_{\boldsymbol{x}\in\mathcal{X}} \max_{\boldsymbol{y}\in\mathcal{Y}} f(\boldsymbol{x};\boldsymbol{y}) := \frac{1}{n}\sum_{i=1}^{n} f_i(\boldsymbol{x};\boldsymbol{y}), \tag{1}$$

where $f_i$ denotes the $i$-th component function. In plain language, we want to minimize the average of $n$ component functions for $\boldsymbol{x}$, while maximizing it for $\boldsymbol{y}$ given $\boldsymbol{x}$. There are many important areas in modern machine learning that fall within the minimax problem, including generative adversarial networks (GANs) (Goodfellow et al., 2020), adversarial attack and robust optimization (Madry et al., 2018; Sinha et al., 2018), multi-agent reinforcement learning (MARL) (Li et al., 2019), AUC maximization (Ying et al., 2016; Liu et al., 2020; Yuan et al., 2021), and many more. In most cases, the objective $f$ is usually nonconvex-nonconcave, *i.e.*, neither convex in $\boldsymbol{x}$ nor concave in $\boldsymbol{y}$. Since general nonconvex-nonconcave problems are known to be intractable, we would like to tackle the problems with some additional structures, such as smoothness and Polyak-Łojasiewicz (PŁ) condition(s). We elaborate the detailed settings for our analysis, *nonconvex-PŁ* and *primal-PŁ-PŁ* (or, *PŁ(Φ)-PŁ*), in Section 2.

One of the simplest and most popular algorithms to solve the problem (1) would be *stochastic gradient descent-ascent* (**SGDA**). This naturally extends the idea of stochastic gradient descent (SGD) used for minimization problems. Given an initial iterate $(\boldsymbol{x}_0; \boldsymbol{y}_0)$, at time $t \in \mathbb{N}$, SGDA (randomly) chooses an index $i(t) \in \{1, \ldots, n\}$ and accesses the $i(t)$-th component to perform a pair of updates

$$\begin{bmatrix} \boldsymbol{x}_t = \boldsymbol{x}_{t-1} - \alpha\nabla_1 f_{i(t)}(\boldsymbol{x}_{t-1};\boldsymbol{y}_{t-1}), \\ \boldsymbol{y}_t = \boldsymbol{y}_{t-1} + \beta\nabla_2 f_{i(t)}(\boldsymbol{x}';\boldsymbol{y}_{t-1}), \end{bmatrix} \quad \text{where } \boldsymbol{x}' = \begin{cases} \boldsymbol{x}_{t-1}, & (\text{simSGDA}), \text{ or} \\ \boldsymbol{x}_t, & (\text{altSGDA}). \end{cases}$$

Here, $\alpha > 0$ and $\beta > 0$ are the step sizes and $\nabla_j$ denotes the gradient with respect to $j$-th argument for $f_{i(t)}$ ($j = 1, 2$). As shown in the update equations above, there are two widely used versions of SGDA: *simultaneous SGDA* (**simSGDA**), and *alternating SGDA* (**altSGDA**).

In such stochastic gradient methods, there are two main categories of sampling schemes for the component indices $i(t)$. One way is to sample $i(t)$ independently (in time) and uniformly at random

from $\{1, \ldots, n\}$, which is called *with-replacement sampling*. This scheme is widely adopted in theory papers because it makes analysis of stochastic methods amenable: the noisy gradients $\nabla f_{i(t)}$ are independent over time $t$ and are unbiased estimators of the full-batch gradient $\nabla f$. In contrast, the vast majority of practical implementations employ *without-replacement sampling*, indicating a huge theory-practice gap. In without-replacement sampling, we sample each index precisely once at each epoch. Perhaps the most popular of such schemes is *random reshuffling* (**RR**), which uniformly randomly shuffles the order of indices at the beginning of every epoch. Unfortunately, it is well-known that without-replacement methods are much more difficult to analyze theoretically, largely because the sampled indices in each epoch are no longer independent of each other.

Interestingly, for minimization problems, several recent works overcome this obstacle and show that SGD using without-replacement sampling leads to faster convergence, given that the number of epochs is large enough (Nagaraj et al., 2019; Ahn et al., 2020; Mishchenko et al., 2020; Rajput et al., 2020; Nguyen et al., 2021; Yun et al., 2021; 2022). On the other hand, for minimax problems like (1), the majority of the studies still assume with-replacement sampling and/or rely on independent unbiased gradient oracles (Nouiehed et al., 2019; Guo et al., 2020; Lin et al., 2020; Yan et al., 2020; Yang et al., 2020; Loizou et al., 2021; Beznosikov et al., 2022). There are very few results on minimax algorithms using without-replacement sampling; even most of the existing ones take advantage of (strong-)convexity (in $\boldsymbol{x}$) and/or (strong-)concavity (in $\boldsymbol{y}$) (Das et al., 2022; Maheshwari et al., 2022; Yu et al., 2022). Detailed comparative analysis of these works is conducted in Section 4.

Putting all these issues into consideration, our main question is the following.

> *Does SGDA using without-replacement component sampling provably converge fast,*
> *even on smooth nonconvex-nonconcave objective $f$ with PŁ structures?*

## 1.1 SUMMARY OF OUR CONTRIBUTIONS

To answer the question, we analyze the convergence of SGDA with random reshuffling (**SGDA-RR**, Algorithm 1). We analyze both the simultaneous and alternating versions of SGDA-RR and prove convergence theorems for the following two regimes. Here we denote the step size ratio as $r = \beta/\alpha$.

- When $-f(\boldsymbol{x}; \boldsymbol{y})$ satisfies $\mu_2$-PŁ condition in $\boldsymbol{y}$ (**nonconvex-PŁ**) and component function $f_i$'s are $L$-smooth, we prove that SGDA-RR with $r \gtrsim (L/\mu_2)^2$ converges to $\varepsilon$-stationarity in expectation after $\mathcal{O}\left(nrL\varepsilon^{-2} + \sqrt{n}r^{1.5}L\varepsilon^{-3}\right)$ gradient evaluations (Theorem 1).

- Further assuming $\mu_1$-PŁ condition on $\Phi(\cdot) := \max_{\boldsymbol{y}} f(\cdot; \boldsymbol{y})$ (**primal-PŁ-PŁ**, or **PŁ($\Phi$)-PŁ**), we prove that SGDA-RR with $r \gtrsim (L/\mu_2)^2$ converges within $\varepsilon$-accuracy in expectation after $\tilde{\mathcal{O}}\left(\frac{nLr}{\mu_1}\log(\varepsilon^{-1}) + \sqrt{n}L(\frac{r}{\mu_1})^{1.5}\varepsilon^{-1}\right)$ gradient evaluations (Theorem 2).

As will be discussed in Section 4, the rates shown above are *faster* than existing results on with-replacement SGDA. In fact, Theorems 1 & 2 are special cases ($b = 1$) of our extended theorems (Theorems 4 & 5 in Appendix A) that analyze *mini-batch* SGDA-RR of batch size $b$; by setting $b = n$, we also recover known convergence rates for full-batch gradient descent ascent (GDA). Hence, our analysis covers the entire spectrum between vanilla SGDA-RR ($b = 1$) and GDA ($b = n$).

- Additionally, we provide complexity lower bounds for solving strongly-convex-strongly-concave (SC-SC) minimax problems using full-batch simultaneous GDA with an arbitrarily fixed step size ratio $r = \beta/\alpha$. Perhaps surprisingly, we find that the lower bound for SC-SC functions matches the convergence upper bound for a much larger class of primal-PŁ-PŁ functions when the step size ratio satisfies $r \gtrsim L^2/\mu_2^2$ (Theorem 3).

## 2 PROBLEM SETUP

### 2.1 NOTATION

In our problem (1), the domain of every $f_i$ is $\mathcal{Z} = \mathcal{X} \times \mathcal{Y}$, where $\mathcal{X} = \mathbb{R}^{d_x}$, $\mathcal{Y} = \mathbb{R}^{d_y}$, and $\mathcal{Z} = \mathbb{R}^d$: we concern unconstrained problems for simplicity. We denote the Euclidean norm and the standard inner product by $\|\cdot\|$ and $\langle \cdot, \cdot \rangle$, respectively. We often use an abbreviated notation $\boldsymbol{z} = (\boldsymbol{x}; \boldsymbol{y}) \in \mathcal{Z}$ for $\boldsymbol{x} \in \mathcal{X}$ and $\boldsymbol{y} \in \mathcal{Y}$. Even when $\boldsymbol{z}$ or $(\boldsymbol{x}; \boldsymbol{y})$ is followed by superscripts and/or subscripts, we use the symbols interchangeably; *e.g.*, $\boldsymbol{z}_i^k = (\boldsymbol{x}_i^k; \boldsymbol{y}_i^k)$. Note that we split the arguments $\boldsymbol{x}$ (for minimization) and $\boldsymbol{y}$ (for maximization) by a semicolon (';'). We use $\nabla_1$ and $\nabla_2$ to denote the

---

**Algorithm 1** simSGDA/altSGDA-RR

---

1: **Given:** The number of components $n$; the number of epochs $K$; step sizes $\alpha, \beta > 0$
2: **Initialize:** $(\boldsymbol{x}_0^1; \boldsymbol{y}_0^1) \in \mathbb{R}^{d_x} \times \mathbb{R}^{d_y}$
3: **for** $k \in [K]$ **do**
4:     Sample $\sigma_k \sim \text{Unif}(\mathbb{S}_n)$            ▷ RR: uniformly randomly shuffle the indices every epoch
5:     **for** $i \in [n]$ **do**
6:         $\boldsymbol{x}_i^k = \boldsymbol{x}_{i-1}^k - \alpha \nabla_1 f_{\sigma_k(i)}(\boldsymbol{x}_{i-1}^k; \boldsymbol{y}_{i-1}^k)$
7:         **if** simSGDA-RR **then**
8:             $\boldsymbol{y}_i^k = \boldsymbol{y}_{i-1}^k + \beta \nabla_2 f_{\sigma_k(i)}(\boldsymbol{x}_{i-1}^k; \boldsymbol{y}_{i-1}^k)$                ▷ simultaneous update: $\boldsymbol{x}$ & $\boldsymbol{y}$
9:         **else if** altSGDA-RR **then**
10:             $\boldsymbol{y}_i^k = \boldsymbol{y}_{i-1}^k + \beta \nabla_2 f_{\sigma_k(i)}(\boldsymbol{x}_i^k; \boldsymbol{y}_{i-1}^k)$                ▷ alternating update: $\boldsymbol{x} \to \boldsymbol{y}$
11:     $(\boldsymbol{x}_0^{k+1}; \boldsymbol{y}_0^{k+1}) = (\boldsymbol{x}_n^k; \boldsymbol{y}_n^k)$

---

gradients with respect to first and second arguments, respectively.Accordingly, we can write the full gradient as, e.g., $\nabla g = [\nabla_1 g^\top; \nabla_2 g^\top]^\top$. For a positive integer $N$, we denote $[N] := \{1, \dots, N\}$. Let the set $\mathbb{S}_N$ be a symmetric group of degree $N$. That is, each *permutation* $\sigma \in \mathbb{S}_N$ is a bijection from $[N]$ to itself, or equivalently, a re-arrangement of $[N]$. Lastly, we use the usual $\mathcal{O}/\Omega/\Theta$ notation for bounds, where $\tilde{\mathcal{O}}/\tilde{\Omega}/\tilde{\Theta}$ are used for hiding some logarithmic factors, respectively.

## 2.2   Algorithms: simSGDA-RR & altSGDA-RR

As we explained in Section 1, we consider simSGDA and altSGDA combined with RR, a without-replacement sampling scheme. We call them **simSGDA-RR** and **altSGDA-RR**, respectively. We present a detailed description of the methods in Algorithm 1. For completeness, we also provide an extended version that uses mini-batches of size $\geq 1$ (Algorithm 2) in Appendix A. For comparison, we call the SGDA algorithms using *with-replacement* sampling by just **simSGDA** and **altSGDA**.

The quantities $\alpha, \beta > 0$ are step sizes associated with $\boldsymbol{x}$ and $\boldsymbol{y}$, respectively. We use two separate symbols $\alpha$ and $\beta$ to allow the two step sizes to be different. Such algorithms are sometimes called *two-time-scale* algorithms, in a broader sense, and they are adopted in nonconvex minimax optimization problems (Heusel et al., 2017; Lin et al., 2020; Yang et al., 2020). In fact, a recent result (Li et al., 2022) shows that having $\alpha \neq \beta$ is sometimes *necessary* for convergence.

## 2.3   Assumptions and definitions

To define the function classes that we are interested in solving, we introduce a few assumptions.

**Assumption 1** (Component smoothness). *Every $i$-th component $f_i : \mathcal{Z} \to \mathbb{R}$ is $L$-**smooth**, i.e., $f_i$ is differentiable and $\nabla f_i$ is $L$-Lipschitz continuous:* $\|\nabla f_i(\boldsymbol{z}) - \nabla f_i(\bar{\boldsymbol{z}})\| \leq L \|\boldsymbol{z} - \bar{\boldsymbol{z}}\|$. *As a result,* $f_i(\bar{\boldsymbol{z}}) - f_i(\boldsymbol{z}) \leq \langle \nabla f_i(\boldsymbol{z}), \bar{\boldsymbol{z}} - \boldsymbol{z} \rangle + \frac{L}{2}\|\bar{\boldsymbol{z}} - \boldsymbol{z}\|^2$ $(\forall \boldsymbol{z}, \bar{\boldsymbol{z}})$ *and the average $f$ of $f_i$'s is also $L$-smooth.*[1]

**Assumption 2** (Component gradient variance). *There exist constants $A, B \geq 0$ such that, for any $\boldsymbol{z} = (\boldsymbol{x}; \boldsymbol{y}) \in \mathcal{Z}$ and $j \in \{1, 2\}$, we have $\frac{1}{n}\sum_{i=1}^n \|\nabla_j f_i(\boldsymbol{z}) - \nabla_j f(\boldsymbol{z})\|^2 \leq A \|\nabla_j f(\boldsymbol{z})\|^2 + B$.*

**Assumption 3.** *For a function $f : \mathcal{X} \times \mathcal{Y} \to \mathbb{R}$, the **primal function** $\Phi : \mathcal{X} \to \mathbb{R}$ is well-defined as $\Phi(\boldsymbol{x}) := \max_{\boldsymbol{y}' \in \mathcal{Y}} f(\boldsymbol{x}; \boldsymbol{y}')$. For each $\boldsymbol{x} \in \mathcal{X}$, the set $\mathcal{Y}_{\boldsymbol{x}}^* := \arg\max_{\boldsymbol{y}' \in \mathcal{Y}} f(\boldsymbol{x}; \boldsymbol{y}')$ is non-empty and closed. Moreover, we assume $\Phi(\boldsymbol{x})$ is bounded below by $\Phi^* = \inf_{\boldsymbol{x}' \in \mathcal{X}} \Phi(\boldsymbol{x}') > -\infty$.*

Note that Assumption 2 controls the discrepancy between the objective function $f$ and its components $f_i$'s; it is similar to Assumption 2 of Nguyen et al. (2021), adapted to minimax problems. Letting $A = 0$ recovers a common assumption of the uniformly bounded variance of component gradients; thus, our assumption is a relaxation. Also, note that $A = B = 0$ when $n = 1$.

We now add an additional structure to our objective function, which is called Polyak-Łojasiewicz (PŁ) condition. A function $g : \mathbb{R}^d \to \mathbb{R}$ is said to be $\mu$-**PŁ** if it has a minimum value $g^*$ and satisfies

$$\|\nabla g(\boldsymbol{t})\|^2 \geq 2\mu(g(\boldsymbol{t}) - g^*). \quad (\forall \, \boldsymbol{t} \in \mathbb{R}^d) \tag{$\mu$-PŁ}$$

---

[1] As we noted, Assumption 1 directly implies the *average smoothness* which is a common requirement in the analysis with unbiased gradient oracles. Nevertheless, we claim that Assumption 1 is not more crucial than without-replacement sampling to obtain faster convergence rates: see Appendix F for details and proofs.

Readers could find several studies and applications that the condition involves, in the papers by Karimi et al. (2016); Nouiehed et al. (2019); Yang et al. (2020); Liu et al. (2020), and more. Note that every $\mu$-strongly convex[2] function satisfies $\mu$-PŁ condition, whereas a PŁ function does not need to be convex. Hence, $\mu$-PŁ is a strict generalization of $\mu$-strong convexity. In addition, every stationary point of a PŁ function is a global optimum, which is a benign property for optimization.

We are interested in the case where our objective function $f(\boldsymbol{x}; \boldsymbol{y})$ has such a structure in terms of $\boldsymbol{y}$ (Assumption 4). Sometimes, we further assume the primal function $\Phi$ is also PŁ (Assumption 5). We emphasize that we do not necessarily assume the PŁ conditions for the individual $f_i$'s.

**Assumption 4** ($\boldsymbol{y}$-side PŁ). *For each (fixed) $\boldsymbol{x} \in \mathcal{X}$, $-f(\boldsymbol{x}; \cdot)$ is $\mu_2$-PŁ, i.e., for every $(\boldsymbol{x}; \boldsymbol{y}) \in \mathcal{Z}$, $\|\nabla_2 f(\boldsymbol{x}; \boldsymbol{y})\|^2 \geq 2\mu_2(\Phi(\boldsymbol{x}) - f(\boldsymbol{x}; \boldsymbol{y}))$, where $\Phi$ is the primal function associated with $f$.*

**Assumption 5** (Primal PŁ, or PŁ($\Phi$)). *The primal function $\Phi(\cdot) = \max_{\boldsymbol{y}'} f(\boldsymbol{x}; \boldsymbol{y}')$ of $f$ is $\mu_1$-PŁ, i.e., for every $\boldsymbol{x} \in \mathcal{X}$, $\|\nabla \Phi(\boldsymbol{x})\|^2 \geq 2\mu_1(\Phi(\boldsymbol{x}) - \Phi^*)$, where $\Phi^* = \min_{\boldsymbol{x}} \Phi(\boldsymbol{x})$ is well-defined.*

We say the function $f$ is **nonconvex-PŁ** when it satisfies Assumption 4. Since we do not assume any convexity/concavity, it is generally hard to reach global optima. Due to the $\boldsymbol{y}$-side PŁ condition, we can guarantee that the primal function $\Phi$ is differentiable and even $L_\Phi$-smooth with $L_\Phi \leq L + L^2/\mu_2$ (Proposition 9 in Appendix B). Since the problem (1) can be reformulated as the minimization problem of $\Phi$ (when we can always find $\boldsymbol{y}$ well that maximizes $f(\boldsymbol{x}; \boldsymbol{y})$ given $\boldsymbol{x}$), we could aim to find an approximate first-order stationary point of $\Phi$, by making the norm of the gradient of $\Phi$ small.

On top of that, if $f$ satisfies both Assumptions 4 and 5, the function is said to be **primal-PŁ-PŁ**, or **PŁ($\Phi$)-PŁ** for short.[3] In this case, we directly aim not only to decrease the primal function $\Phi$ associated with the objective function $f$ but also to increase the function value $f(\boldsymbol{x}; \boldsymbol{y})$ in terms of $\boldsymbol{y}$. To evaluate how close we are to our goal, we define a *potential function* $V_\lambda$ later in Section 3. When we attain $V_\lambda(\boldsymbol{x}^*, \boldsymbol{y}^*) = 0$, it implies that we arrive at a global minimax point: $f(\boldsymbol{x}^*, \boldsymbol{y}^*) = \Phi(\boldsymbol{x}^*) = \Phi^*$. The function $V_\lambda$ enables us to develop a unified analysis for nonconvex-PŁ and PŁ($\Phi$)-PŁ objective functions; we discuss this in greater detail in Section 3.

## 3  MAIN RESULTS

Based on the assumptions stated in the previous section, we present the convergence results for both smooth nonconvex-PŁ objectives and smooth PŁ($\Phi$)-PŁ objectives. Before stating the main theorems, we first introduce the most important tool for our analyses: the *potential function*.

### 3.1  POTENTIAL FUNCTION $V_\lambda$

For our convergence analyses, we utilize a function $V_\lambda : \mathcal{X} \times \mathcal{Y} \to \mathbb{R}$ defined as

$$V_\lambda(\boldsymbol{x}; \boldsymbol{y}) := \lambda(\Phi(\boldsymbol{x}) - \Phi^*) + (\Phi(\boldsymbol{x}) - f(\boldsymbol{x}; \boldsymbol{y})), \tag{2}$$

where $\lambda > 0$ is a constant. We borrow inspiration from Yang et al. (2020) and Das et al. (2022) to come up with this function, although the placement of $\lambda$ of ours is different. In fact, the convergence to a neighborhood of a global minimax point (if it exists) implies the reduction of this function. For each $\boldsymbol{x}$, a non-negative term $\Phi(\boldsymbol{x}) - f(\boldsymbol{x}; \boldsymbol{y})$ gets smaller as $\boldsymbol{y}$ makes $f(\boldsymbol{x}; \boldsymbol{y})$ larger. The term becomes zero when $\boldsymbol{y} = \boldsymbol{y}^*(\boldsymbol{x})$ for some $\boldsymbol{y}^*(\boldsymbol{x}) \in \mathcal{Y}_{\boldsymbol{x}}^*$, since $\Phi(\boldsymbol{x}) = f(\boldsymbol{x}; \boldsymbol{y}^*(\boldsymbol{x}))$. Also, another non-negative term $\Phi(\boldsymbol{x}) - \Phi^*$ gets smaller as $\boldsymbol{x}$ makes $\Phi(\boldsymbol{x})$ smaller. Thus, as $(\boldsymbol{x}; \boldsymbol{y})$ approaches to a minimax optimal point, $V_\lambda(\boldsymbol{x}; \boldsymbol{y})$ decreases to near zero. In general, $V_\lambda$ is not guaranteed to attain exact zero, especially when the objective function $f(\boldsymbol{x}; \boldsymbol{y})$ is nonconvex in $\boldsymbol{x}$ (*e.g.*, $f$ is nonconvex-PŁ). Nevertheless, the potential function is still useful for deriving our convergence results.

### 3.2  MAIN THEOREMS: UPPER BOUNDS OF CONVERGENCE RATES

Now, we present our main results. We provide a detailed comparison of our theorems against existing results in Section 4. We present the full proof in Appendices C and D. We remark that both Theorems 1 and 2 are special cases (for mini-batch size $b = 1$) of their *mini-batch* extensions: Theorems 4 and 5 in Appendix A.

---

[2]We say a function $g : \mathbb{R}^d \to \mathbb{R}$ is $\mu$-strongly convex for some $\mu > 0$ if it holds $g(\boldsymbol{x}') \geq g(\boldsymbol{x}) + \langle \nabla g(\boldsymbol{x}), \boldsymbol{x}' - \boldsymbol{x} \rangle + (\mu/2) \|\boldsymbol{x}' - \boldsymbol{x}\|^2$ ($\forall \boldsymbol{x}, \boldsymbol{x}'$); we say $g$ is $\mu$-strongly concave if $-g$ is $\mu$-strongly convex.

[3]The PŁ($\Phi$)-PŁ condition is much weaker than **two-sided PŁ** condition assuming "$\boldsymbol{x}$-side" PŁ condition: see Proposition 10. As pointed out by Guo et al. (2020), there exist a PŁ($\Phi$)-PŁ function $g(\boldsymbol{x}; \boldsymbol{y})$ that is not $\boldsymbol{x}$-side $\mu$-PŁ for any $\mu > 0$ but even strongly *concave* in $\boldsymbol{x}$.

**Theorem 1** (Nonconvex-PŁ). *Suppose that $f$ satisfies Assumptions 1, 2, 3, and 4. Let $\kappa_2 = L/\mu_2$, where $\mu_2$ is PŁ constant of $-f(\boldsymbol{x}; \cdot)$ at all $\boldsymbol{x}$. Let $\lambda = 4$. Choose the step sizes $\alpha$ and $\beta$ such that*

$$\beta = \min\left\{ \frac{1}{6L\sqrt{n(n+A)}},\ \mathcal{O}\left(\left(\frac{V_\lambda(\boldsymbol{z}_0^1)}{Bn^2K}\right)^{\frac{1}{3}}\right)\right\} \quad and \quad \alpha = \frac{\beta}{r},$$

*for some $r \geq 14\kappa_2^2$. Then, both simSGDA-RR and altSGDA-RR (Algorithm 1) satisfy*

$$\frac{1}{K}\sum_{k=1}^{K} \mathbb{E}\left\|\nabla\Phi(\boldsymbol{x}_0^k)\right\|^2 \leq \mathcal{O}\left(\frac{rLV_\lambda(\boldsymbol{z}_0^1)}{K}\sqrt{1+\frac{A}{n}} + r\left(\frac{L^2BV_\lambda(\boldsymbol{z}_0^1)^2}{nK^2}\right)^{1/3}\right).$$

**Upper bound on gradient complexity.** To achieve $\varepsilon$-stationarity of the primal function, *i.e.*, $\frac{1}{K}\sum_{k=1}^{K} \mathbb{E}\left\|\nabla\Phi(\boldsymbol{x}_0^k)\right\|^2 \leq \varepsilon^2$, a sufficient number of gradient evaluations (denoted by $T_\varepsilon = nK$) is

$$T_\varepsilon = \mathcal{O}\left(\frac{rLV_\lambda(\boldsymbol{z}_0^1)}{\varepsilon^2}\max\left\{\sqrt{n^2+nA}, \frac{\sqrt{rnB}}{\varepsilon}\right\}\right).$$

**Theorem 2** (PŁ($\Phi$)-PŁ). *Suppose that $f$ satisfies Assumptions 1, 2, 3, 4, and 5. Let $\kappa_1 = L/\mu_1$ and $\kappa_2 = L/\mu_2$, where $\mu_1$ and $\mu_2$ are PŁ constants of $\Phi(\cdot)$ and $-f(\boldsymbol{x}; \cdot)$ (at all $\boldsymbol{x}$), respectively. Let $\lambda = 4$. Choose appropriate step sizes $\alpha$ and $\beta$ such that*

$$\beta = \min\left\{\frac{1}{6L\sqrt{n(n+A)}},\ \tilde{\mathcal{O}}\left(\frac{\kappa_2^2}{\mu_1 nK}\right)\right\} \quad and \quad \alpha = \frac{\beta}{r},$$

*for some $r \geq 14\kappa_2^2$. Then, both simSGDA-RR and altSGDA-RR (Algorithm 1) satisfy*

$$\mathbb{E}[V_\lambda(\boldsymbol{z}_0^{K+1})] \leq \mathcal{O}\left(V_\lambda(\boldsymbol{z}_0^1)\cdot\exp\left(-\frac{K}{12\kappa_1 r\sqrt{1+\frac{A}{n}}}\right)\right) + \tilde{\mathcal{O}}\left(\frac{\kappa_1^2 r^3 B}{\mu_1 nK^2}\right).$$

**Upper bound on gradient complexity.** To achieve $\varepsilon^2$-accuracy on expectation of $V_\lambda(\boldsymbol{z}_n^K)$, *i.e.*, $\mathbb{E}[V_\lambda(\boldsymbol{z}_n^K)] \leq \varepsilon^2$, a sufficient number of gradient evaluations (denoted by $T_\varepsilon' = nK$) is

$$T_\varepsilon' = \max\left\{\mathcal{O}\left(\kappa_1 r\sqrt{n^2+nA}\cdot\log\left(\frac{V_\lambda(\boldsymbol{z}_0^1)}{\varepsilon}\right)\right),\ \tilde{\mathcal{O}}\left(\frac{\kappa_1 r^{3/2}}{\varepsilon}\sqrt{\frac{nB}{\mu_1}}\right)\right\}.$$

**Remark on step size ratio.** In both theorems, we use the step sizes of ratio $r = \beta/\alpha \gtrsim \kappa_2^2$. It is common to use such a step size scheme $r = \Theta(\kappa_2^2)$ to analyze two-time-scale (S)GDA for nonconvex minimax problems (Jin et al., 2020; Lin et al., 2020; Yang et al., 2020).

**Remark on the parameter $\lambda$.** In our convergence analyses, we arbitrarily choose $\lambda = 4$ which makes the numerical calculations easier. The value of $\lambda > 0$ does not matter for the equivalence between the equation $V_\lambda(\boldsymbol{x}^*; \boldsymbol{y}^*) = 0$ and global minimax condition (Proposition 11 in Appendix B). Also, the choice of $\lambda$ in both theorems can be arbitrary as long as $\lambda > 1$; our logic does not fall apart if other appropriate step sizes for that $\lambda$ are chosen. That is to say, we can show that the sequence $V_\lambda(\boldsymbol{z}_0^k)$ almost monotonically decreases, ignoring some small variance terms.

## 4 COMPARISON WITH RELATED WORKS

### 4.1 COMPARISON WITH STOCHASTIC WITH-REPLACEMENT SETTING

First of all, we confirm that SGDA with random reshuffling (RR) has *faster* convergence rates (*i.e.*, fewer gradient computations) than SGDA based on with-replacement sampling. In particular, we compare our results with the analyses on the purely stochastic minimax settings which assume that every stochastic gradient oracle is *independently sampled and unbiased*: this assumption is naturally satisfied by with-replacement sampling for the finite-sum settings we consider. To make the comparisons fair and easy, we simply let $r = \beta/\alpha = \Theta(\kappa_2^2)$, $A = 0$, and $B = \tau^2$.

Lin et al. (2020, Theorem 4.5) present a convergence rate for with-replacement simSGDA with $r = \Theta(\kappa_2^2)$ run on nonconvex $\mu_2$-strongly-concave problems with a convex *bounded* constraint set $\mathcal{Y}$ for dual variable $\boldsymbol{y}$. Their gradient complexity to achieve $\frac{1}{T}\sum_{t=1}^{T}\mathbb{E}\left\|\nabla\Phi(\boldsymbol{x}_t)\right\|^2 \leq \varepsilon^2$ (where $T$ is the

number of iterations) is written as $T_\varepsilon = \mathcal{O}\left(\frac{\kappa_2^2 L \Delta_\Phi + \kappa_2 L^2 D^2}{\varepsilon^2} \max\left\{1, \frac{\kappa_2 \tau^2}{\varepsilon^2}\right\}\right)$, where $\kappa_2 = L/\mu_2$, $\Delta_\Phi = \Phi(\boldsymbol{x}_0) - \Phi^*$, $D = \operatorname{diam} \mathcal{Y}$, and $\tau^2$ is the variance of the (unbiased) stochastic gradient oracles. Their complexity can be simplified as $\mathcal{O}(\kappa_2^3 \tau^2 \varepsilon^{-4})$, treating other factors as constants. In contrast, our Theorem 1 has a better gradient complexity in terms of $\varepsilon$ and $\tau$, thanks to shuffling:

$$\mathcal{O}\left(\frac{\kappa_2^2 L V_\lambda(\boldsymbol{z}_0^1)}{\varepsilon^2} \max\left\{n, \frac{\kappa_2 \tau \sqrt{n}}{\varepsilon}\right\}\right), \qquad \text{(Ours, from Theorem 1)}$$

or simply $\mathcal{O}(\kappa_2^3 \tau \sqrt{n} \varepsilon^{-3})$. Thus, our gradient complexity for both simSGDA-RR *and* altSGDA-RR is better than that of with-replacement simSGDA when $\varepsilon$ is small as $\varepsilon \leq \mathcal{O}(\tau/\sqrt{n})$. Our rate has three more strengths: (i) we do not require strong concavity in $\boldsymbol{y}$, which is a strictly stronger assumption than requiring $\boldsymbol{y}$-side PL condition; (ii) we do not require the constraint set $\mathcal{Y}$ to be bounded; (iii) our result can easily extend to the case of *any* mini-batch sizes, whereas Lin et al. (2020) need a particular choice of mini-batch size $M = \mathcal{O}(\kappa_2 \tau^2/\varepsilon)$ to ensure convergence.

For nonconvex-PŁ objectives, Yang et al. (2022, Theorem 3.1) provide a convergence rate for with-replacement altSGDA with $r = \Theta(\kappa_2^2)$. Their rate can be translated to a gradient complexity for achieving $\frac{1}{T}\sum_{t=1}^T \mathbb{E}\|\nabla\Phi(\boldsymbol{x}_t)\|^2 \leq \varepsilon^2$, written as $\mathcal{O}\left(\frac{\kappa_2^2 L V_\lambda(\boldsymbol{z}_0)}{\varepsilon^2}\left(1 + \frac{\kappa_2^2 V_\lambda(\boldsymbol{z}_0)^2 \tau^2}{\Delta_\Phi \varepsilon^2}\right)\right)$ or simply $\mathcal{O}(\kappa_2^4 \tau^2 \varepsilon^{-4})$. Therefore, our gradient complexity for both altSGDA-RR *and* simSGDA-RR is better when $\varepsilon$ is small as $\varepsilon \leq \mathcal{O}(\kappa_2 \tau/\sqrt{n})$.

For PŁ($\Phi$)-PŁ objectives, Yang et al. (2020, Theorem 3.3) obtain a convergence rate for with-replacement altSGDA with $r = \Theta(\kappa_2^2)$.[4] They apply diminishing step sizes ($\mathcal{O}(1/t)$, $t \in \mathbb{N}$) to derive a gradient complexity bound $\mathcal{O}\left(\frac{\kappa_1 \kappa_2^4 \tau^2}{\mu_1 \varepsilon^2}\right)$ to achieve $\mathbb{E}[V_\lambda(\boldsymbol{z}_T)] \leq \varepsilon^2$. One can apply the constant step sizes depending on the total number $T$ of iterations to their analysis and derive a similar complexity with only deterioration in a logarithmic factor. In contrast, our gradient complexity for both sim/altSGDA-RR using constant step sizes can be written as, for small enough $\varepsilon$,

$$\tilde{\mathcal{O}}\left(\frac{\kappa_1 \kappa_2^3 \tau \sqrt{n}}{\varepsilon \sqrt{\mu_1}}\right). \qquad \text{(Ours, from Theorem 2)}$$

This is a better complexity in $\varepsilon$ and $\kappa_2$, especially when $\varepsilon \leq \tilde{\mathcal{O}}\left(\kappa_2 \tau/\sqrt{n\mu_1}\right)$, even without the requirement of diminishing step size.

### 4.2 COMPARISON WITH OTHER WORKS ON STOCHASTIC WITHOUT-REPLACEMENT SETTING

One of the most relevant works to this paper is Das et al. (2022, Theorem 3). The authors obtain a similar convergence rate to us for the two-sided PŁ objective, based on linearization of gradients, but for a dissimilar algorithm which they refer to as *AGDA-RR*. The algorithm can be also thought of as *epoch-wise*-alternating SGDA-RR, whereas our algorithm (altSGDA-RR) can be called as *step-wise*-alternating SGDA-RR. In epoch $k$, their algorithm (*i*) performs updates only on $\boldsymbol{x}$ ($\boldsymbol{x}_0^k, \ldots, \boldsymbol{x}_n^k$) while fixing $\boldsymbol{y}$ to $\boldsymbol{y}_0^k$, and then (*ii*) performs updates only on $\boldsymbol{y}$ ($\boldsymbol{y}_0^k, \ldots, \boldsymbol{y}_n^k$) while fixing $\boldsymbol{x}$ to $\boldsymbol{x}_0^{k+1} = \boldsymbol{x}_n^k$. We believe that our step-wise algorithm is closer to practice, especially when $n$ is large. Because of the distinction between algorithms, the proof techniques are also different.

Xie et al. (2021, Theorem 3) present a convergence rate of *CD-MA*, an extension of simSGDA to the cross-device federated learning setup, on nonconvex-PŁ setting. Their convergence result for *CD-MA* also assumes mini-batch sampling by random reshuffling. As a consequence, they yield a rate analogous to our Theorem 1 if we reduce their result to the single-machine setup. Nevertheless, our convergence bound contains a term that shrinks with the number of components or mini-batches, whereas theirs does not. For a more detailed comparison, please refer to Appendix H.

There are also some works on RR-based (constrained) minimax optimization algorithms other than SGDA, but for convex-concave problems. Maheshwari et al. (2022) present *OGDA-RR*, a gradient-free RR-based optimistic GDA algorithm. Yu et al. (2022) study *stochastic proximal point with RR*, consisting of double-loop epochs. Their analyses exploit convex-concavity and Lipschitz continuity of their objective, based on the arguments by Nagaraj et al. (2019). This enables a direct usage of the duality gap, the difference between primal function $\Phi(\cdot)$ and dual function $\Psi(\cdot) = \min_{\boldsymbol{x}} f(\boldsymbol{x}; \cdot)$, as a criterion for optimality. On the contrary, our work relies on a different structure of the functions, which in turn differentiates the constructions of convergence rates.

---

[4]Although they consider two-sided PŁ problems, their analysis applies to PŁ($\Phi$)-PŁ problems as well.

### 4.3 COMPARISON WITH DETERMINISTIC SETTING

Here, we compare our rates with (full-batch) *gradient descent-ascent* (GDA):

$$
\begin{bmatrix} \boldsymbol{x}_k = \boldsymbol{x}_{k-1} - \alpha \nabla_1 f(\boldsymbol{x}_{k-1}; \boldsymbol{y}_{k-1}), \\ \boldsymbol{y}_k = \boldsymbol{y}_{k-1} + \beta \nabla_2 f(\boldsymbol{x}'; \boldsymbol{y}_{k-1}), \end{bmatrix} \quad \text{where } \boldsymbol{x}' = \begin{cases} \boldsymbol{x}_{k-1}, & (\textit{simGDA}), \text{ or} \\ \boldsymbol{x}_k, & (\textit{altGDA}). \end{cases}
$$

It uses the whole information of the objective $f$ at every iteration without any noise. For comparison with GDA, we utilize our extended theorems for arbitrary mini-batch size $b$ (Theorems 4 and 5 in Appendix A). By letting $b = n$ and matching our iterate $\boldsymbol{z}_0^k = (\boldsymbol{x}_0^k; \boldsymbol{y}_0^k)$ to a GDA iterate $\boldsymbol{z}_k = (\boldsymbol{x}_k; \boldsymbol{y}_k)$, our results reduce to upper convergence bounds for simGDA and altGDA.

For nonconvex-PŁ problems (Theorems 1 & 4), the convergence rate and *iteration* complexity (*i.e.*, sufficient number of iterations $K_\varepsilon$) become

$$
\min_{k \in [K]} \|\nabla \Phi(\boldsymbol{x}_k)\|^2 \leq \mathcal{O}\left(\frac{\kappa_2^2 L V_\lambda(\boldsymbol{z}_1)}{K}\right); \quad \textit{i.e.,} \;\; K_\varepsilon = \mathcal{O}\left(\frac{\kappa_2^2 L V_\lambda(\boldsymbol{z}_1)}{\varepsilon^2}\right), \tag{3}
$$

when $r = \Theta(\kappa_2^2)$. This is similar to a known rate of simGDA with $r = \Theta(\kappa_2^2)$ for nonconvex-strongly-concave problems by Lin et al. (2020, Theorem 4.4) as a special case. Their iteration complexity is written as $\mathcal{O}((\kappa_2^2 L \Delta_\Phi + \kappa_2 L^2 D^2)/\varepsilon^2)$, where the symbols are already defined in Section 4.1. To see how the two bounds compare in terms of the factors other than $\varepsilon$, notice that we have $\Phi(\boldsymbol{x}) - f(\boldsymbol{x}; \boldsymbol{y}) \leq \frac{L}{2} \|\boldsymbol{y} - \boldsymbol{y}^*(\boldsymbol{x})\|^2$ for any $(\boldsymbol{x}; \boldsymbol{y})$, due to the $L$-smoothness of $-f$. Here, $\boldsymbol{y}^*(\boldsymbol{x})$ is an element of $\mathcal{Y}_{\boldsymbol{x}}^* = \arg\max_{\boldsymbol{y}} f(\boldsymbol{x}; \boldsymbol{y})$. Thus, we have $V_\lambda(\boldsymbol{z}_1) = \lambda[\Phi(\boldsymbol{x}_1) - \Phi^*] + [\Phi(\boldsymbol{x}_1) - f(\boldsymbol{z}_1)] \leq \lambda \Delta_\Phi + L D^2/2$. As a result, we could *loosely* translate our iteration complexity (3) to $\mathcal{O}((\kappa_2^2 L \Delta_\Phi + \kappa_2^2 L^2 D^2)/\varepsilon^2)$. We suspect that the discrepancy in terms of $\kappa_2$ comes from the fact that our analysis does not require the (strong) concavity in terms of $\boldsymbol{y}$ or a bounded constraint $\mathcal{Y}$: these requirements made a considerable difference in proofs.

For PŁ($\Phi$)-PŁ problems (Theorems 2 & 5), the rate and iteration complexity ($K_\varepsilon'$) become

$$
V_\lambda(\boldsymbol{z}_{K+1}) \leq V_\lambda(\boldsymbol{z}_1) \cdot \exp\left(-\frac{K}{C \kappa_1 \kappa_2^2}\right); \quad \textit{i.e.,} \;\; K_\varepsilon' = \mathcal{O}\left(\kappa_1 \kappa_2^2 \log(1/\varepsilon)\right) \tag{4}
$$

where $r = \Theta(\kappa_2^2)$ and $C$ is a numerical constant. This recovers the linear convergence by Yang et al. (2020, Theorem 3.2) as a special case, where they prove convergence of altGDA with step size ratio $r = \Theta(\kappa_2^2)$ for two-sided PŁ problem. Following the proof of (Yang et al., 2020, Theorem 3.2), one can show that the bound (4) indeed implies the actual convergence to a global minimax point $\boldsymbol{z}^*$, in the sense that we can achieve $\|\boldsymbol{z}_k - \boldsymbol{z}^*\| \leq \varepsilon$ in $\mathcal{O}\left(\kappa_1 \kappa_2^2 \log(1/\varepsilon)\right)$ iterations.

## 5 LOWER BOUND FOR (FULL-BATCH) SIMGDA USING SEPARATE STEP SIZES

As an extension of the discussion from Section 4.3, we characterize a lower complexity bound of deterministic simGDA with separate step sizes $(\alpha, \beta)$ of arbitrary ratio $r = \beta/\alpha$, for smooth strongly-convex-strongly-concave (SC-SC) cases. Surprisingly, at least for $r \gtrsim \kappa_2^2$, our lower bound matches the upper complexity bound of GDA for a much wider class of smooth PŁ($\Phi$)-PŁ problems,[5] which is quite surprising.

For a smooth PŁ($\Phi$)-PŁ problems, simGDA with at least $r = \Omega(\kappa_2^2)$ has an upper complexity bound $K = \mathcal{O}(\kappa_1 r \log(1/\varepsilon))$ for a *global* $\varepsilon$-convergence $V_\lambda(\boldsymbol{z}_K) \leq \varepsilon^2$ in terms of potential function. This means that the lowest complexity is $\mathcal{O}(\kappa_1 \kappa_2^2 \log(1/\epsilon))$ achieved when $r = \Theta(\kappa_2^2)$. On the other hand, for a $L$-smooth $\mu$-SC-SC problem with saddle point $\boldsymbol{z}^*$, it is well-known that the simGDA with a single step-size ($\alpha = \beta$) has a tight upper/lower complexity $K = \Theta(\kappa^2 \log(1/\varepsilon))$ to achieve $\|\boldsymbol{z}_K - \boldsymbol{z}^*\|^2 \leq \varepsilon^2$, where $\kappa = L/\mu$ (*e.g.*, Das et al. (2022, Theorem C.1)). The difference of complexity bounds in condition number ($\kappa_1 \kappa_2^2$ v.s. $\kappa^2$) is somewhat questionable because, at least in smooth minimization problems, strongly convex problems and PŁ problems have identical gradient descent (GD) iteration complexity $\mathcal{O}(\kappa \log(1/\varepsilon))$ (Karimi et al., 2016, Theorem 1).

One could ask where the discrepancy in terms of $\kappa$ comes from: is it due to (*i*) the criteria ($V_\lambda(\boldsymbol{z}_K)$ v.s. $\|\boldsymbol{z}_K - \boldsymbol{z}^*\|^2$) for $\varepsilon$-accuracy, (*ii*) the function classes (PŁ($\Phi$)-PŁ v.s. SC-SC), or (*iii*) the step size

---

[5]strongly-convex-strongly-concave (SC-SC) $\subset$ two-sided PŁ $\subset$ PŁ($\Phi$)-PŁ $\subset$ nonconvex-PŁ.

ratios ($\Omega(\kappa_2^2)$ v.s. 1)? We answer the question by showing the following theorem: the discrepancy in $\kappa$ comes from the step size ratio difference. We defer the proof to Appendix E.

**Theorem 3** (Lower bound, ratio-specific). *Consider a class $\mathcal{F}(L, \mu_1, \mu_2)$ of functions $f(\boldsymbol{x}; \boldsymbol{y})$ with two arguments $\boldsymbol{x}$ and $\boldsymbol{y}$, which is $L$-smooth, $\mu_1$-strongly-convex in $\boldsymbol{x}$, and $\mu_2$-strongly-concave in $\boldsymbol{y}$. Suppose $\kappa_1 = L/\mu_1 \geq c$ and $\kappa_2 = L/\mu_2 \geq c$ for some constant $c > 1$. Then, for any step size ratio $r = \beta/\alpha > 0$, there exists a function $f \in \mathcal{F}(L, \mu_1, \mu_2)$ with a unique saddle point $\boldsymbol{z}^*$, for which simGDA with any step sizes $(\alpha, \beta) = (\beta/r, \beta)$ requires at least*

$$K = \begin{cases} \Omega\left(\kappa_1 r \log(1/\varepsilon)\right), & \text{if } r \geq \kappa_2/c, \\ \Omega\left(\kappa_1 \kappa_2 \log(1/\varepsilon)\right), & \text{if } c/\kappa_1 \leq r \leq \kappa_2/c, \\ \Omega((\kappa_2/r) \log(1/\varepsilon)), & \text{if } 0 < r \leq c/\kappa_1 \end{cases}$$

*iterations to achieve either $\|\boldsymbol{z}_K - \boldsymbol{z}^*\|^2 \leq \varepsilon^2$ or $V_\lambda(\boldsymbol{z}_K) \leq \varepsilon^2$.*

Thanks to Theorem 3, we can say from Theorem 5 that for any step size ratio $r \gtrsim \kappa_2^2$, we have a *tight* upper bound on the iteration complexity $K = \mathcal{O}(\kappa_1 r \log(1/\varepsilon))$ of simGDA for general PŁ($\Phi$)-PŁ problems. Note that Theorem 3 also subsumes the existing lower bound of the equal-step-size ($r = 1$) simGDA for $\mu$-SC-SC problems.

Given the tightness of bounds for $r \gtrsim \kappa_2^2$, a natural next step is to discuss $1 \lesssim r \lesssim \kappa_2^2$. Recent work by Li et al. (2022) also discusses the step size ratio of simGDA. In Li et al. (2022, Theorem 4.1), the authors construct a $y$-side strongly-concave function[6] and show that simGDA with a step size ratio $r \leq \kappa_2$ is *impossible* to converge. The *necessity* of $r \gtrsim \kappa_2$ implied by this theorem also applies to the PŁ($\Phi$)-PŁ case. Thus, there is no hope for showing an upper convergence bound of simGDA with $1 \lesssim r \lesssim \kappa_2$ for general nonconvex-PŁ problems. We remark that their theorem does not contradict nor subsume Theorem 3 because we consider a much smaller function class (SC-SC) to construct the lower bounds.

On the *sufficiency* of $r \gtrsim \kappa_2$ for convergence, Li et al. (2022, Theorem 4.2) show that simGDA with $r \geq c\kappa$ (for some $c > 1$) can *locally* converge at the iteration complexity $\mathcal{O}(\kappa_1 r \log(1/\varepsilon))$ for some nonconvex-strongly-concave problems, which matches the bound in Theorem 3. Our upper bounds (Theorems 4 and 5) do require $r \gtrsim \kappa_2^2$, which may look suboptimal, but we claim that our results are not necessarily weaker. One reason is that our convergence guarantee is *global*, *i.e.*, independent of the initialization. Another reason is that their analysis is only valid when a *differential Stackelberg equilibrium*[7] exists, whereas a general PŁ($\Phi$)-PŁ function may not have such an equilibrium (for an example, see Proposition 13 in Appendix B).

As far as we know, it is still an open problem whether a *global* convergence bound for simGDA on nonconvex-PŁ problems can be shown when the step size ratio $r$ is between $\Omega(\kappa_2)$ and $\mathcal{O}(\kappa_2^2)$.

## 6 EXPERIMENTS

To validate our main theoretical findings, here we present some numerical results. We focus on the primal-PŁ-strongly-concave (or PŁ($\Phi$)-SC, which is PŁ($\Phi$)-PŁ as well) quadratic games of the form

$$\min_{\boldsymbol{x} \in \mathbb{R}^d} \max_{\boldsymbol{y} \in \mathbb{R}^d} f(\boldsymbol{x}; \boldsymbol{y}) = \tfrac{1}{2}\boldsymbol{x}^\top \boldsymbol{A}\boldsymbol{x} + \boldsymbol{x}^\top \boldsymbol{B}\boldsymbol{y} - \tfrac{1}{2}\boldsymbol{y}^\top \boldsymbol{C}\boldsymbol{y} = \tfrac{1}{n}\sum_{i=1}^n f_i(\boldsymbol{x}; \boldsymbol{y}),$$

$$\text{where} \quad f_i(\boldsymbol{x}; \boldsymbol{y}) = \tfrac{1}{2}\boldsymbol{x}^\top \boldsymbol{A}_i\boldsymbol{x} + \boldsymbol{x}^\top \boldsymbol{B}_i\boldsymbol{y} - \tfrac{1}{2}\boldsymbol{y}^\top \boldsymbol{C}_i\boldsymbol{y} + \boldsymbol{u}_i^\top \boldsymbol{x} - \boldsymbol{v}_i^\top \boldsymbol{y}. \tag{5}$$

This toy example is often used to numerically evaluate the minimax algorithms (Yang et al., 2020; Loizou et al., 2021; Das et al., 2022) and appears in various domains such as AUC maximization (Ying et al., 2016), policy evaluation (Du et al., 2017), and imitation learning (Cai et al., 2019)

To make the game in Equation (5) satisfy PŁ($\Phi$)-SC and component $L$-smoothness, we should sample the coefficient matrices and vectors carefully. First, they need to be $\|\boldsymbol{A}_i\|_2, \|\boldsymbol{B}_i\|_2, \|\boldsymbol{C}_i\|_2 \leq L$ and $\sum_{i=1}^n \boldsymbol{u}_i = \sum_{i=1}^n \boldsymbol{v}_i = \boldsymbol{0}$. To make the primal function $\Phi$ a well-defined real-valued function

---

[6] $g(x; y) = -\tfrac{L}{2}x^2 + Lxy - \tfrac{\mu}{2}y^2$, where $L/\mu > 1$: its primal function is strongly convex.

[7] Loosely speaking, a differential Stackelberg equilibrium is a stationary point $(\boldsymbol{x}^*; \boldsymbol{y}^*)$ where $f(\boldsymbol{x}^*; \cdot)$ is locally strongly concave near $\boldsymbol{y}^*$ and $\Phi(\cdot)$ is locally strongly convex near $\boldsymbol{x}^*$.

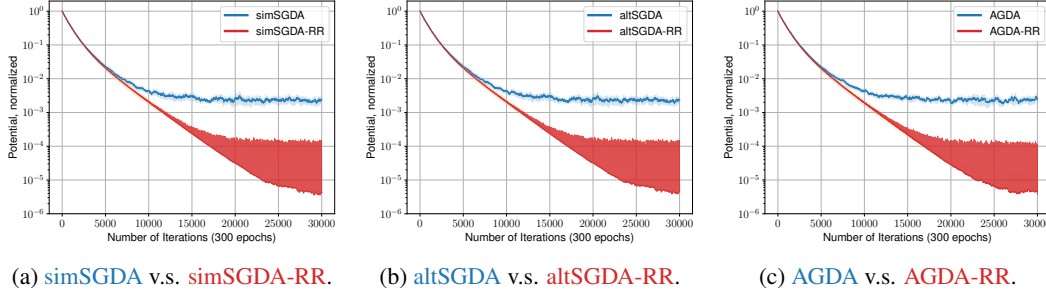

(a) simSGDA v.s. simSGDA-RR.  (b) altSGDA v.s. altSGDA-RR.  (c) AGDA v.s. AGDA-RR.

Figure 1: Experimental results on quadratic games (5). Solid lines: average across 10 different runs. Shaded regions: 95% confidence intervals ($\pm 1.96$ std). Dots: start/end of epochs. The vertical axes are on a *logarithmic scale*.

for any $\boldsymbol{x} \in \mathbb{R}^d$, we choose $\boldsymbol{C} = \frac{1}{n} \sum_{i=1}^{n} \boldsymbol{C}_i$ to be positive definite, *i.e.*, $\mu \boldsymbol{I} \preceq \boldsymbol{C}$ for an identity matrix $\boldsymbol{I}$ and $\mu > 0$. Then, the primal function can be explicitly written as

$$\Phi(\boldsymbol{x}) = \max_{\boldsymbol{y} \in \mathbb{R}^d} f(\boldsymbol{x}; \boldsymbol{y}) = \tfrac{1}{2} \boldsymbol{x}^\top \left( \boldsymbol{A} + \boldsymbol{B} \boldsymbol{C}^{-1} \boldsymbol{B}^\top \right) \boldsymbol{x} := \tfrac{1}{2} \boldsymbol{x}^\top \boldsymbol{M} \boldsymbol{x}.$$

We construct a matrix $\boldsymbol{M} := \boldsymbol{A} + \boldsymbol{B} \boldsymbol{C}^{-1} \boldsymbol{B}^\top$ to be rank-deficient positive semi-definite. Letting the smallest nonzero eigenvalue of $\boldsymbol{M}$ by $\mu$, we ensure that $\Phi$ is $\mu$-PŁ but not strongly convex. We emphasize that the objective function $f$ is not even (strongly-)convex in $\boldsymbol{x}$ in general.

We compare six algorithms in total: simSGDA-RR, altSGDA-RR, AGDA-RR (as defined in Das et al. (2022)), and the with-replacement counterparts of these three algorithms. To this end, on 5 different randomly-generated quadratic games and under 2 random seeds per game (*i.e.*, 10 runs per algorithm), we run each algorithm for the same number of epochs using constant step sizes of ratio $\beta/\alpha = c \kappa_2^2$ for some constant $c$ and $\kappa_2 = L/\mu$.

We report the potential function values ($V_\lambda$, defined in Equation (2)) at every iteration.[8] Results are presented in Figure 1: the values are normalized by dividing them by the initial value. As we discussed in Section 4.1, we observe that the random reshuffling considerably accelerates the convergence of the algorithms. Furthermore, all three algorithms with random reshuffling show more or less the same performance. Specifically, the plots for simSGDA (resp. simSGDA-RR) and altSGDA (resp. altSGDA-RR) are almost identical. We believe this is because we choose a random seed for each of the 10 different runs and share it across different algorithms.

Please refer to Appendix G for more detailed construction, discussion, and comparative study of the experimental results.

## 7 CONCLUSION

We investigated stochastic algorithms based on without-replacement component sampling, called simSGDA-RR and altSGDA-RR, for solving smooth nonconvex finite-sum minimax optimization problems. We established convergence rates under the $\boldsymbol{y}$-side PŁ condition (nonconvex-PŁ) and, additionally, the primal PŁ condition (PŁ($\Phi$)-PŁ). We ascertain that the SGDA-RR can achieve a faster rate than its with-replacement counterpart, which agrees with the existing theory on without-replacement SGD for minimization. Lastly, we provided complexity lower bounds for simGDA with an arbitrarily fixed step size ratio $r$, demonstrating that the full-batch upper bound with $r \gtrsim \kappa_2^2$ for PŁ($\Phi$)-PŁ functions is tight.

Possible future directions include widening our results beyond sim/altSGDA (*e.g.*, extra-gradient or optimistic GDA) and beyond RR (*e.g.*, single/adversarial shuffling). As also discussed in Section 5, an interesting open question remains open: can we identify tight convergence rates for stochastic (with-/without-replacement) and/or deterministic GDA with step size ratio $r$ satisfying $\kappa_2 \lesssim r \lesssim \kappa_2^2$, for general nonconvex-PŁ problems?

---

[8]As described in Section 4.2, AGDA-RR uses only one-side gradient ($\nabla_1$ or $\nabla_2$) at each iteration; given a fixed budget of gradient computations, it should access components twice as many times as SGDA-RR. Hence, we report the values at every other iteration of AGDA & AGDA-RR, for a fair comparison.

ACKNOWLEDGMENTS

This work was supported by Institute of Information & communications Technology Planning & evaluation (IITP) grant (No. 2019-0-00075, Artificial Intelligence Graduate School Program (KAIST)) funded by the Korea government (MSIT). The work was also supported by the National Research Foundation of Korea (NRF) grant (No. NRF-2019R1A5A1028324) funded by the Korea government (MSIT). CY acknowledges support from a grant funded by Samsung Electronics Co., Ltd.

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

CONTENTS

## A  MINI-BATCH SGDA-RR AND CONVERGENCE RATES

In this appendix, we present an algorithm that extends simSGDA-RR and altSGDA-RR by using mini-batches of size $b \geq 1$. For simplicity, we assume that the number of components $n$ is an integer multiple of the mini-batch size $b$ in our analysis; *i.e.*, $n = bq$ for some integer $q \geq 1$. One can extend this to the case when $n$ is not necessarily a multiple of $b$ (*e.g.*, $n = b(q-1) + s$, where $q \geq 1$, $s \in [b]$) so that there are $q-1$ mini-batches of size $b$ and one more mini-batch of size $s \leq b$.

---

**Algorithm 2** Mini-batch simSGDA/altSGDA-RR

---

1: **Given:** The number of components $n = b(q-1) + s$ ($q$: number of iterations per epoch); mini-batch size $b$; the number of epochs $K$; step sizes $\alpha, \beta > 0$
2: **Initialize:** $(\boldsymbol{x}_0^1; \boldsymbol{y}_0^1) \in \mathbb{R}^{d_x} \times \mathbb{R}^{d_y}$
3: **for** $k \in [K]$ **do**
4:     Sample $\sigma_k \sim \mathrm{Unif}(\mathbb{S}_n)$          ▷ RR: uniformly randomly shuffle the indices every epoch
5:     **for** $t \in [q]$ **do**
6:         $\mathcal{B}_t^k := \{\sigma_k(j) : b(t-1) < j \leq bt, j \in [n]\}$   ▷ Mini-batch : a set of component indices
7:         $\boldsymbol{x}_t^k = \boldsymbol{x}_{t-1}^k - \frac{\alpha}{b} \sum_{i \in \mathcal{B}_t^k} \nabla_1 f_i(\boldsymbol{x}_{t-1}^k; \boldsymbol{y}_{t-1}^k)$
8:         **if** simSGDA-RR **then**
9:             $\boldsymbol{y}_t^k = \boldsymbol{y}_{t-1}^k + \frac{\beta}{b} \sum_{i \in \mathcal{B}_t^k} \nabla_2 f_i(\boldsymbol{x}_{t-1}^k; \boldsymbol{y}_{t-1}^k)$          ▷ simultaneous update: $\boldsymbol{x}$ & $\boldsymbol{y}$
10:         **else if** altSGDA-RR **then**
11:             $\boldsymbol{y}_t^k = \boldsymbol{y}_{t-1}^k + \frac{\beta}{b} \sum_{i \in \mathcal{B}_t^k} \nabla_2 f_i(\boldsymbol{x}_t^k; \boldsymbol{y}_{t-1}^k)$          ▷ alternating update: $\boldsymbol{x} \to \boldsymbol{y}$
12:     $(\boldsymbol{x}_0^{k+1}; \boldsymbol{y}_0^{k+1}) = (\boldsymbol{x}_{n/b}^k; \boldsymbol{y}_{n/b}^k)$

---

Next, we illustrate the generalized versions of our main results (Theorems 1 and 2) for Algorithm 2 with mini-batches of size $b \geq 1$. Let us assume $n \geq 2$ because the case $n = 1$ trivially boils down to simGDA or altGDA. We defer the proofs for simultaneous updates to Appendix C. We present the parts that change in the proof for alternating updates in Appendix D.

**Theorem 4** (Nonconvex-PŁ, mini-batch SGDA-RR). *Suppose $f$ satisfies Assumptions 1, 2, 3, and 4. Let $\lambda = 4$. Choose the step sizes $\alpha$ and $\beta$ by $\alpha = \beta/r$ for some $r \geq 14\kappa_2^2$ and*

$$\beta = b \cdot \min \left\{ \frac{1}{6Ln\sqrt{1 + \frac{n-b}{n-1} \cdot \frac{A}{n}}}, \frac{1}{c} \left( \frac{V_\lambda(\boldsymbol{z}_0^1)}{Ln^2(\frac{n-b}{n-1})BK} \right)^{\frac{1}{3}} \right\},$$

*for some numerical constant $c > 0$. Then, mini-batch simSGDA-RR and altSGDA-RR with mini-batch size $b$ (a divisor of $n$) satisfy*

$$\frac{1}{K} \sum_{k=1}^K \mathbb{E} \left\| \nabla \Phi(\boldsymbol{x}_0^k) \right\|^2 \leq \frac{6rLV_\lambda(\boldsymbol{z}_0^1)}{K} \sqrt{1 + \left( \frac{n-b}{n-1} \right) \frac{A}{n}} + 2cr \left( \frac{L^2 B V_\lambda(\boldsymbol{z}_0^1)^2}{nK^2} \cdot \frac{n-b}{n-1} \right)^{1/3}.$$

**Theorem 5** (PŁ($\Phi$)-PŁ, mini-batch SGDA-RR). *Suppose $f$ satisfies Assumptions 1, 2, 3, 4, and 5. Let $\lambda = 4$. Choose the step sizes $\alpha$ and $\beta$ by $\alpha = \beta/r$ for some $r \geq 14\kappa_2^2$ and*

$$\beta = b \cdot \min \left\{ \frac{1}{6Ln\sqrt{1 + \frac{n-b}{n-1} \cdot \frac{A}{n}}}, \frac{2r}{\mu_1 nK} \max \left\{ 1, \log \left( \frac{V_\lambda(\boldsymbol{z}_0^1)\mu_1 nK^2}{8c^3\kappa_1^2 r^3 \left( \frac{n-b}{n-1} \right) B} \right) \right\} \right\},$$

*for some numerical constant $c > 0$. Then, mini-batch simSGDA-RR and altSGDA-RR with mini-batch size $b$ (a divisor of $n$) satisfy*

$$\mathbb{E}[V_\lambda(\boldsymbol{z}_0^{K+1})] \leq \mathcal{O} \left( V_\lambda(\boldsymbol{z}_0^1) \cdot \exp \left( -\frac{K}{12\kappa_1 r\sqrt{1 + \frac{n-b}{n-1}\frac{A}{n}}} \right) \right) + \tilde{\mathcal{O}} \left( \frac{\kappa_1^2 r^3 B}{\mu_1 nK^2} \right) \cdot \frac{n-b}{n-1}.$$

As a side remark, some works consider a sampling method called *b-minibatch sampling* where all the elements in each mini-batch are distinct (*i.e.*, without-replacement component sampling per mini-batch), *e.g.*, Loizou et al. (2021, Definition 2.1). However, there is a significant gap between this method and ours: any two distinct mini-batches sampled by the $b$-minibatch sampling can intersect with each other (*i.e.*, mini-batches are sampled with replacement), whereas, in each epoch of our Algorithm 2, all the mini-batches are mutually disjoint.

# B    TECHNICAL PROPOSITIONS

**Notation.** Throughout this appendix, we use $\mathcal{X} = \mathbb{R}^{d_x}$ and $\mathcal{Y} = \mathbb{R}^{d_y}$. Given a closed set $\mathcal{S} \subset \mathbb{R}^d$, we denote the set of all projection(s) of $\boldsymbol{v} \in \mathbb{R}^d$ onto $\mathcal{S}$, *i.e.*, the nearest point(s) in $\mathcal{S}$ from $\boldsymbol{v}$, by $\Pi_{\mathcal{S}}(\boldsymbol{v}) := \arg\min_{\boldsymbol{w} \in \mathcal{S}} \|\boldsymbol{v} - \boldsymbol{w}\|$.

## B.1    FUNCTION CLASSES: PŁ CONDITION, SMOOTHNESS, AND MORE

**Proposition 6** ($\kappa \geq 1$)**.** *Let $g$ be an $L$-smooth function which is bounded below by $g^*$. Then, for any $\boldsymbol{x}$,*
$$\|\nabla g(\boldsymbol{x})\|^2 \leq 2L\left[g(\boldsymbol{x}) - g^*\right].$$
*If $g$ is $\mu$-PŁ as well, then $\mu \leq L$. Consequently, the condition number $\kappa := L/\mu$ of $g$ is $\geq 1$.*

*Proof.* Since $g$ is $L$-smooth, for any $\boldsymbol{x}$ and $\boldsymbol{y}$,
$$g^* \leq g(\boldsymbol{y}) \leq g(\boldsymbol{x}) + \langle \nabla g(\boldsymbol{x}), \boldsymbol{y} - \boldsymbol{x} \rangle + \frac{L}{2} \|\boldsymbol{y} - \boldsymbol{x}\|^2. \tag{6}$$
Now define a convex quadratic function $h_x(\boldsymbol{y})$ of $\boldsymbol{y}$ as
$$h_x(\boldsymbol{y}) := g(\boldsymbol{x}) + \langle \nabla g(\boldsymbol{x}), \boldsymbol{y} - \boldsymbol{x} \rangle + \frac{L}{2} \|\boldsymbol{y} - \boldsymbol{x}\|^2.$$
Since its gradient is
$$\nabla h_x(\boldsymbol{y}) = \nabla g(\boldsymbol{x}) + L(\boldsymbol{y} - \boldsymbol{x}),$$
$\boldsymbol{y}^* := \boldsymbol{x} - \frac{1}{L}\nabla g(\boldsymbol{x})$ is a minimum of $h_x$. Plugging $\boldsymbol{y} = \boldsymbol{y}^*$ to the equation (6), we get
$$g^* \leq g(\boldsymbol{x}) + \left\langle \nabla g(\boldsymbol{x}), -\frac{1}{L}\nabla g(\boldsymbol{x}) \right\rangle + \frac{L}{2} \left\| -\frac{1}{L}\nabla g(\boldsymbol{x}) \right\|^2 = g(\boldsymbol{x}) - \frac{1}{2L} \|\nabla g(\boldsymbol{x})\|^2.$$
Rearranging the terms,
$$\|\nabla g(\boldsymbol{x})\|^2 \leq 2L\left[g(\boldsymbol{x}) - g^*\right].$$
If we additionally utilize PŁ inequality with $g^* := \min g(\boldsymbol{x})$,
$$\|\nabla g(\boldsymbol{x})\|^2 \geq 2\mu\left[g(\boldsymbol{x}) - g^*\right],$$
we directly yield $\mu \leq L$ and thus $\kappa = L/\mu \geq 1$. $\qquad\square$

**Definition 1** (Karimi et al. (2016))**.** *Consider $g : \mathcal{X} \to \mathbb{R}$. Let $\boldsymbol{x}_p \in \Pi_{\mathcal{X}^*}(\boldsymbol{x})$ be a projection of $\boldsymbol{x}$ onto the optimal set $\mathcal{X}^* = \arg\min_{\boldsymbol{x} \in \mathcal{X}} g(\boldsymbol{x})$.*

    *(1) We say $g$ satisfies $\mu$-strong convexity (SC) if $g(\boldsymbol{x}') \geq g(\boldsymbol{x}) + \langle \nabla g(\boldsymbol{x}), \boldsymbol{x}' - \boldsymbol{x} \rangle + \frac{\mu}{2}\|\boldsymbol{x}' - \boldsymbol{x}\|^2$ for any $\boldsymbol{x}, \boldsymbol{x}' \in \mathcal{X}$.*

    *(2) We say $g$ satisfies $\mu$-restricted secant inequality (RSI) if $\langle \nabla g(\boldsymbol{x}), \boldsymbol{x} - \boldsymbol{x}_p \rangle \geq \mu\|\boldsymbol{x}_p - \boldsymbol{x}\|^2$ for any $\boldsymbol{x} \in \mathcal{X}$.*

    *(3) We say $g$ satisfies $\mu$-error bound (EB) condition if $\|\nabla g(\boldsymbol{x})\| \geq \mu\|\boldsymbol{x}_p - \boldsymbol{x}\|$ for any $\boldsymbol{x} \in \mathcal{X}$.*

    *(4) We say $g$ satisfies $\mu$-quadratic growth (QG) condition if $g(\boldsymbol{x}) - \min_{\boldsymbol{x}'} g(\boldsymbol{x}') \geq \frac{\mu}{2}\|\boldsymbol{x}_p - \boldsymbol{x}\|^2$ for any $\boldsymbol{x} \in \mathcal{X}$.*

**Proposition 7.** *From Definition 1, The following implications are true.*

- *$\mu$-SC implies $\mu$-PŁ and $\mu$-RSI.*

- *$\mu$-PŁ implies $\mu$-QG and $\mu$-EB.*

- *$\mu$-RSI implies $\mu$-EB.*

- *$\mu$-EB and $L$-smoothness together imply $(\mu^2/L)$-PŁ.*

*Proof.* Most of the proofs originated from Karimi et al. (2016, Theorem 2).

(SC ⇒ PŁ) Substitute $\boldsymbol{x}$ to $\boldsymbol{x}_p$ and $\boldsymbol{x}'$ to $\boldsymbol{x}$, respectively, from Definition 1.(1).

(PŁ ⇒ QG & EB) See the proof in Karimi et al. (2016, Theorem 2)

(SC ⇒ RSI) We know $\mu$-SC ⇒ $\mu$-PŁ ⇒ $\mu$-QG. From Definition 1.(1) & 1.(4),

$$\langle \nabla g(\boldsymbol{x}), \boldsymbol{x} - \boldsymbol{x}_p \rangle \stackrel{\text{SC}}{\geq} g(\boldsymbol{x}) - g(\boldsymbol{x}_p) + \frac{\mu}{2} \|\boldsymbol{x}_p - \boldsymbol{x}\|^2$$
$$\stackrel{\text{QG}}{\geq} \frac{\mu}{2} \|\boldsymbol{x}_p - \boldsymbol{x}\|^2 + \frac{\mu}{2} \|\boldsymbol{x}_p - \boldsymbol{x}\|^2 = \mu \|\boldsymbol{x}_p - \boldsymbol{x}\|^2 .$$

This implies $\mu$-RSI.

(RSI ⇒ EB) See the proof in Karimi et al. (2016, Theorem 2).

(EB & smooth ⇒ PŁ) We use $\nabla g(\boldsymbol{x}_p) = \boldsymbol{0}$. By $L$-smoothness and $\mu$-EB condition,

$$g(\boldsymbol{x}) - g(\boldsymbol{x}_p) \stackrel{\text{smooth}}{\leq} \langle \nabla g(\boldsymbol{x}_p), \boldsymbol{x} - \boldsymbol{x}_p \rangle + \frac{L}{2} \|\boldsymbol{x} - \boldsymbol{x}_p\|^2 = \frac{L}{2} \|\boldsymbol{x} - \boldsymbol{x}_p\|^2$$
$$\stackrel{\text{EB}}{\leq} \frac{L}{2\mu^2} \|\nabla g(\boldsymbol{x})\|^2 .$$

This implies $(\mu^2/L)$-PŁ condition on $g$. □

**Proposition 8** (Lipschitz continuity-like property of $\boldsymbol{y}^*(\boldsymbol{x})$). *For an $L$-smooth function $g : \mathcal{X} \times \mathcal{Y} \to \mathbb{R}$, suppose $-g(\boldsymbol{x}; \cdot)$ is $\mu_2$-PŁ. Let $\kappa_2 = L/\mu_2$.*

*Consider any $\boldsymbol{x}_0, \boldsymbol{x}_1 \in \mathcal{X}$. For any $\boldsymbol{y}_0^* \in \mathcal{Y}_{\boldsymbol{x}_0}^* = \arg\max_{\boldsymbol{y} \in \mathcal{Y}} g(\boldsymbol{x}_0; \boldsymbol{y})$, there exists a $\boldsymbol{y}_1^* \in \mathcal{Y}_{\boldsymbol{x}_1}^* = \arg\max_{\boldsymbol{y} \in \mathcal{Y}} g(\boldsymbol{x}_1; \boldsymbol{y})$ such that $\|\boldsymbol{y}_0^* - \boldsymbol{y}_1^*\| \leq \kappa_2 \|\boldsymbol{x}_0 - \boldsymbol{x}_1\|$.*

*In fact, it is enough to choose $\boldsymbol{y}_1^*$ as a projection of $\boldsymbol{y}_0^*$ onto the set $\mathcal{Y}_{\boldsymbol{x}_1}^*$, namely, $\boldsymbol{y}_1^* \in \Pi_{\mathcal{Y}_{\boldsymbol{x}_1}^*}(\boldsymbol{y}_0^*)$.*

*Proof.* We borrow the proof from Nouiehed et al. (2019, Lemma A.3).

Recall $\Phi(\boldsymbol{x}) := \max_{\boldsymbol{y}' \in \mathcal{Y}} g(\boldsymbol{x}; \boldsymbol{y}')$. By PŁ inequality and smoothness of $g$,

$$2\mu_2 \left(\Phi(\boldsymbol{x}_1) - g(\boldsymbol{x}_1; \boldsymbol{y}_0^*)\right) \leq \|\nabla_2 g(\boldsymbol{x}_1; \boldsymbol{y}_0^*)\|^2$$
$$= \|\nabla_2 g(\boldsymbol{x}_1; \boldsymbol{y}_0^*) - \nabla_2 g(\boldsymbol{x}_0; \boldsymbol{y}_0^*)\|^2 \leq L^2 \|\boldsymbol{x}_1 - \boldsymbol{x}_0\|^2 .$$

The second equality applies $\nabla_2 g(\boldsymbol{x}_0; \boldsymbol{y}_0^*) = \boldsymbol{0}$, since $\boldsymbol{y}_0^* \in \arg\max_{\boldsymbol{y}} g(\boldsymbol{x}_0; \boldsymbol{y})$.

Moreover, note that $-g(\boldsymbol{x}_1; \cdot)$ satisfies $\mu_2$-QG condition ($\because$ Proposition 7). To apply this, we utilize our choice of $\boldsymbol{y}_1^*$:

$$\Phi(\boldsymbol{x}_1) - g(\boldsymbol{x}_1; \boldsymbol{y}_0^*) \geq \frac{\mu_2}{2} \|\boldsymbol{y}_1^* - \boldsymbol{y}_0^*\|^2 .$$

As a result, we have $\mu_2^2 \|\boldsymbol{y}_0^* - \boldsymbol{y}_1^*\|^2 \leq L^2 \|\boldsymbol{x}_0 - \boldsymbol{x}_1\|^2$. This completes the proof. □

**Proposition 9** (Smoothness of primal function). *Consider the same function $g$ as Proposition 8. Then, the function $\Phi(\boldsymbol{x}) := \max_{\boldsymbol{y}' \in \mathcal{Y}} g(\boldsymbol{x}; \boldsymbol{y}')$ is differentiable with*

$$\nabla \Phi(\boldsymbol{x}) = \nabla_1 g(\boldsymbol{x}; \boldsymbol{y}^*(\boldsymbol{x})), \quad \textit{regardless of the choice of } \boldsymbol{y}^*(\boldsymbol{x}) \in \arg\max_{\boldsymbol{y}' \in \mathcal{Y}} g(\boldsymbol{x}; \boldsymbol{y}').$$

*Moreover, $\Phi$ is $L(\kappa_2 + 1)$-smooth, where $\kappa_2 = L/\mu_2$.*

*Proof.* This is already proved in Lemma A.5 of Nouiehed et al. (2019). However, we present a bit different proof without using second-order Taylor expansion. To start, recall $\mathcal{Y}_{\boldsymbol{x}}^* := \arg\max_{\boldsymbol{y} \in \mathcal{Y}} g(\boldsymbol{x}; \boldsymbol{y})$. That is, we could choose any $\boldsymbol{y}^*(\boldsymbol{x}) \in \mathcal{Y}_{\boldsymbol{x}}^*$.

We first show the differentiability of $\Phi$. Fix a unit vector $\boldsymbol{u} \in \mathcal{X} = \mathbb{R}^{d_x}: \|\boldsymbol{u}\| = 1$. Let any $h > 0$. We first claim that there exists a path $\boldsymbol{p} : (-h, h] \to \mathcal{Y} = \mathbb{R}^{d_y}$ which is continuous at $t = 0$ and $\boldsymbol{p}(t) \in \mathcal{Y}_{(\boldsymbol{x} + t\boldsymbol{u})}^*$. In fact, let $\boldsymbol{p}(t)$ be a projection of $\boldsymbol{y}^*(\boldsymbol{x})$ (that we chose) onto the set $\mathcal{Y}_{(\boldsymbol{x} + t\boldsymbol{u})}^*$.

Then, $\boldsymbol{p}(0) = \boldsymbol{y}^*(\boldsymbol{x})$, and by Proposition 8, we have $\|\boldsymbol{p}(0) - \boldsymbol{p}(t)\| \leq \kappa_2 \|\boldsymbol{x} - (\boldsymbol{x} + t\boldsymbol{u})\| = \kappa_2 t$. This shows the continuity of $\boldsymbol{p}(t)$ at $t = 0$. Now, note that there exists a $t_1 \in (0, h)$ such that,

$$
\begin{aligned}
&\Phi(\boldsymbol{x} + h\boldsymbol{u}) - \Phi(\boldsymbol{x}) \\
&= g(\boldsymbol{x} + h\boldsymbol{u}; \boldsymbol{p}(h)) - g(\boldsymbol{x}; \boldsymbol{p}(0)) \\
&= \{g(\boldsymbol{x} + h\boldsymbol{u}; \boldsymbol{p}(h)) - g(\boldsymbol{x} + h\boldsymbol{u}; \boldsymbol{p}(0))\} + \{g(\boldsymbol{x} + h\boldsymbol{u}; \boldsymbol{p}(0)) - g(\boldsymbol{x}; \boldsymbol{p}(0))\} \\
&\geq 0 + \langle \nabla_1 g(\boldsymbol{x} + t_1 \boldsymbol{u}; \boldsymbol{p}(0)), h\boldsymbol{u} \rangle,
\end{aligned}
$$

by mean value theorem (applied to the first argument). We have the inequality at the last line because $g(\boldsymbol{x} + h\boldsymbol{u}; \boldsymbol{p}(h)) \geq g(\boldsymbol{x} + h\boldsymbol{u}; \boldsymbol{p}(0))$, since $\boldsymbol{p}(h) \in \mathcal{Y}^*_{(\boldsymbol{x}+h\boldsymbol{u})}$. With a similar logic, there exists a $t_2 \in (0, h)$ such that,

$$
\begin{aligned}
&\Phi(\boldsymbol{x} + h\boldsymbol{u}) - \Phi(\boldsymbol{x}) \\
&= g(\boldsymbol{x} + h\boldsymbol{u}; \boldsymbol{p}(h)) - g(\boldsymbol{x}; \boldsymbol{p}(0)) \\
&= \{g(\boldsymbol{x} + h\boldsymbol{u}; \boldsymbol{p}(h)) - g(\boldsymbol{x}; \boldsymbol{p}(h))\} + \{g(\boldsymbol{x}; \boldsymbol{p}(h)) - g(\boldsymbol{x}; \boldsymbol{p}(0))\} \\
&\leq \langle \nabla_1 g(\boldsymbol{x} + t_2 \boldsymbol{u}; \boldsymbol{p}(h)), h\boldsymbol{u} \rangle + 0.
\end{aligned}
$$

To combine these two inequalities into a single line,

$$
\langle \nabla_1 g(\boldsymbol{x} + t_1 \boldsymbol{u}; \boldsymbol{p}(0)), \boldsymbol{u} \rangle \leq \frac{\Phi(\boldsymbol{x} + h\boldsymbol{u}) - \Phi(\boldsymbol{x})}{h} \leq \langle \nabla_1 g(\boldsymbol{x} + t_2 \boldsymbol{u}; \boldsymbol{p}(h)), \boldsymbol{u} \rangle.
$$

Using the continuity of $p(\cdot)$ and $\nabla_1 g(\cdot; \cdot)$ ($\because g$ has Lipschitz continuous gradient), we can deduce that the directional derivative of $\Phi$ in a direction $\boldsymbol{u}$ (denoted by $D_{\boldsymbol{u}}\Phi$) is in fact

$$
D_{\boldsymbol{u}}\Phi(\boldsymbol{x}) = \langle \nabla_1 g(\boldsymbol{x}; \boldsymbol{y}^*(\boldsymbol{x})), \boldsymbol{u} \rangle,
$$

by taking the limit $h \to 0+$. Since $\boldsymbol{u}$ is arbitrary, we can conclude that $\nabla\Phi(\boldsymbol{x}) = \nabla_1 g(\boldsymbol{x}; \boldsymbol{y}^*(\boldsymbol{x}))$.

The proof of Lipschitz smoothness of $\Phi$ exactly follows the proof by Nouiehed et al. (2019). Consider any $\boldsymbol{x}_0, \boldsymbol{x}_1 \in \mathcal{X}$. As in Proposition 8, choose any $\boldsymbol{y}_0^* \in \mathcal{Y}^*_{\boldsymbol{x}_0}$ and $\boldsymbol{y}_1^* \in \Pi_{\mathcal{Y}^*_{\boldsymbol{x}_1}}(\boldsymbol{y}_0^*)$. Then,

$$
\begin{aligned}
&\|\nabla\Phi(\boldsymbol{x}_0) - \nabla\Phi(\boldsymbol{x}_1)\| \\
&= \|\nabla_1 g(\boldsymbol{x}_0; \boldsymbol{y}_0^*) - \nabla_1 g(\boldsymbol{x}_1; \boldsymbol{y}_1^*)\| \\
&\leq \|\nabla_1 g(\boldsymbol{x}_0; \boldsymbol{y}_0^*) - \nabla_1 g(\boldsymbol{x}_1; \boldsymbol{y}_0^*)\| + \|\nabla_1 g(\boldsymbol{x}_1; \boldsymbol{y}_0^*) - \nabla_1 g(\boldsymbol{x}_1; \boldsymbol{y}_1^*)\| \\
&\leq L\{\|\boldsymbol{x}_0 - \boldsymbol{x}_1\| + \|\boldsymbol{y}_0^* - \boldsymbol{y}_1^*\|\} \\
&\leq L(1 + \kappa_2)\|\boldsymbol{x}_0 - \boldsymbol{x}_1\|.
\end{aligned}
$$

The last inequality holds because of Proposition 8. □

**Proposition 10** ($\boldsymbol{x}$-side PŁ $\Rightarrow$ primal PŁ). *Suppose $g : \mathcal{X} \times \mathcal{Y} \to \mathbb{R}$ is $L$-smooth and two-sided PŁ with constants $\mu_1$ and $\mu_2$. Then, $g$ satisfies primal PŁ condition: the function $\Phi(\boldsymbol{x}) := \max_{\boldsymbol{y}' \in \mathcal{Y}} g(\boldsymbol{x}; \boldsymbol{y}')$ is $\mu_1$-PŁ. As a result, a smooth two-sided PŁ function is PŁ($\Phi$)-PŁ.*

*Proof.* See Lemma A.3 of Yang et al. (2020). □

**Definition 2.** *Consider $g : \mathcal{X} \times \mathcal{Y} \to \mathbb{R}$. Then, the point $(\boldsymbol{x}^*; \boldsymbol{y}^*) \in \mathcal{X} \times \mathcal{Y}$ is called*

  *(i) a stationary point of $g$ if $\nabla_1 g(\boldsymbol{x}^*; \boldsymbol{y}^*) = \nabla_2 g(\boldsymbol{x}^*; \boldsymbol{y}^*) = 0$.*

  *(ii) a saddle point of $g$ if $g(\boldsymbol{x}^*; \boldsymbol{y}) \leq g(\boldsymbol{x}^*; \boldsymbol{y}^*) \leq g(\boldsymbol{x}; \boldsymbol{y}^*)$ for all $\boldsymbol{x}, \boldsymbol{y}$.*

  *(iii) a global minimax point of $g$ if $g(\boldsymbol{x}^*; \boldsymbol{y}) \leq g(\boldsymbol{x}^*; \boldsymbol{y}^*) \leq \max_{\boldsymbol{y}'} g(\boldsymbol{x}; \boldsymbol{y}')$ for all $\boldsymbol{x}, \boldsymbol{y}$.*

  *(iv) a global maximin point of $g$ if $\min_{\boldsymbol{x}'} g(\boldsymbol{x}'; \boldsymbol{y}) \leq g(\boldsymbol{x}^*; \boldsymbol{y}^*) \leq g(\boldsymbol{x}; \boldsymbol{y}^*)$ for all $\boldsymbol{x}, \boldsymbol{y}$.*

**Proposition 11.** *Consider a function $g : \mathcal{X} \times \mathcal{Y} \to \mathbb{R}$.*

  *(1) In general, a saddle point of $g$ is a global minimax/maximin point.*

  *(2) Let $\Phi(\boldsymbol{x}) := \max_{\boldsymbol{y}} g(\boldsymbol{x}; \boldsymbol{y})$ and $\Phi^* := \min_{\boldsymbol{x}} \Phi(\boldsymbol{x})$ be well-defined. Let $\lambda > 0$ be a constant. In general, a point $(\boldsymbol{x}^*; \boldsymbol{y}^*)$ is a global minimax point of $g$ if and only if*

$$
V_\lambda(\boldsymbol{x}^*; \boldsymbol{y}^*) := \lambda[\Phi(\boldsymbol{x}) - \Phi^*] + [\Phi(\boldsymbol{x}) - g(\boldsymbol{x}; \boldsymbol{y})] = 0.
$$

*(3) If g is smooth nonconvex-PŁ, then a global minimax point is a stationary point.*

*(4) If g is PŁ(Φ)-PŁ, then there exists a global minimax point $(\boldsymbol{x}^*; \boldsymbol{y}^*)$ of g. As a result, if g is also smooth, then the point $(\boldsymbol{x}^*; \boldsymbol{y}^*)$ is a stationary point.*

*(5) If g is smooth two-sided PŁ, every stationary point is a saddle point. As a result, there exists a saddle point $(\boldsymbol{x}^*; \boldsymbol{y}^*)$ of g.*

In particular, smooth two-sided PŁ functions enjoy the "minimax theorem," which establishes "minimax = maximin."

*Proof.* (1) (saddle point $\Rightarrow$ global minimax & global maximin) This is straightforward by the definitions: for any $\boldsymbol{x}$ and $\boldsymbol{y}$,

$$\min_{\boldsymbol{x}'} g(\boldsymbol{x}'; \boldsymbol{y}) \leq g(\boldsymbol{x}^*; \boldsymbol{y}) \leq g(\boldsymbol{x}^*; \boldsymbol{y}^*) \leq g(\boldsymbol{x}; \boldsymbol{y}^*) \leq \max_{\boldsymbol{y}'} g(\boldsymbol{x}; \boldsymbol{y}').$$

(2) (global minimax $\iff V_\lambda = 0$) The terms $\Phi(\boldsymbol{x}) - \Phi^*$ and $\Phi(\boldsymbol{x}) - g(\boldsymbol{x}; \boldsymbol{y})$ are non-negative. Hence, $V_\lambda(\boldsymbol{x}; \boldsymbol{y})$ is non-negative, and $V_\lambda(\boldsymbol{x}^*; \boldsymbol{y}^*) = 0$ if and only if $\Phi^* = \Phi(\boldsymbol{x}^*) = g(\boldsymbol{x}^*; \boldsymbol{y}^*)$, which is equivalent to the global minimax point condition.

(3) (smooth nonconvex-PŁ: global minimax $\Rightarrow$ stationary) Suppose $(\boldsymbol{x}^*; \boldsymbol{y}^*)$ is a global minimax point. Since $g(\boldsymbol{x}^*; \boldsymbol{y}) \leq g(\boldsymbol{x}^*; \boldsymbol{y}^*)$ for any $\boldsymbol{y}$, $\Phi(\boldsymbol{x}^*) = \max_y g(\boldsymbol{x}^*; \boldsymbol{y}) = g(\boldsymbol{x}^*; \boldsymbol{y}^*)$. Thus, $\Phi$ has a minimum $g(\boldsymbol{x}^*; \boldsymbol{y}^*)$ at $\boldsymbol{x} = \boldsymbol{x}^*$. By Proposition 9, $\Phi(\cdot)$ is a differentiable function and we have

$$\nabla_1 g(\boldsymbol{x}^*; \boldsymbol{y}^*) = \nabla \Phi(\boldsymbol{x}^*) = 0.$$

Also, since a differentiable function $g(\boldsymbol{x}^*; \boldsymbol{y})$ has a maximum at $\boldsymbol{y} = \boldsymbol{y}^*$, we also have $\nabla_2 g(\boldsymbol{x}^*; \boldsymbol{y}^*) = 0$. Therefore, $(\boldsymbol{x}^*; \boldsymbol{y}^*)$ is a stationary point.

(4) (PŁ(Φ)-PŁ: $\exists$ global minimax) Let $\boldsymbol{x}^* \in \arg\min_{\boldsymbol{x}} \Phi(\boldsymbol{x})$ and $\boldsymbol{y}^* \in \arg\max_{\boldsymbol{y}} f(\boldsymbol{x}^*; \boldsymbol{y})$. Then, $f(\boldsymbol{x}^*, \boldsymbol{y}^*) = \Phi(\boldsymbol{x}^*) = \Phi^*$. as noted in (2), $(\boldsymbol{x}^*, \boldsymbol{y}^*)$ is a global minimax point. By (3), it is in fact a stationary point, when $g$ is smooth as well.

(5) (smooth two-sided PŁ: stationary $\Rightarrow$ saddle) Let $(\boldsymbol{x}^*; \boldsymbol{y}^*)$ be a stationary point. By PŁ inequalities, for any $\boldsymbol{x}$ and $\boldsymbol{y}$,

$$0 = \|\nabla_2 g(\boldsymbol{x}^*; \boldsymbol{y}^*)\|^2 \geq 2\mu_2(\max_{\boldsymbol{y}} g(\boldsymbol{x}^*; \boldsymbol{y}) - g(\boldsymbol{x}^*; \boldsymbol{y}^*)) \geq 0,$$

$$0 = \|\nabla_1 g(\boldsymbol{x}^*; \boldsymbol{y}^*)\|^2 \geq 2\mu_1(g(\boldsymbol{x}^*; \boldsymbol{y}^*) - \min_{\boldsymbol{x}} g(\boldsymbol{x}; \boldsymbol{y}^*)) \geq 0.$$

Since $\mu_1, \mu_2 > 0$, these imply $\max_{\boldsymbol{y}} g(\boldsymbol{x}^*; \boldsymbol{y}) = g(\boldsymbol{x}^*; \boldsymbol{y}^*) = \min_{\boldsymbol{x}} g(\boldsymbol{x}; \boldsymbol{y}^*)$. Thus, $(\boldsymbol{x}^*; \boldsymbol{y}^*)$ is a saddle point. Note that (4) and Proposition 10 together proves the existence of a stationary point of $g$. Therefore, there must exists a saddle point, which is also pointed out by Guo et al. (2020, Lemma 8). This concludes the proof. $\square$

We remark that, in the proof above, (3) is false for general (nonconvex-nonconcave) functions. Only *local* minimax point can ensure stationarity (Jin et al., 2020). As remarked by Jin et al. (2020) (Figure 2 of their paper), the function $xy - \cos(y)$ has non-stationary global minimax points $(0, \pm\pi)$.

The following two propositions are for showing that general two-sided PŁ function may not have a *differential Stackelberg equilibrium* defined as Li et al. (2022, Definition 3.1).

**Proposition 12.** *Let g be a $\mu$-strongly convex function on $\mathbb{R}^n$. Consider any matrix $\boldsymbol{M} \in \mathbb{R}^{n \times m}$ with a positive rank. Suppose that $\theta$ is the smallest nonzero singular value of $\boldsymbol{M}$. Then $g(\boldsymbol{M}\boldsymbol{y})$ is a $\mu\theta^2$-PŁ function of $\boldsymbol{y} \in \mathbb{R}^m$.*

*Proof.* See Karimi et al. (2016, Appendix B) for the proof. $\square$

**Proposition 13.** *Consider a twice continuously differentiable strongly-convex-strongly-concave function $h : \mathbb{R}^r \times \mathbb{R}^s \to \mathbb{R}$. That is, for some constants $\mu_1, \mu_2 > 0$, $h(\boldsymbol{x}; \boldsymbol{y})$ is $\mu_1$-strongly-convex in $\boldsymbol{x}$ and $-h(\boldsymbol{x}; \boldsymbol{y})$ is $\mu_2$-strongly-convex in $\boldsymbol{y}$. Let $(\boldsymbol{x}^*; \boldsymbol{y}^*)$ be the unique stationary point of h. Of*

*course, it is a **differential Stackelberg equilibrium** of $h$. That is, if the hessian matrix $\nabla^2 h(\boldsymbol{x}^*; \boldsymbol{y}^*)$ at that point is written as*

$$\nabla^2 h(\boldsymbol{x}^*; \boldsymbol{y}^*) = \begin{bmatrix} \nabla^2_{1,1} h(\boldsymbol{x}^*; \boldsymbol{y}^*) & \nabla^2_{1,2} h(\boldsymbol{x}^*; \boldsymbol{y}^*) \\ \nabla^2_{2,1} h(\boldsymbol{x}^*; \boldsymbol{y}^*) & \nabla^2_{2,2} h(\boldsymbol{x}^*; \boldsymbol{y}^*) \end{bmatrix} = \begin{bmatrix} \boldsymbol{C} & \boldsymbol{B} \\ \boldsymbol{B}^\top & -\boldsymbol{A} \end{bmatrix},$$

*then $\boldsymbol{A}$ and $\boldsymbol{C} - \boldsymbol{B}\boldsymbol{A}^{-1}\boldsymbol{B}^\top$ are both positive definite matrices. Consider a function $g : \mathbb{R}^p \times \mathbb{R}^q \to \mathbb{R}$ defined by $g(\boldsymbol{x}; \boldsymbol{y}) = h(\boldsymbol{M}\boldsymbol{x}; \boldsymbol{N}\boldsymbol{y})$ for some matrices $\boldsymbol{M} \in \mathbb{R}^{r \times p}$, $\boldsymbol{N} \in \mathbb{R}^{s \times q}$. Then, $g$ is two-sided PŁ. Moreover, each stationary point of $g$ may not be a differential Stackelberg equilibrium in general, for example, when $s < q$.*

*Proof.* Because of Proposition 12, $g$ is clearly a two-sided PŁ function.

If $(\boldsymbol{x}; \boldsymbol{y})$ is a stationary point of $g$, then it must be an element of an affine set $\{(\boldsymbol{x}; \boldsymbol{y}) \in \mathbb{R}^p \times \mathbb{R}^q : \boldsymbol{M}\boldsymbol{x} = \boldsymbol{x}^*; \boldsymbol{N}\boldsymbol{y} = \boldsymbol{y}^*\}$. This is because

$$\nabla g(\boldsymbol{x}; \boldsymbol{y}) = \begin{bmatrix} \nabla_1 g(\boldsymbol{x}; \boldsymbol{y}) \\ \nabla_2 g(\boldsymbol{x}; \boldsymbol{y}) \end{bmatrix} = \begin{bmatrix} \boldsymbol{M}^\top \nabla_1 h(\boldsymbol{M}\boldsymbol{x}; \boldsymbol{N}\boldsymbol{y}) \\ \boldsymbol{N}^\top \nabla_2 h(\boldsymbol{M}\boldsymbol{x}; \boldsymbol{N}\boldsymbol{y}) \end{bmatrix} = \boldsymbol{0}$$

if and only if $\nabla_1 h(\boldsymbol{M}\boldsymbol{x}; \boldsymbol{N}\boldsymbol{y}) = \boldsymbol{0}$ and $\nabla_2 h(\boldsymbol{M}\boldsymbol{x}; \boldsymbol{N}\boldsymbol{y}) = \boldsymbol{0}$, being equivalent to $\boldsymbol{M}\boldsymbol{x} = \boldsymbol{x}^*$ and $\boldsymbol{N}\boldsymbol{y} = \boldsymbol{y}^*$. Furthermore, the hessian of $g$ at $(\boldsymbol{x}; \boldsymbol{y})$ is

$$\nabla^2 g(\boldsymbol{x}; \boldsymbol{y}) = \begin{bmatrix} \boldsymbol{M}^\top \nabla^2_{1,1} h(\boldsymbol{M}\boldsymbol{x}; \boldsymbol{N}\boldsymbol{y})\boldsymbol{M} & \boldsymbol{M}^\top \nabla^2_{1,2} h(\boldsymbol{M}\boldsymbol{x}; \boldsymbol{N}\boldsymbol{y})\boldsymbol{N} \\ \boldsymbol{N}^\top \nabla^2_{2,1} h(\boldsymbol{M}\boldsymbol{x}; \boldsymbol{N}\boldsymbol{y})\boldsymbol{M} & \boldsymbol{N}^\top \nabla^2_{2,2} h(\boldsymbol{M}\boldsymbol{x}; \boldsymbol{N}\boldsymbol{y})\boldsymbol{N} \end{bmatrix}$$
$$= \begin{bmatrix} \boldsymbol{M}^\top \boldsymbol{C}\boldsymbol{M} & \boldsymbol{M}^\top \boldsymbol{B}\boldsymbol{N} \\ (\boldsymbol{M}^\top \boldsymbol{B}\boldsymbol{N})^\top & -\boldsymbol{N}^\top \boldsymbol{A}\boldsymbol{N} \end{bmatrix}.$$

If $s < q$, the $q \times q$ matrix $\boldsymbol{N}^\top \boldsymbol{A}\boldsymbol{N}$ cannot have a full rank, thereby it cannot be even invertible. This implies the stationary point $(\boldsymbol{x}; \boldsymbol{y})$ cannot be a differential Stackelberg equilibrium. $\square$

## B.2 WITHOUT-REPLACEMENT SAMPLING

In this subsection, we provide a useful proposition for analysis of mini-batching approach under without-replacement sampling. We consider the case of mutually disjoint mini-batches in a whole epoch, not only applying without-replacement sampling to each individual mini-batch.

Consider a collection of $n$ vectors $\boldsymbol{v}_1, \ldots, \boldsymbol{v}_n \in \mathbb{R}^d$. Suppose we uniformly randomly sample a permutation $\sigma : [n] \to [n]$; *i.e.*, $\sigma \sim \mathrm{Unif}(\mathbb{S}_n)$. Define

$$\boldsymbol{m} = \frac{1}{n} \sum_{i=1}^n \boldsymbol{v}_i \quad \text{(sample mean)} \quad \text{and} \quad \tau^2 = \frac{1}{n} \sum_{i=1}^n \|\boldsymbol{v}_i - \boldsymbol{m}\|^2 \quad \text{(sample variance)}.$$

Fix any $b \in [n]$ and let $n = b(q-1) + s$ for some integers $q \geq 1$ and $s \in [b]$. Now, divide the indices $[n]$ into $q$ batches, with exactly $b$ items per batch (except for the last batch when $s < b$), as follows:

$$\mathcal{W}_t = \{\sigma(j) : b(t-1) < j \leq bt, j \in [n]\} \quad (t \in [q]).$$

For each batch $\mathcal{W}_t$, define

$$\boldsymbol{w}_t = \frac{1}{|\mathcal{W}_t|} \sum_{i \in \mathcal{W}_t} \boldsymbol{v}_i \quad \text{(batch mean)}.$$

For any $k \in [q-1]$, define

$$\boldsymbol{m}_k := \frac{1}{k} \sum_{t=1}^k \boldsymbol{w}_t \quad \text{(accumulative average of batch means over } 1 \leq t \leq k).$$

Of course, we may simply take $\boldsymbol{m}_q = \boldsymbol{m}$ (deterministically) for $k = q$. Thus, because of the randomness of $\sigma$, we can obtain the mean (vector) and the variance (scalar) of $\boldsymbol{m}_k$ as follows.

**Proposition 14** (Without-replacement sampling)**.** *Given the setup above, for any $k < q$ and $n > 1$,*

$$\mathbb{E}[\boldsymbol{m}_k] = \boldsymbol{m} \quad \text{and} \quad \mathbb{E}\left[\|\boldsymbol{m}_k - \boldsymbol{m}\|^2\right] = \frac{(n - bk)}{bk(n-1)}\tau^2.$$

*(Of course, if $k = q$ or $n = 1 = q$, $\mathbb{E}[\|\boldsymbol{m}_q - \boldsymbol{m}\|^2] = 0$ since $\boldsymbol{m}_q = \boldsymbol{m}$.)*

**Remark.** As a special case, if $n = bq$ (namely, $b$ divides $n$ and $s = b$), then for any $k \le q$,

$$\mathbb{E}\left[\|\boldsymbol{m}_k - \boldsymbol{m}\|^2\right] = \frac{(q-k)}{k(n-1)}\tau^2.$$

If we further assume $b = s = 1$ and $q = n$, this proposition recovers Lemma 1 of Mishchenko et al. (2020).

*Proof of Proposition 14.* Since $\sigma$ is a uniformly randomly sampled permutation, it is easy to obtain that

$$\mathbb{E}[\boldsymbol{v}_{\sigma(i)}] = \mathbb{E}[\boldsymbol{w}_t] = \mathbb{E}[\boldsymbol{m}_k] = \boldsymbol{m},$$

for any $i \in [n]$, $t \in [q]$, and $k \in [q]$.

The covariances between $\boldsymbol{v}_{\sigma(i)}$'s can be deduced from the proof by Mishchenko et al. (2020, Lemma 1) as follows:

$$\mathrm{Cov}(\boldsymbol{v}_{\sigma(i)}, \boldsymbol{v}_{\sigma(j)}) := \mathbb{E}\left[\langle \boldsymbol{v}_{\sigma(i)} - \boldsymbol{m}, \boldsymbol{v}_{\sigma(j)} - \boldsymbol{m}\rangle\right] = \begin{cases} -\frac{\tau^2}{n-1}, & \text{if } i \ne j, \\ \tau^2 & \text{if } i = j. \end{cases}$$

Thus, for each $t \in [q]$, the variance of $\boldsymbol{w}_t$ is obtained as

$$\mathbb{E}\left[\|\boldsymbol{w}_t - \boldsymbol{m}\|^2\right] = \mathbb{E}\left[\left\|\frac{1}{|\mathcal{W}_t|}\sum_{i \in \mathcal{W}_t}(\boldsymbol{v}_i - \boldsymbol{m})\right\|^2\right]$$

$$= \frac{1}{|\mathcal{W}_t|^2}\left\{\sum_{i \in \mathcal{W}_t}\mathbb{E}\left[\|\boldsymbol{v}_i - \boldsymbol{m}\|^2\right] + \sum_{\substack{i,j \in \mathcal{W}_t \\ i \ne j}}\mathrm{Cov}(\boldsymbol{v}_i, \boldsymbol{v}_j)\right\}$$

$$= \frac{1}{|\mathcal{W}_t|^2}\left\{|\mathcal{W}_t|\tau^2 + |\mathcal{W}_t|(|\mathcal{W}_t| - 1)\left(-\frac{\tau^2}{n-1}\right)\right\} = \frac{n - |\mathcal{W}_t|}{|\mathcal{W}_t|(n-1)}\tau^2,$$

which can also be directly deduced by Lemma 1 of Mishchenko et al. (2020). We notice that this does not depends on the size of the batch $\mathcal{W}_t$.

Next, we look at the covariances between distinct $\boldsymbol{w}_t$'s. For a pair of distinct integers $t, u \in [q]$, by the bi-linearity of covariance,

$$\mathrm{Cov}(\boldsymbol{w}_t, \boldsymbol{w}_u) = \frac{1}{|\mathcal{W}_t| \cdot |\mathcal{W}_u|}\sum_{(i,j) \in \mathcal{W}_t \times \mathcal{W}_u}\mathrm{Cov}(\boldsymbol{v}_i, \boldsymbol{v}_j)$$

$$= \frac{1}{|\mathcal{W}_t| \cdot |\mathcal{W}_u|}\sum_{(i,j) \in \mathcal{W}_t \times \mathcal{W}_u}\left(-\frac{\tau^2}{n-1}\right) = -\frac{\tau^2}{n-1}.$$

The second equality holds because $\mathcal{W}_t$ and $\mathcal{W}_u$ are a disjoint set of integers whenever $t \ne u$.

Now, fix any $k \in [q-1]$. Note that, by our mini-batching strategy, $|\mathcal{W}_t| = b$ for every $t < q$. Therefore, by definition of $\boldsymbol{m}_k$,

$$\mathbb{E}\left[\|\boldsymbol{m}_k - \boldsymbol{m}\|^2\right] = \mathbb{E}\left[\left\|\frac{1}{k}\sum_{t=1}^{k}(\boldsymbol{w}_t - \boldsymbol{m})\right\|^2\right]$$

$$= \frac{1}{k^2}\left\{\sum_{t=1}^{k}\mathbb{E}\left[\|\boldsymbol{w}_t - \boldsymbol{m}\|^2\right] + \sum_{\substack{t,u \in [k] \\ t \ne u}}\mathrm{Cov}(\boldsymbol{w}_t, \boldsymbol{w}_u)\right\}$$

$$= \frac{1}{k^2}\left\{k \cdot \left(\frac{n-b}{b(n-1)}\tau^2\right) + k(k-1) \cdot \left(-\frac{\tau^2}{n-1}\right)\right\} = \frac{n - bk}{bk(n-1)}\tau^2.$$

$\square$

### B.3 BASIC RECURRENCE INEQUALITY

In this subsection, we present a basic result of a recurrence inequality. It serves as a stepping-stone of our convergence bound, particularly at the end of the proof (Appendix C.5).

**Proposition 15.** *Let $\{a_k\}_{k=1}^{\infty}$ be a sequence of non-negative numbers satisfying the following recurrence inequality:*

$$a_{k+1} \leq (1 - b\eta)a_k + c\eta^{m+1},$$

*where $b, c$, and $\eta$ are non-negative real numbers such that $b\eta \in (0, 1)$, and $m$ is a non-negative integer. Then, for any integer $K \geq 1$, we have*

$$a_{K+1} \leq (1 - b\eta)^K a_1 + c\eta^m/b.$$

*Proof.* We proceed with induction on $K = 0, 1, 2, \cdots$. Note that

$$a_1 \leq (1 - b\eta)^0 a_1 + c\eta^m/b.$$

This shows the case when $K = 0$. On the other hand, if $K \geq 1$, by an inductive assumption,

$$
\begin{aligned}
a_{K+1} &\leq (1 - b\eta)a_K + c\eta^{m+1} \\
&\leq (1 - b\eta) \cdot \left( (1 - b\eta)^{K-1}a_1 + c\eta^m/b \right) + c\eta^{m+1} \\
&= (1 - b\eta)^K a_1 + c\eta^m/b.
\end{aligned}
$$

$\square$

## C PROOFS FOR (MINI-BATCH) SIMULTANEOUS SGDA-RR

In this appendix, we provide a convergence analysis for the mini-batch **sim**SGDA-RR (Algorithm 2) on both general nonconvex-PŁ problems and primal-PŁ-PŁ problems. The two cases mostly share the same proof strategies; they only diverge at the end of the proofs. The proof is long; we first provide the sketch of proof in subsection C.1; then, we provide the full proof by dividing it into 4 follow-up subsections of this appendix. The proof for the alternating counterpart (minibatch altSGDA-RR) can be done with some modifications illustrated in Appendix D. All technical propositions required for the proofs can be found in Appendix B.

### C.1 WARM-UP: PROOF SKETCH FOR $b = 1$

Here we simply consider the proofs of Theorem 1 and 2 for **sim**SGDA-RR, which is a fully stochastic case (mini-batches of size $b = 1$). The proofs for altSGDA-RR can be done with slight modifications.

We start the proof by aggregating all updates throughout an epoch to obtain an "epoch-wise" update:

$$
\begin{aligned}
\boldsymbol{x}_0^{k+1} &= \boldsymbol{x}_0^k - n\alpha \boldsymbol{g}^k, & \boldsymbol{g}^k &= \tfrac{1}{n}\sum_{i=1}^n \nabla_1 f_{\sigma_k(i)}(\boldsymbol{z}_{i-1}^k), \\
\boldsymbol{y}_0^{k+1} &= \boldsymbol{y}_0^k + n\beta \boldsymbol{h}^k, & \boldsymbol{h}^k &= \tfrac{1}{n}\sum_{i=1}^n \nabla_2 f_{\sigma_k(i)}(\boldsymbol{z}_{i-1}^k).
\end{aligned}
$$

The reason is that the sampled components in each epoch are dependent to each other so that it is much harder to deal with each iteration individually. The strategy of update-aggregation is quite general for analysis of optimization algorithms involving without-replacement sampling (Ahn et al., 2020; Mishchenko et al., 2020; Nguyen et al., 2021; Das et al., 2022). We assume that the intermediate iterates $\boldsymbol{z}_1^k, \ldots, \boldsymbol{z}_n^k$ stay close to the starting iterate $\boldsymbol{z}_0^k$ of an epoch $k$, which can be ensured by small step sizes. Then, we can approximate the aggregated epoch of SGDA-RR as a step of simGDA applied to $f = \frac{1}{n}\sum_{i=1}^n f_i$, with approximations of $\boldsymbol{g}^k \approx \nabla_1 f(\boldsymbol{z}_0^k)$ and $\boldsymbol{h}^k \approx \nabla_2 f(\boldsymbol{z}_0^k)$.

With Assumptions 1 and 4, note that the primal function $\Phi(\cdot)$ is $(L + L^2/\mu_2)$-smooth (Proposition 9). Applying this and $L$-smoothness of $-f$, we can have the following inequality (Lemma 16):

$$
\begin{aligned}
V_\lambda(\boldsymbol{z}_0^{k+1}) - V_\lambda(\boldsymbol{z}_0^k) \leq{} & -((\lambda + 1)/2)\, n\alpha \left\| \nabla\Phi(\boldsymbol{x}_0^k) \right\|^2 + (\lambda + 1)n\alpha \left\| \nabla\Phi(\boldsymbol{x}_0^k) - \nabla_1 f(\boldsymbol{z}_0^k) \right\|^2 \\
& + (n\alpha/2) \left\| \nabla_1 f(\boldsymbol{z}_0^k) \right\|^2 - (n\beta/2) \left\| \nabla_2 f(\boldsymbol{z}_0^k) \right\|^2 \\
& + (\lambda + 1/2)\, n\alpha \left\| \boldsymbol{g}^k - \nabla_1 f(\boldsymbol{z}_0^k) \right\|^2 + (n\beta/2) \left\| \boldsymbol{h}^k - \nabla_2 f(\boldsymbol{z}_0^k) \right\|^2.
\end{aligned}
$$

Hence, to guarantee the fast decrease of $V_\lambda(\boldsymbol{z}_0^k)$, it is important to control the "noise" terms for GDA approximations, $\left\|\boldsymbol{g}^k - \nabla_1 f(\boldsymbol{z}_0^k)\right\|^2$ and $\left\|\boldsymbol{h}^k - \nabla_2 f(\boldsymbol{z}_0^k)\right\|^2$, in the last line of inequality above. By applying the tools for without-replacement sampling (Proposition 14), we can actually upper-bound the conditional expectations of both noise terms by

$$2L^2 n(n+A)\left(\alpha^2 \left\|\nabla_1 f(\boldsymbol{z}_0^k)\right\|^2 + \beta^2 \left\|\nabla_2 f(\boldsymbol{z}_0^k)\right\|^2\right) + 2L^2 n(\alpha^2 + \beta^2)B. \quad \text{(Lemma 17 \& 18)}$$

Then, by taking advantage of several properties of smooth nonconvex-PŁ functions (*e.g.*, Propositions 7, 8, and 9) and some small-step-size assumptions (*e.g.*, $\beta = \mathcal{O}(1/nL)$, $\beta/\alpha = r \gtrsim \kappa_2^2$), we eventually have

$$\mathbb{E}\left[V_\lambda(\boldsymbol{z}_0^{k+1})\right] - \mathbb{E}\left[V_\lambda(\boldsymbol{z}_0^k)\right] \le -n\alpha\mathbb{E}\left[\left\|\nabla\Phi(\boldsymbol{x}_0^k)\right\|^2\right] - (L\kappa_2 n\alpha/2)\mathbb{E}\left[\Phi(\boldsymbol{x}_0^k) - f(\boldsymbol{z}_0^k)\right] + C\alpha^3,$$

where $C \ge 0$ is a constant (with respect to $k$) depending on $L$, $n$, $B$, and $r = \beta/\alpha$. (Lemma 20). We note that the step size ratio $r \gtrsim \kappa_2^2$ is crucial for showing that the coefficient in front of the term $\mathbb{E}\left[\Phi(\boldsymbol{x}_0^k) - f(\boldsymbol{z}_0^k)\right]$ is non-positive: even if it is possible with $r \lesssim \kappa_2^2$, we must assume that $\kappa_2$ upper-bounded by a positive numerical constant, which is not desirable for showing convergence bounds. Thus, we expect that a different proof strategy should be applied to avoid the requirement $r \gtrsim \kappa_2^2$ on the step size ratio.

The proofs of Theorems 1 and 2 diverge from here. The rest of the proof is mostly about choosing appropriate step sizes and solving the recurrence inequalities.

The full proof of Theorems 4 and 5 starts from the following subsection.

## C.2 EPOCH-WISE REPRESENTATIONS AND BOUNDING NOISE TERMS

Before starting the proof, we again remark that we assume that the mini-batch size $b$ divides the number of components $n$ (namely, $q := n/b$ is a positive integer) for simplicity: thus, readers who want to read proofs for fully stochastic case (*i.e.*, $b = 1$) can substitute $n$ to every $q$. Also, there is no problem in treating any fraction with a positive numerator and a zero denominator as $+\infty$. Moreover, we simply regard $(q-1)/(n-1) = 1$ when $n = 1$.

We start the proof by aggregating all updates throughout an epoch to obtain an "epoch-wise" update equation. The reason is that the sampled components in each epoch depend on each other, so it is much harder to deal with each iteration individually. At iteration $t \in [n/b] = [q]$ of epoch $k \in [K]$, we use a mini-batch

$$\mathcal{B}_t^k := \{\sigma_k(j) : b(t-1) < j \le bt, j \in [n]\}.$$

To ease the analysis of Algorithm 2, define the following sums associated with (partial) gradient oracles at a point $\boldsymbol{z} = (\boldsymbol{x}; \boldsymbol{y})$ over the mini-batch:

$$\boldsymbol{g}_t^k(\boldsymbol{z}) := \frac{1}{b}\sum_{i\in\mathcal{B}_t^k}\nabla_1 f_i(\boldsymbol{z}), \quad \boldsymbol{h}_t^k(\boldsymbol{z}) := \frac{1}{b}\sum_{i\in\mathcal{B}_t^k}\nabla_2 f_i(\boldsymbol{z}).$$

By Assumption 1, $\boldsymbol{g}_t^k$ and $\boldsymbol{h}_t^k$ are $L$-Lipschitz continuous. Computing the average of them over a whole epoch $(\boldsymbol{z}_0^k, \cdots, \boldsymbol{z}_{q-1}^k)$, we define

$$\boldsymbol{g}^k := \frac{1}{q}\sum_{t=1}^q \boldsymbol{g}_t^k(\boldsymbol{z}_{t-1}^k), \quad \boldsymbol{h}^k := \frac{1}{q}\sum_{t=1}^q \boldsymbol{h}_t^k(\boldsymbol{z}_{t-1}^k).$$

Then, by summing up the updates in the epoch $k$, we can summarize the epoch as follows.

$$\boldsymbol{x}_0^{k+1} = \boldsymbol{x}_0^k - q\alpha\boldsymbol{g}^k, \quad \boldsymbol{y}_0^{k+1} = \boldsymbol{y}_0^k + q\beta\boldsymbol{h}^k. \quad \text{(simSGDA-RR)}$$

We may assume that the intermediate iterates $\boldsymbol{z}_1^k, \ldots, \boldsymbol{z}_q^k$ stay close to the starting iterate $\boldsymbol{z}_0^k$ of an epoch $k$, which results from, *e.g.*, small step sizes. Then, we can approximate the aggregated epoch of SGDA-RR as a step of simGDA applied to $f = \frac{1}{n}\sum_{i=1}^n f_i$: $\boldsymbol{g}^k \approx \nabla_1 f(\boldsymbol{z}_0^k)$, $\boldsymbol{h}^k \approx \nabla_2 f(\boldsymbol{z}_0^k)$. In other words,

$$\boldsymbol{x}_0^{k+1} \approx \boldsymbol{x}_0^k - q\alpha\nabla_1 f(\boldsymbol{z}_0^k), \quad \boldsymbol{y}_0^{k+1} \approx \boldsymbol{y}_0^k + q\beta\nabla_2 f(\boldsymbol{z}_0^k), \quad (\approx\text{simGDA})$$

With Assumptions 1, 3 and 4, we can yield a naive (but complicated) upper bound of the gap $V_\lambda(\boldsymbol{z}_0^{k+1}) - V_\lambda(\boldsymbol{z}_0^k)$, only applying the smoothness of $\Phi$ and $-f$, without any assumptions on step sizes.

**Lemma 16.** *Suppose that Assumptions 1, 3 and 4 hold. Let $\kappa_2 = L/\mu_2$, where $\mu_2$ is PŁ constant of $-f(\boldsymbol{x}; \cdot)$. Then, the mini-batch simSGDA-RR satisfies that*

$$
V_\lambda(\boldsymbol{z}_0^{k+1}) - V_\lambda(\boldsymbol{z}_0^k)
$$
$$
\leq -\left(\frac{\lambda + 1}{2}\right) q\alpha \left\|\nabla \Phi(\boldsymbol{x}_0^k)\right\|^2 + (\lambda + 1)q\alpha \left\|\nabla \Phi(\boldsymbol{x}_0^k) - \nabla_1 f(\boldsymbol{z}_0^k)\right\|^2
$$
$$
+ \frac{q\alpha}{2} \left\|\nabla_1 f(\boldsymbol{z}_0^k)\right\|^2 - \frac{q\beta}{2} \left\|\nabla_2 f(\boldsymbol{z}_0^k)\right\|^2
$$
$$
+ \left(\lambda + \frac{1}{2}\right) q\alpha \left\|\boldsymbol{g}^k - \nabla_1 f(\boldsymbol{z}_0^k)\right\|^2 + \frac{q\beta}{2} \left\|\boldsymbol{h}^k - \nabla_2 f(\boldsymbol{z}_0^k)\right\|^2
$$
$$
- \left[\lambda - \{(\lambda + 1)(\kappa_2 + 1) + 1\} Lq\alpha\right] \frac{q\alpha}{2} \left\|\boldsymbol{g}^k\right\|^2 - (1 - Lq\beta)\frac{q\beta}{2} \left\|\boldsymbol{h}^k\right\|^2. \tag{7}
$$

*Proof.* By definition of $V_\lambda$, the following equation holds:

$$
V_\lambda(\boldsymbol{z}_0^{k+1}) - V_\lambda(\boldsymbol{z}_0^k) = (\lambda + 1)\left[\Phi(\boldsymbol{x}_0^{k+1}) - \Phi(\boldsymbol{x}_0^k)\right] + \left[f(\boldsymbol{z}_0^k) - f(\boldsymbol{z}_0^{k+1})\right]. \tag{8}
$$

First, we seek for an upper bound of $\Phi(\boldsymbol{x}_0^{k+1}) - \Phi(\boldsymbol{x}_0^k)$. By Proposition 9, $\Phi$ is $L(\kappa_2 + 1)$-smooth. Hence, we have

$$
\Phi(\boldsymbol{x}_0^{k+1}) - \Phi(\boldsymbol{x}_0^k)
$$
$$
\leq \left\langle \nabla \Phi(\boldsymbol{x}_0^k), \boldsymbol{x}_0^{k+1} - \boldsymbol{x}_0^k \right\rangle + \frac{L(\kappa_2 + 1)}{2} \left\|\boldsymbol{x}_0^{k+1} - \boldsymbol{x}_0^k\right\|^2
$$
$$
= -q\alpha \left\langle \nabla \Phi(\boldsymbol{x}_0^k), \boldsymbol{g}^k \right\rangle + \frac{L(\kappa_2 + 1)}{2} q^2 \alpha^2 \left\|\boldsymbol{g}^k\right\|^2
$$
$$
= -\frac{q\alpha}{2} \left\{\left\|\nabla \Phi(\boldsymbol{x}_0^k)\right\|^2 + \left\|\boldsymbol{g}^k\right\|^2 - \left\|\nabla \Phi(\boldsymbol{x}_0^k) - \boldsymbol{g}^k\right\|^2\right\} + \frac{L(\kappa_2 + 1)}{2} q^2 \alpha^2 \left\|\boldsymbol{g}^k\right\|^2
$$
$$
= -\frac{q\alpha}{2} \left\|\nabla \Phi(\boldsymbol{x}_0^k)\right\|^2 + \frac{q\alpha}{2} \left\|\nabla \Phi(\boldsymbol{x}_0^k) - \boldsymbol{g}^k\right\|^2 - \frac{q\alpha}{2}(1 - L(\kappa_2 + 1)q\alpha) \left\|\boldsymbol{g}^k\right\|^2
$$
$$
\leq -\frac{q\alpha}{2} \left\|\nabla \Phi(\boldsymbol{x}_0^k)\right\|^2 + q\alpha \left\|\nabla \Phi(\boldsymbol{x}_0^k) - \nabla_1 f(\boldsymbol{z}_0^k)\right\|^2 + q\alpha \left\|\boldsymbol{g}^k - \nabla_1 f(\boldsymbol{z}_0^k)\right\|^2
$$
$$
- \frac{q\alpha}{2}(1 - L(\kappa_2 + 1)q\alpha) \left\|\boldsymbol{g}^k\right\|^2. \tag{9}
$$

The third line is due to polarization equality[9] and the last inequality applies Young's inequality.[10]

Next, applying Assumption 1, $L$-smoothness of $-f(\cdot; \cdot)$ yields an upper bound of $f(\boldsymbol{z}_0^k) - f(\boldsymbol{z}_0^{k+1})$.

$$
f(\boldsymbol{z}_0^k) - f(\boldsymbol{z}_0^{k+1})
$$
$$
\leq -\left\langle \nabla f(\boldsymbol{z}_0^k), \boldsymbol{z}_0^{k+1} - \boldsymbol{z}_0^k \right\rangle + \frac{L}{2} \left\|\boldsymbol{z}_0^{k+1} - \boldsymbol{z}_0^k\right\|^2
$$
$$
= -\left\langle \nabla_1 f(\boldsymbol{z}_0^k), \boldsymbol{x}_0^{k+1} - \boldsymbol{x}_0^k \right\rangle - \left\langle \nabla_2 f(\boldsymbol{z}_0^k), \boldsymbol{y}_0^{k+1} - \boldsymbol{y}_0^k \right\rangle + \frac{L}{2} \left\|\boldsymbol{x}_0^{k+1} - \boldsymbol{x}_0^k\right\|^2 + \frac{L}{2} \left\|\boldsymbol{y}_0^{k+1} - \boldsymbol{y}_0^k\right\|^2
$$
$$
= q\alpha \left\langle \nabla_1 f(\boldsymbol{z}_0^k), \boldsymbol{g}^k \right\rangle - q\beta \left\langle \nabla_2 f(\boldsymbol{z}_0^k), \boldsymbol{h}^k \right\rangle + \frac{L}{2} q^2 \alpha^2 \left\|\boldsymbol{g}^k\right\|^2 + \frac{L}{2} q^2 \beta^2 \left\|\boldsymbol{h}^k\right\|^2
$$
$$
= \frac{q\alpha}{2} \left\|\nabla_1 f(\boldsymbol{z}_0^k)\right\|^2 - \frac{q\alpha}{2} \left\|\boldsymbol{g}^k - \nabla_1 f(\boldsymbol{z}_0^k)\right\|^2 + \frac{q\alpha}{2}(1 + Lq\alpha) \left\|\boldsymbol{g}^k\right\|^2
$$
$$
- \frac{q\beta}{2} \left\|\nabla_2 f(\boldsymbol{z}_0^k)\right\|^2 + \frac{q\beta}{2} \left\|\boldsymbol{h}^k - \nabla_2 f(\boldsymbol{z}_0^k)\right\|^2 - \frac{q\beta}{2}(1 - Lq\beta) \left\|\boldsymbol{h}^k\right\|^2. \tag{10}
$$

The last equality is due to polarization equality. Lastly, substituting (9) and (10) to (8) finishes the proof. □

We remark that the last two terms of the inequality (7) can be simply ignored by applying small enough step sizes. However, the terms in the third line of (7) are non-negatives terms related to the

---

[9]For any $\boldsymbol{a}, \boldsymbol{b} \in \mathbb{R}^d$, $2\langle \boldsymbol{a}, \boldsymbol{b} \rangle = \|\boldsymbol{a}\|^2 + \|\boldsymbol{b}\|^2 - \|\boldsymbol{a} - \boldsymbol{b}\|^2$.

[10]For any $\boldsymbol{a}, \boldsymbol{b} \in \mathbb{R}^d$, $\|\boldsymbol{a} + \boldsymbol{b}\|^2 \leq 2\|\boldsymbol{a}\|^2 + 2\|\boldsymbol{b}\|^2$.

"noise" of approximation $\boldsymbol{g}^k \approx \nabla_1 f(\boldsymbol{z}_0^k)$, $\boldsymbol{h}^k \approx \nabla_2 f(\boldsymbol{z}_0^k)$. Hence, it is important to control the noise terms $\left\| \boldsymbol{g}^k - \nabla_1 f(\boldsymbol{z}_0^k) \right\|^2$ and $\left\| \boldsymbol{h}^k - \nabla_2 f(\boldsymbol{z}_0^k) \right\|^2$ to guarantee a fast decrease of $V_\lambda(\boldsymbol{z}_0^k)$.

**Lemma 17.** *For mini-batch simSGDA-RR, define*

$$G_k := \frac{1}{q} \sum_{t=1}^{q} \left\| \boldsymbol{z}_{t-1}^k - \boldsymbol{z}_0^k \right\|^2. \tag{11}$$

*With Assumption 1, then*

$$\left\| \boldsymbol{g}^k - \nabla_1 f(\boldsymbol{z}_0^k) \right\|^2 \le L^2 G_k \quad and \quad \left\| \boldsymbol{h}^k - \nabla_2 f(\boldsymbol{z}_0^k) \right\|^2 \le L^2 G_k.$$

As a side remark, $G_k = 0$ when $q = 1$ and, in particular, $n = 1$.

*Proof.* Recall that $\frac{1}{q} \sum_{t=1}^{q} \boldsymbol{g}_t^k(\boldsymbol{z}) = \nabla_1 f(\boldsymbol{z})$ and $\frac{1}{q} \sum_{t=1}^{q} \boldsymbol{h}_t^k(\boldsymbol{z}) = \nabla_2 f(\boldsymbol{z})$. By Lipschitz continuity and Jensen's inequality,[11]

$$
\begin{aligned}
\left\| \boldsymbol{g}^k - \nabla_1 f(\boldsymbol{z}_0^k) \right\|^2 &= \left\| \frac{1}{q} \sum_{t=1}^{q} \left[ \boldsymbol{g}_t^k(\boldsymbol{z}_{t-1}^k) - \boldsymbol{g}_t^k(\boldsymbol{z}_0^k) \right] \right\|^2 \\
&\le \frac{1}{q} \sum_{t=1}^{q} \left\| \boldsymbol{g}_t^k(\boldsymbol{z}_{t-1}^k) - \boldsymbol{g}_t^k(\boldsymbol{z}_0^k) \right\|^2 \le \frac{L^2}{q} \sum_{t=1}^{q} \left\| \boldsymbol{z}_{t-1}^k - \boldsymbol{z}_0^k \right\|^2.
\end{aligned}
$$

Similarly,

$$\left\| \boldsymbol{h}^k - \nabla_2 f(\boldsymbol{z}_0^k) \right\|^2 \le \frac{L^2}{q} \sum_{t=1}^{q} \left\| \boldsymbol{z}_{t-1}^k - \boldsymbol{z}_0^k \right\|^2.$$

This concludes the proof. $\qquad\square$

Thanks to the lemma, it suffices to bound the term $G_k$. One can notice that it also represents how far the intermediate iterates $\boldsymbol{z}_t^k$ are from the pivot $\boldsymbol{z}_0^k$ in average. Before moving on, we define an algorithm-specific symbol denoting a conditional expectation.

**Definition 3.** *We denote a conditional expectation of a random variable $X$ given all iterates of the first $k-1$ epochs by $\mathbb{E}_k[X] = \mathbb{E}[X | \boldsymbol{z}_0^1, \boldsymbol{z}_1^1, \dots, \boldsymbol{z}_n^{k-1}]$. In particular, if $k = 1$, it boils down to a conditional expectation given only the initial iterate $\boldsymbol{z}_0^1$.*

We get an upper bound of a (conditional) expectation $\mathbb{E}_k[G_k]$ in the following lemma, which extends a lemma of Nguyen et al. (2021, Lemma 6) to our minimax problems.

**Lemma 18.** *Suppose that Assumptions 1 and 2 hold. Assume that the permutation $\sigma_k$ is sampled uniformly at random from $\mathbb{S}_n$. Then, for any step sizes $\alpha, \beta$ satisfying $\alpha^2 + \beta^2 \le \frac{1}{3q(q-1)L^2}$, the iterates $\{\boldsymbol{z}_t^k\}_{t=0}^{q-1}$ of the $k$-th epoch of mini-batch simSGDA-RR satisfies (for $n > 1$)*

$$\mathbb{E}_k G_k \le 2 \left( q^2 + \frac{q(q-1)}{n-1} A \right) \left( \alpha^2 \left\| \nabla_1 f(\boldsymbol{z}_0^k) \right\|^2 + \beta^2 \left\| \nabla_2 f(\boldsymbol{z}_0^k) \right\|^2 \right) + \frac{2q(q-1)}{n-1} (\alpha^2 + \beta^2) B.$$

*Proof.* Note that $G_k = 0$ when $q = 1$ by its definition. From now, we may assume $q > 1$ and $n > 1$ in this proof. By summing the first $t \in [q-1]$ updates of the $k$-th epoch of mini-batch simSGDA-RR, we have

$$\boldsymbol{x}_t^k = \boldsymbol{x}_0^k - t\alpha \left( \frac{1}{t} \sum_{j=1}^{t} \boldsymbol{g}_j^k(\boldsymbol{z}_{j-1}^k) \right), \quad \boldsymbol{y}_t^k = \boldsymbol{y}_0^k + t\beta \left( \frac{1}{t} \sum_{j=1}^{t} \boldsymbol{h}_j^k(\boldsymbol{z}_{j-1}^k) \right).$$

---

[11] For any $n$ vectors $a_1, \cdots, a_n$, $\left\| \frac{1}{n} \sum_{j=1}^{n} a_j \right\|^2 \le \frac{1}{n} \sum_{j=1}^{n} \|a_j\|^2$.

Then we can bound the following squared distance.

$$\left\|\boldsymbol{x}_t^k - \boldsymbol{x}_0^k\right\|^2 = \alpha^2 t^2 \left\|\frac{1}{t}\sum_{j=1}^t \boldsymbol{g}_j^k(\boldsymbol{z}_{j-1}^k)\right\|^2$$

$$\leq 3\alpha^2 t^2 \left[\left\|\frac{1}{t}\sum_{j=1}^t \left[\boldsymbol{g}_j^k(\boldsymbol{z}_{j-1}^k) - \boldsymbol{g}_j^k(\boldsymbol{z}_0^k)\right]\right\|^2 + \left\|\frac{1}{t}\sum_{j=1}^t \boldsymbol{g}_j^k(\boldsymbol{z}_0^k) - \nabla_1 f(\boldsymbol{z}_0^k)\right\|^2 + \left\|\nabla_1 f(\boldsymbol{z}_0^k)\right\|^2\right]$$

$$\leq 3\alpha^2 t \sum_{j=1}^t \left\|\boldsymbol{g}_j^k(\boldsymbol{z}_{j-1}^k) - \boldsymbol{g}_j^k(\boldsymbol{z}_0^k)\right\|^2 + 3\alpha^2 t^2 \left[\left\|\frac{1}{t}\sum_{j=1}^t \boldsymbol{g}_j^k(\boldsymbol{z}_0^k) - \nabla_1 f(\boldsymbol{z}_0^k)\right\|^2 + \left\|\nabla_1 f(\boldsymbol{z}_0^k)\right\|^2\right]$$

$$\leq 3\alpha^2 L^2 t \cdot \sum_{j=1}^t \left\|\boldsymbol{z}_{j-1}^k - \boldsymbol{z}_0^k\right\|^2 + 3\alpha^2 t^2 \left[\left\|\frac{1}{t}\sum_{j=1}^t \boldsymbol{g}_j^k(\boldsymbol{z}_0^k) - \nabla_1 f(\boldsymbol{z}_0^k)\right\|^2 + \left\|\nabla_1 f(\boldsymbol{z}_0^k)\right\|^2\right]$$

$$\leq 3\alpha^2 L^2 t \cdot qG_k + 3\alpha^2 t^2 \left[\left\|\frac{1}{t}\sum_{j=1}^t \boldsymbol{g}_j^k(\boldsymbol{z}_0^k) - \nabla_1 f(\boldsymbol{z}_0^k)\right\|^2 + \left\|\nabla_1 f(\boldsymbol{z}_0^k)\right\|^2\right]. \tag{12}$$

The second and third lines are due to Jensen's inequality. The fourth line is due to $L$-Lipschitz continuity of $\boldsymbol{g}_j^k$. Likewise,

$$\left\|\boldsymbol{y}_t^k - \boldsymbol{y}_0^k\right\|^2 \leq 3\beta^2 L^2 t \cdot qG_k + 3\beta^2 t^2 \left[\left\|\frac{1}{t}\sum_{j=1}^t \boldsymbol{h}_j^k(\boldsymbol{z}_0^k) - \nabla_2 f(\boldsymbol{z}_0^k)\right\|^2 + \left\|\nabla_2 f(\boldsymbol{z}_0^k)\right\|^2\right]. \tag{13}$$

Summing up (12) and (13),

$$\left\|\boldsymbol{z}_t^k - \boldsymbol{z}_0^k\right\|^2 = \left\|\boldsymbol{x}_t^k - \boldsymbol{x}_0^k\right\|^2 + \left\|\boldsymbol{y}_t^k - \boldsymbol{y}_0^k\right\|^2$$

$$\leq 3(\alpha^2 + \beta^2)L^2 tqG_k + 3\alpha^2 t^2 \left[\left\|\frac{1}{t}\sum_{j=1}^t \boldsymbol{g}_j^k(\boldsymbol{z}_0^k) - \nabla_1 f(\boldsymbol{z}_0^k)\right\|^2 + \left\|\nabla_1 f(\boldsymbol{z}_0^k)\right\|^2\right]$$

$$+ 3\beta^2 t^2 \left[\left\|\frac{1}{t}\sum_{j=1}^t \boldsymbol{h}_j^k(\boldsymbol{z}_0^k) - \nabla_2 f(\boldsymbol{z}_0^k)\right\|^2 + \left\|\nabla_2 f(\boldsymbol{z}_0^k)\right\|^2\right]. \tag{14}$$

Taking (conditional) expectation $\mathbb{E}_k$ (given $\boldsymbol{z}_0^k$) to inequality (14),

$$\mathbb{E}_k \left\|\boldsymbol{z}_t^k - \boldsymbol{z}_0^k\right\|^2$$

$$\overset{(14)}{\leq} 3(\alpha^2 + \beta^2)L^2 tq \cdot (\mathbb{E}_k[G_k]) + 3\alpha^2 t^2 \left\|\nabla_1 f(\boldsymbol{z}_0^k)\right\|^2 + 3\beta^2 t^2 \left\|\nabla_2 f(\boldsymbol{z}_0^k)\right\|^2$$

$$+ 3\alpha^2 t^2 \mathbb{E}_k \left\|\frac{1}{t}\sum_{j=1}^t \boldsymbol{g}_j^k(\boldsymbol{z}_0^k) - \nabla_1 f(\boldsymbol{z}_0^k)\right\|^2 + 3\beta^2 t^2 \mathbb{E}_k \left\|\frac{1}{t}\sum_{j=1}^t \boldsymbol{h}_j^k(\boldsymbol{z}_0^k) - \nabla_2 f(\boldsymbol{z}_0^k)\right\|^2. \tag{15}$$

Here we take advantage of the without-replacement sampling. Putting $\nabla_s f_i(\boldsymbol{z}_0^k) \mapsto \boldsymbol{v}_i$ ($s \in \{1, 2\}$), one can realize a correspondence between the quantities that arise from our algorithm and the symbols in Appendix B.2: for $s = 1$ ($\nabla_1 f_i(\boldsymbol{z}_0^k) \mapsto \boldsymbol{v}_i$),

$$\boldsymbol{m} = \nabla_1 f(\boldsymbol{z}_0^k), \quad \tau^2 \leq A\left\|\nabla_1 f(\boldsymbol{z}_0^k)\right\|^2 + B, \quad \boldsymbol{w}_t = \boldsymbol{g}_t^k(\boldsymbol{z}_0^k), \quad \boldsymbol{m}_t = \frac{1}{t}\sum_{j=1}^t \boldsymbol{g}_j^k(\boldsymbol{z}_0^k),$$

and for $s = 2$ $(\nabla_2 f_i(\boldsymbol{z}_0^k) \mapsto \boldsymbol{v}_i)$,

$$\boldsymbol{m} = \nabla_2 f(\boldsymbol{z}_0^k), \quad \tau^2 \leq A \left\| \nabla_2 f(\boldsymbol{z}_0^k) \right\|^2 + B, \quad \boldsymbol{w}_t = \boldsymbol{h}_t^k(\boldsymbol{z}_0^k), \quad \boldsymbol{m}_t = \frac{1}{t} \sum_{j=1}^t \boldsymbol{h}_j^k(\boldsymbol{z}_0^k).$$

The upper bounds of $\tau^2$'s come from Assumption 2. Then by Proposition 14, for any $t \leq q$,

$$t^2 \mathbb{E}_k \left\| \frac{1}{t} \sum_{j=1}^t \boldsymbol{g}_j^k(\boldsymbol{z}_0^k) - \nabla_1 f(\boldsymbol{z}_0^k) \right\|^2 \leq \frac{t(q-t)}{n-1} \left( A \left\| \nabla_1 f(\boldsymbol{z}_0^k) \right\|^2 + B \right),$$

$$t^2 \mathbb{E}_k \left\| \frac{1}{t} \sum_{j=1}^t \boldsymbol{h}_j^k(\boldsymbol{z}_0^k) - \nabla_2 f(\boldsymbol{z}_0^k) \right\|^2 \leq \frac{t(q-t)}{n-1} \left( A \left\| \nabla_2 f(\boldsymbol{z}_0^k) \right\|^2 + B \right).$$

Putting these to the inequality (15),

$$\mathbb{E}_k \left\| \boldsymbol{z}_t^k - \boldsymbol{z}_0^k \right\|^2 \leq 3(\alpha^2 + \beta^2) \left[ L^2 t q \mathbb{E}_k[G_k] + \frac{t(q-t)}{n-1} B \right]$$
$$+ 3 \left( \alpha^2 \left\| \nabla_1 f(\boldsymbol{z}_0^k) \right\|^2 + \beta^2 \left\| \nabla_2 f(\boldsymbol{z}_0^k) \right\|^2 \right) \left[ t^2 + \frac{t(q-t)}{n-1} A \right].$$

Taking an average of the inequality above over $0 \leq t \leq q-1$,

$$\mathbb{E}_k G_k = \frac{1}{q} \sum_{t=0}^{q-1} \mathbb{E}_k \left\| \boldsymbol{z}_t^k - \boldsymbol{z}_0^k \right\|^2$$
$$\leq \frac{3q(q-1)}{2} (\alpha^2 + \beta^2) L^2 \mathbb{E}_k G_k + (\alpha^2 + \beta^2) \frac{q^2-1}{2(n-1)} B$$
$$+ \left( \alpha^2 \left\| \nabla_1 f(\boldsymbol{z}_0^k) \right\|^2 + \beta^2 \left\| \nabla_2 f(\boldsymbol{z}_0^k) \right\|^2 \right) \left( \frac{(q-1)(2q-1)}{2} + \frac{q^2-1}{2(n-1)} A \right), \quad (16)$$

where we used the facts

$$\sum_{t=0}^{q-1} t = \frac{q(q-1)}{2}, \quad \frac{1}{q} \sum_{t=0}^{q-1} t^2 = \frac{(q-1)(2q-1)}{6}, \quad \text{and} \quad \frac{1}{q} \sum_{t=0}^{q-1} \frac{t(q-t)}{n-1} = \frac{q^2-1}{6(n-1)}.$$

Since we assumed $\alpha^2 + \beta^2 \leq \frac{1}{3q(q-1)L^2}$, we have $1 \leq 2 \left( 1 - \frac{3q(q-1)L^2}{2} (\alpha^2 + \beta^2) \right)$. Using this,

$$\mathbb{E}_k G_k \leq 2 \left( 1 - \frac{3q(q-1)L^2}{2} (\alpha^2 + \beta^2) \right) \mathbb{E}_k G_k$$
$$\stackrel{(16)}{\leq} \left( (q-1)(2q-1) + \frac{q^2-1}{(n-1)} A \right) \left( \alpha^2 \left\| \nabla_1 f(\boldsymbol{z}_0^k) \right\|^2 + \beta^2 \left\| \nabla_2 f(\boldsymbol{z}_0^k) \right\|^2 \right) + \frac{q^2-1}{n-1} (\alpha^2 + \beta^2) B$$
$$\leq 2 \left( q^2 + \frac{q(q-1)}{n-1} A \right) \left( \alpha^2 \left\| \nabla_1 f(\boldsymbol{z}_0^k) \right\|^2 + \beta^2 \left\| \nabla_2 f(\boldsymbol{z}_0^k) \right\|^2 \right) + \frac{2q(q-1)}{n-1} (\alpha^2 + \beta^2) B,$$

where the last inequality used $(q-1)(2q-1) \leq 2q^2$ and $q+1 \leq 2q$ for $q \geq 1$. $\qquad\square$

### C.3 RECURRENCE INEQUALITIES FOR GENERAL SMOOTH NONCONVEX-PŁ OBJECTIVE

Subsequently, we obtain recurrence inequalities about (expected) potential function $\mathbb{E}_k[V_\lambda(\boldsymbol{z}_0^k)]$ for nonconvex-PŁ problem. Since primal-PŁ-PŁ problem is a subclass of nonconvex-PŁ problem, the recurrence relations can serve as stepping-stones of our convergence rates.

We introduce some assumptions on *small* step sizes which enable us to get rid of a few troublesome terms from our bound. On top of that, combining the PŁ condition (Assumption 4) with Lemmas 16, 17, and 18, we eventually obtain a much more concise bound on the expected per-epoch change of $V_\lambda$. This simple recurrence inequality becomes the key to proving our convergence bounds.

**Lemma 19.** *Suppose that Assumptions 1, 2, 3, and 4 hold. Assume that the step sizes $\alpha$ and $\beta$ satisfy*

$$\alpha \leq \frac{\lambda}{\{(\lambda + 1)(\kappa_2 + 1) + 1\}qL}, \quad \beta \leq \frac{1}{qL}, \quad \alpha^2 + \beta^2 \leq \frac{1}{3q(q-1)L^2}, \tag{17}$$

*and the condition*

$$C_0 := q\beta - 2L^2 q \left( q^2 + \frac{q(q-1)}{n-1} A \right) ((2\lambda + 1)\alpha + \beta) \beta^2 \geq 0$$

*as well. Then, the iterates of mini-batch simSGDA-RR satisfy*

$$\mathbb{E}_k[V_\lambda(z_0^{k+1})] - V_\lambda(z_0^k) \leq -C_1 \left\| \nabla \Phi(x_0^k) \right\|^2 - C_2 \left[ \Phi(x_0^k) - f(z_0^k) \right] + C_3$$

*where*

$$C_1 = \left( \frac{\lambda - 1}{2} \right) q\alpha - 2L^2 q \left( q^2 + \frac{q(q-1)}{n-1} A \right) ((2\lambda + 1)\alpha + \beta)\alpha^2,$$

$$C_2 = \mu_2 C_0 - 2(\lambda + 2)L\kappa_2 q\alpha - 4L^3 \kappa_2 q \left( q^2 + \frac{q(q-1)}{n-1} A \right) ((2\lambda + 1)\alpha + \beta)\alpha^2$$

$$= \mu_2 q\beta - 2(\lambda + 2)L\kappa_2 q\alpha - 2L^2 \mu_2 q \left( q^2 + \frac{q(q-1)}{n-1} A \right) ((2\lambda + 1)\alpha + \beta) \left( 2\kappa_2^2 \alpha^2 + \beta^2 \right),$$

$$C_3 = \left( \frac{L^2 q^2 (q-1)}{n-1} \right) ((2\lambda + 1)\alpha + \beta) (\alpha^2 + \beta^2)B.$$

*Proof.* The first two inequalities of (17) eliminate the last two terms on the right-hand side of the inequality in Lemma 16. In addition, applying Lemma 17 to Lemma 16 as well, we have

$$V_\lambda(z_0^{k+1}) - V_\lambda(z_0^k) \leq - \left( \frac{\lambda + 1}{2} \right) q\alpha \left\| \nabla \Phi(x_0^k) \right\|^2 + (\lambda + 1)q\alpha \left\| \nabla \Phi(x_0^k) - \nabla_1 f(z_0^k) \right\|^2$$

$$+ \frac{q\alpha}{2} \left\| \nabla_1 f(z_0^k) \right\|^2 - \frac{q\beta}{2} \left\| \nabla_2 f(z_0^k) \right\|^2 + \frac{(2\lambda + 1)\alpha + \beta}{2} qL^2 G_k. \tag{18}$$

If we take the conditional expectation $\mathbb{E}_k$ and apply Lemma 18 (which requires the third inequality of (17) to hold) to (18)

$$\mathbb{E}_k[V_\lambda(z_0^{k+1})] - V_\lambda(z_0^k)$$

$$\leq - \left( \frac{\lambda + 1}{2} \right) q\alpha \left\| \nabla \Phi(x_0^k) \right\|^2 + (\lambda + 1)q\alpha \left\| \nabla \Phi(x_0^k) - \nabla_1 f(z_0^k) \right\|^2$$

$$+ \frac{1}{2} \left[ q\alpha + 2L^2 q \left( q^2 + \frac{q(q-1)}{n-1} A \right) ((2\lambda + 1)\alpha + \beta) \alpha^2 \right] \left\| \nabla_1 f(z_0^k) \right\|^2$$

$$- \frac{1}{2} \underbrace{\left[ q\beta - 2L^2 q \left( q^2 + \frac{q(q-1)}{n-1} A \right) ((2\lambda + 1)\alpha + \beta) \beta^2 \right]}_{C_0} \left\| \nabla_2 f(z_0^k) \right\|^2$$

$$+ \underbrace{\left( \frac{L^2 q^2 (q-1)}{n-1} \right) ((2\lambda + 1)\alpha + \beta) (\alpha^2 + \beta^2)B}_{C_3}. \tag{19}$$

It is now left to bound terms in (19) using the tools developed so far. First, recall that $\Phi(x) := \max_{y' \in \mathcal{Y}} f(x; y')$. Since $-f(x; y)$ is $\mu_2$-PŁ in $y$, we have

$$- \left\| \nabla_2 f(z_0^k) \right\|^2 \leq -2\mu_2 (\Phi(x_0^k) - f(z_0^k)). \tag{20}$$

Given any $x$, $\nabla \Phi(x) = \nabla_1 f(x; y^*(x))$ for any $y^*(x) \in \arg\max_{y' \in \mathcal{Y}} f(x; y')$ by Proposition 9. Besides, $-f(x; \cdot)$ satisfies QG condition with constant $\mu_2$ by Proposition 7. Thus, by choosing $y^*(x_0^k)$ to be the projection of $y_0^k$ onto $\arg\max_{y' \in \mathcal{Y}} f(x_0^k; y')$,

$$\left\| \nabla \Phi(x_0^k) - \nabla_1 f(z_0^k) \right\|^2 \leq L^2 \left\| y^*(x_0^k) - y_0^k \right\|^2 \leq 2L\kappa_2 \left[ \Phi(x_0^k) - f(z_0^k) \right]. \tag{21}$$

Here, the first inequality applies $L$-Lipschitz continuity of $\nabla_1 f(\boldsymbol{x}_0^k; \cdot)$, implied by Assumption 1. On top of that, applying the Young's inequality to the term $\left\| \nabla_1 f(\boldsymbol{z}_0^k) \right\|^2$,

$$
\left\| \nabla_1 f(\boldsymbol{z}_0^k) \right\|^2 \leq 2 \left\| \nabla\Phi(\boldsymbol{x}_0^k) \right\|^2 + 2 \left\| \nabla\Phi(\boldsymbol{x}_0^k) - \nabla_1 f(\boldsymbol{z}_0^k) \right\|^2
$$
$$
\overset{(21)}{\leq} 2 \left\| \nabla\Phi(\boldsymbol{x}_0^k) \right\|^2 + 4L\kappa_2 \left[ \Phi(\boldsymbol{x}_0^k) - f(\boldsymbol{z}_0^k) \right] \tag{22}
$$

By applying inequalities (20), (21), and (22) to the bound (19), we conclude the proof. $\qquad\square$

In Lemma 19, we saw that if step sizes are chosen to satisfy certain conditions, then we can simplify the per-epoch progress a great deal. It is now left to choose appropriate step sizes and parameters (*e.g.*, $\lambda$) so as to make sure not only that $\alpha$ and $\beta$ meet the *small* step size conditions (17) but also that the constants $C_0$, $C_1$, $C_2$, and $C_3$ are positive.

**Lemma 20.** *Suppose that Assumptions 1, 2, 3 and 4 hold. Let $\lambda = 4$ and assume that*

$$
0 < \beta \leq \frac{1}{6L\sqrt{q^2 + \frac{q(q-1)}{n-1}A}}, \quad \alpha = \frac{\beta}{r}, \quad \text{where } r \geq 14\kappa_2^2.
$$

*Then these satisfy all the inequalities (17) and the terms defined in Lemma 19 satisfy*

$$
C_0 > 0, \quad C_1 > q\alpha, \quad C_2 > L\kappa_2 q\alpha/2, \quad C_3 \geq 0.
$$

*Consequently, due to the recurrence inequality in Lemma 19, mini-batch simSGDA-RR satisfies, for some numerical constant $c > 0$,*

$$
\mathbb{E}_k[V_\lambda(\boldsymbol{z}_0^{k+1})] - V_\lambda(\boldsymbol{z}_0^k)
$$
$$
\leq -q\alpha \left\| \nabla\Phi(\boldsymbol{x}_0^k) \right\|^2 - (L\kappa_2 q\alpha/2) \left[ \Phi(\boldsymbol{x}_0^k) - f(\boldsymbol{z}_0^k) \right] + (cr)^3 L^2 \left( \frac{q^2(q-1)}{n-1} \right) B\alpha^3. \tag{$\star$}
$$

Please note that we mark the recurrence inequality above with a special symbol $(\star)$ because this inequality is the exact point where the proofs of Theorems 4 and 5 start to deviate.

*Proof.* Regardless of $A \geq 0$, we have

$$
\beta \leq \frac{1}{6Lq} \quad \text{and} \quad \alpha \leq \frac{1}{6Lqr} \leq \frac{1}{84L\kappa_2^2 q}. \tag{23}
$$

This is enough to guarantee that the inequalities (17) hold with $\lambda = 4$. Since $C_0 > C_2/\mu_2$, it is enough to show $C_2 > 0$ to prove that $C_0 > 0$. Applying $\lambda = 4$, $\kappa_2 \geq 1$, and $\beta/\alpha = r \geq 14\kappa_2^2$,

$$
\frac{C_1}{q\alpha} = \frac{3}{2} - 2L^2 \left( q^2 + \frac{q(q-1)}{n-1}A \right) (9+r)\alpha^2
$$
$$
\geq \frac{3}{2} - \frac{2}{6^2} \cdot \frac{9+r}{r^2} \geq \frac{3}{2} - \frac{2 \cdot 23}{6^2 \cdot 14^2} > 1,
$$
$$
\frac{C_2}{\mu_2 q\beta} = 1 - \frac{12\kappa_2^2}{r} - 2L^2 \left( q^2 + \frac{q(q-1)}{n-1}A \right) \left( \frac{9}{r} + 1 \right) \left( \frac{2\kappa_2^2}{r^2} + 1 \right) \beta^2
$$
$$
\geq 1 - \frac{12}{14} - \frac{2}{6^2} \left( \frac{9}{14\kappa_2^2} + 1 \right) \left( \frac{2}{14^2\kappa_2^2} + 1 \right) \geq \frac{2}{14} - \frac{2 \cdot 23 \cdot 198}{6^2 \cdot 14^3} > \frac{1}{2 \cdot 14}.
$$

Thus, $C_1 > q\alpha$ and

$$
C_2 > \frac{\mu_2 q\beta}{2 \cdot 14} = \frac{\mu_2 qr\alpha}{2 \cdot 14} \geq L\kappa_2 q\alpha/2.
$$

Then we conclude the proof by bounding the term $C_3$. We can already check from the definition that $C_3 \geq 0$. We can upper-bound $C_3$ by

$$
C_3 = \left( \frac{L^2 q^2(q-1)}{n-1} \right) (9+r)(1+r^2)B\alpha^3 \leq (cr)^3 L^2 \left( \frac{q^2(q-1)}{n-1} \right) B\alpha^3,
$$

for some numerical constant $c > 0$. $\qquad\square$

## C.4 Convergence rates for smooth nonconvex-PŁ problem

In this subsection, we show the convergence bound of general smooth nonconvex-PŁ problems in terms of $\min_{k \in [K]} \mathbb{E}\left[\left\|\nabla \Phi(\boldsymbol{x}_0^k)\right\|^2\right]$. From the inequality $(\star)$ in Lemma 20, we can simply ignore the second term

$$-(L\kappa_2 q\alpha/2)\left[\Phi(\boldsymbol{x}_0^k) - f(\boldsymbol{z}_0^k)\right] \leq 0$$

of the right-hand side because $\Phi(\boldsymbol{x}) \geq f(\boldsymbol{x}; \boldsymbol{y})$ for any $(\boldsymbol{x}; \boldsymbol{y})$. In other words, we may deal with the inequality

$$\mathbb{E}_k[V_\lambda(\boldsymbol{z}_0^{k+1})] - V_\lambda(\boldsymbol{z}_0^k) \leq -q\alpha \left\|\nabla \Phi(\boldsymbol{x}_0^k)\right\|^2 + (cr)^3 L^2 \left(\frac{q^2(q-1)}{n-1}\right) B\alpha^3. \qquad \text{(nc-PŁ)}$$

Plugging $q = n/b$, we eventually show the convergence rate (Theorem 4). (Recall that $b$ is the size of mini-batches.)

**Theorem 21** (Equivalent to Theorem 4, for simSGDA-RR). *Suppose that $f$ satisfies Assumptions 1, 2, 3, and 4 are satisfied. Let $\lambda = 4$. Choose the step sizes $\alpha$ and $\beta$ by $\alpha = \beta/r$ for some $r \geq 14\kappa_2^2$ and*

$$\beta = \min\left\{\frac{1}{6L\sqrt{q^2 + \frac{q(q-1)}{n-1}A}}, \ \frac{1}{c}\left(\frac{V_\lambda(\boldsymbol{z}_0^1)}{L^2 q^2 (\frac{q-1}{n-1})BK}\right)^{\frac{1}{3}}\right\},$$

*for some numerical constant $c > 0$. Then, mini-batch simSGDA-RR satisfies*

$$\frac{1}{K}\sum_{k=1}^K \mathbb{E}\left[\left\|\nabla \Phi(\boldsymbol{x}_0^k)\right\|^2\right] \leq \frac{6rLV_\lambda(\boldsymbol{z}_0^1)}{K}\sqrt{1 + \left(\frac{q-1}{n-1}\right)\frac{A}{q}} + 2cr\left(\frac{L^2 B V_\lambda(\boldsymbol{z}_0^1)^2}{qK^2} \cdot \frac{q-1}{n-1}\right)^{1/3}.$$

*Proof.* To replace the conditional expectations with unconditional expectations, we take expectation to both sides of the inequality (nc-PŁ):

$$\mathbb{E}[V_\lambda(\boldsymbol{z}_0^{k+1}) - V_\lambda(\boldsymbol{z}_0^k)] \leq -q\alpha \mathbb{E}\left[\left\|\nabla \Phi(\boldsymbol{x}_0^k)\right\|^2\right] + (cr)^3 L^2 \left(\frac{q^2(q-1)}{n-1}\right) B\alpha^3.$$

Rearranging the terms and taking a sum from $k = 1$ to $k = K$, we have

$$q\alpha \sum_{k=1}^K \mathbb{E}\left[\left\|\nabla \Phi(\boldsymbol{x}_0^k)\right\|^2\right] \leq \mathbb{E}[V_\lambda(\boldsymbol{z}_0^1) - V_\lambda(\boldsymbol{z}_0^{K+1})] + (cr)^3 L^2 \left(\frac{q^2(q-1)}{n-1}\right) B\alpha^3 K.$$

Dividing both sides by $qK\alpha$, we get the following. Note that $V_\lambda$ is non-negative.

$$\frac{1}{K}\sum_{k=1}^K \mathbb{E}\left[\left\|\nabla \Phi(\boldsymbol{x}_0^k)\right\|^2\right] \leq \frac{V_\lambda(\boldsymbol{z}_0^1)}{qK\alpha} + (cr)^3 L^2 \left(\frac{q(q-1)}{n-1}\right) B\alpha^2$$

Since our choice of step sizes implies

$$\alpha = \min\left\{\frac{1}{6rL\sqrt{q^2 + \frac{q(q-1)}{n-1}A}}, \ \frac{1}{cr}\left(\frac{V_\lambda(\boldsymbol{z}_0^1)}{L^2 B q^2 (\frac{q-1}{n-1})K}\right)^{\frac{1}{3}}\right\},$$

we eventually prove the theorem by using the inequality $\max\{a, b\} \leq a + b$ (for $a, b \geq 0$). $\qquad \square$

## C.5 Convergence rates for smooth primal-PŁ-PŁ problem

In this subsection, we prove the convergence bound of primal-PŁ-PŁ (or, PŁ($\Phi$)-PŁ) problems in terms of $\mathbb{E}\left[V_\lambda(\boldsymbol{z}_0^{K+1})\right]$.

Unlike the previous subsection, we additionally utilize Assumption 5 stating that $f(\boldsymbol{x}; \boldsymbol{y})$ satisfies primal PŁ condition, namely, the primal function $\Phi(\boldsymbol{x}) = \max_{\boldsymbol{y}'} f(\boldsymbol{x}; \boldsymbol{y}')$ is a $\mu_1$-PŁ function. With this assumption, we yield another recurrence inequality from the inequality $(\star)$. We note that it uses the $\mu_1$-PŁ condition for $\Phi$ ($\because$ Proposition 10) but not necessarily for $f(\cdot; \boldsymbol{y})$.

**Lemma 22.** *Suppose that $f$ satisfies Assumptions 1, 2, 3, 4, and 5. Then, with the same choice of $\lambda = 4$ and the same condition of the step sizes $\alpha$ and $\beta$ as in Lemma 20, the mini-batch simSGDA-RR satisfies that, for some numerical constant $c > 0$,*

$$\mathbb{E}_k[V_\lambda(\boldsymbol{z}_0^{k+1})] \le (1 - \mu_1 q\alpha/2)V_\lambda(\boldsymbol{z}_0^k) + (cr)^3 L^2 \left(\frac{q^2(q-1)}{n-1}\right) B\alpha^3. \qquad \text{(PŁ($\Phi$)-PŁ)}$$

*Proof.* Since the primal function $\Phi$ is a $\mu_1$-PŁ function,

$$-\left\|\nabla\Phi(\boldsymbol{x}_0^k)\right\|^2 \le -2\mu_1\left[\Phi(\boldsymbol{x}_0^k) - \Phi^*\right].$$

Also, since $\mu_1 \le L$ and $\kappa_2 \ge 1$, we know that $-L\kappa_2 \le -\mu_1$. Applying these to the inequality ($\star$), we have

$$\mathbb{E}_k\left[V_\lambda(\boldsymbol{z}_0^{k+1})\right] - V_\lambda(\boldsymbol{z}_0^k)$$

$$\le -(2\mu_1 q\alpha/\lambda)\cdot\lambda\left[\Phi(\boldsymbol{x}_0^k) - \Phi^*\right] - (\mu_1 q\alpha/2)\left[\Phi(\boldsymbol{x}_0^k) - f(\boldsymbol{z}_0^k)\right] + (cr)^3 L^2\left(\frac{q^2(q-1)}{n-1}\right)B\alpha^3$$

$$= -(\mu_1 q\alpha/2)\cdot V_\lambda(\boldsymbol{z}_0^k) + (cr)^3 L^2\left(\frac{q^2(q-1)}{n-1}\right)B\alpha^3,$$

since $\lambda = 4$. By re-arranging the terms, we conclude the proof. $\qquad\square$

Of course, the multiplier $1 - \mu_1 q\alpha/2$ has a value between 0 and 1. To see why, note that from Equation (23),

$$0 < \mu_1 q\alpha/2 \le \mu_1 q\cdot\frac{1}{2\cdot 84L\kappa_2^2 q} = \frac{1}{168\kappa_1\kappa_2^2} < 1.$$

**Theorem 23** (Equivalent to Theorem 5, for simSGDA-RR). *Assume that $f$ satisfies Assumptions 1, 2, 3, 4, and 5. Let $\lambda = 4$. Choose the step sizes by $\alpha = \beta/r$ for some $r \ge 14\kappa_2^2$ and*

$$\beta = \min\left\{\frac{1}{6L\sqrt{q^2 + \frac{q(q-1)}{n-1}A}}, \frac{2r}{\mu_1 qK}\max\left\{1, \log\left(\frac{V_\lambda(\boldsymbol{z}_0^1)\mu_1 qK^2}{8(cr)^3\kappa_1^2\left(\frac{q-1}{n-1}\right)B}\right)\right\}\right\},$$

*for some numerical constant $c > 0$. Then, mini-batch simSGDA-RR satisfies*

$$\mathbb{E}[V_\lambda(\boldsymbol{z}_n^K)] \le \mathcal{O}\left(V_\lambda(\boldsymbol{z}_0^1)\cdot\exp\left(-\frac{K}{12\kappa_1 r\sqrt{1 + \left(\frac{q-1}{n-1}\right)\frac{A}{q}}}\right)\right) + \tilde{\mathcal{O}}\left(\frac{\kappa_1^2 r^3 B}{\mu_1 qK^2}\right)\cdot\frac{q-1}{n-1}.$$

*Proof.* To replace the conditional expectations with unconditional expectations, we take expectation to both sides of the inequality (PŁ($\Phi$)-PŁ):

$$\mathbb{E}\left[V_\lambda(\boldsymbol{z}_0^{k+1})\right] \le (1 - \mu_1 q\alpha/2)\mathbb{E}\left[V_\lambda(\boldsymbol{z}_0^k)\right] + (cr)^3 L^2\left(\frac{q^2(q-1)}{n-1}\right)B\alpha^3.$$

Unrolling the recurrence inequality (Proposition 15) and using the facts $\beta = 14\kappa_2^2\alpha$, we have

$$\mathbb{E}[V_\lambda(\boldsymbol{z}_n^K)] \le (1 - \mu_1 q\alpha/2)^K V_\lambda(\boldsymbol{z}_0^1) + \frac{2\cdot(cr)^3 L^2}{\mu_1 q\alpha}\left(\frac{q^2(q-1)}{n-1}\right)B\alpha^3$$

$$\le \exp(-\mu_1 qK\alpha/2)V_\lambda(\boldsymbol{z}_0^1) + 2(cr)^3\mu_1\kappa_1^2\left(\frac{q(q-1)}{n-1}\right)B\alpha^2. \qquad (24)$$

Note that, in the inequality above, the second term of the right hand side becomes zero when $q = 1$. In that case, we can prove exponential decay of $\mathbb{E}[V_\lambda(\boldsymbol{z}_0^k)]$. Thus, we simply assume $q > 1$ hereafter.

*Case 1:* If $K$ is as large as

$$K > \frac{\kappa_1 r^{3/2}}{\sqrt{\mu_1}}\cdot\sqrt{\frac{8c^3 eB}{V_\lambda(\boldsymbol{z}_0^1)q}\left(\frac{q-1}{n-1}\right)}, \quad (e = \exp(1))$$

we have a step size $\alpha$ as

$$\alpha = \min\left\{\frac{1}{6Lr\sqrt{q^2 + \frac{q(q-1)}{n-1}A}}, \frac{2}{\mu_1 qK}\log(\clubsuit)\right\}, \quad \text{where } \clubsuit = \frac{V_\lambda(z_0^1)\mu_1 qK^2}{8(cr)^3\kappa_1^2\kappa_2^6\left(\frac{q-1}{n-1}\right)B}.$$

Due to the lower bound of epoch size $K$, the fraction $\clubsuit$ inside the log factor is indeed greater than $e > 1$, which guarantees the step size is positive. Putting this to the inequality (24) and using the fact that $\max\{a, b\} \leq a + b$ (for $a, b \geq 0$), we eventually have

$$\mathbb{E}\left[V_\lambda(z_n^K)\right]$$

$$\leq V_\lambda(z_0^1) \cdot \exp\left(-\frac{K}{12\kappa_1 r\sqrt{1 + \left(\frac{q-1}{n-1}\right)\frac{A}{q}}}\right) + \frac{2 \cdot 8(cr)^3\kappa_1^2 B}{\mu_1 qK^2}\left(\frac{q-1}{n-1}\right)\left[1 + \log^2(\clubsuit)\right]$$

$$= V_\lambda(z_0^1) \cdot \exp\left(-\frac{K}{12\kappa_1 r\sqrt{1 + \left(\frac{q-1}{n-1}\right)\frac{A}{q}}}\right) + \tilde{\mathcal{O}}\left(\frac{\kappa_1^2 r^3 B}{\mu_1 qK^2}\right) \cdot \frac{q-1}{n-1}.$$

*Case 2:* Otherwise, the log factor might have a negative value when $K$ is too small. However, in this case, we have

$$V_\lambda(z_0^1) \leq \frac{8(cr)^3 e\kappa_1^2 B}{\mu_1 qK^2} \cdot \frac{q-1}{n-1}; \quad \alpha = \min\left\{\frac{1}{84L\kappa_2^2\sqrt{q^2 + \frac{q(q-1)}{n-1}A}}, \frac{2}{\mu_1 qK}\right\}.$$

Putting these to the inequality (24), we have

$$\mathbb{E}\left[V_\lambda(z_n^K)\right] \leq \frac{8(cr)^3 e\kappa_1^2 B}{\mu_1 qK^2}\left(\frac{q-1}{n-1}\right)\left[\exp(-\mu_1 qK\alpha/2) + \frac{1}{e} \cdot (\mu_1 qK\alpha/2)^2\right]$$

$$\leq \frac{8(cr)^3 e\kappa_1^2 B}{\mu_1 qK^2}\left(\frac{q-1}{n-1}\right) = \mathcal{O}\left(\frac{\kappa_1^2 r^3 B}{\mu_1 qK^2}\right) \cdot \frac{q-1}{n-1}.$$

The inequality in the last line is due to the fact that $e^{-t} + t^2/e \leq 1$ for each $t \in (0, 1]$, and that $\mu_1 qK\alpha/2 \in (0, 1]$.

Combining both *Case 1* and *Case 2*, we conclude the proof of the theorem. $\qquad\square$

# D PROOFS FOR (MINI-BATCH) ALTERNATING SGDA-RR: FOCUSING ON CHANGES IN THE PROOF

In this appendix, we prove the same convergence rates for altSGDA-RR as the simultaneous update counterpart. Since most of the steps in the proof are similar to those in Appendix C, we only describe which steps change in the proof.

## D.1 EPOCH-WISE REPRESENTATIONS AND BOUNDING NOISE TERMS

To analyze altSGDA-RR, we modify the notation for epoch-wise updates. The only change is that an update $y_t^k \mapsto y_{t+1}^k$ uses $x_{t+1}^k$ instead of $x_t^k$. Hence, the definition of $h^k$ should be modified. Recall that

$$g_t^k(z) := \frac{1}{b}\sum_{i \in \mathcal{B}_t^k} \nabla_1 f_i(z), \quad h_t^k(z) := \frac{1}{b}\sum_{i \in \mathcal{B}_t^k} \nabla_2 f_i(z),$$

where $\mathcal{B}_t^k$ is a mini-batch of size $b$ formed at iteration $t$ of epoch $k$. Then, at epoch $k$, by re-definition of $h^k$,

$$g^k := \frac{1}{q}\sum_{t=1}^{q} g_t^k(x_{t-1}^k; y_{t-1}^k), \quad h^k := \frac{1}{q}\sum_{t=1}^{q} h_t^k(x_t^k; y_{t-1}^k).$$

$$x_0^{k+1} = x_0^k - q\alpha g^k, \quad y_0^{k+1} = y_0^k + q\beta h^k. \qquad \text{(altSGDA-RR)}$$

We still approximate this epoch-wise update rule to a full-batch simultaneous GDA update ($\approx$simGDA) with step sizes $q\alpha$ and $q\beta$. Again, we control the "noise" terms $\left\|\boldsymbol{g}^k - \nabla_1 f(\boldsymbol{z}_0^k)\right\|^2$ and $\left\|\boldsymbol{h}^k - \nabla_2 f(\boldsymbol{z}_0^k)\right\|^2$ not to be large. Because of the modification of $\boldsymbol{h}^k$, we have a different result for $\left\|\boldsymbol{h}^k - \nabla_2 f(\boldsymbol{z}_0^k)\right\|^2$ as follows.

**Lemma 24.** *For mini-batch altSGDA-RR, recall that*

$$G_k := \frac{1}{q} \sum_{t=1}^{q} \left\|\boldsymbol{z}_{t-1}^k - \boldsymbol{z}_0^k\right\|^2.$$

*If we have Assumption 1, then we have*

$$\left\|\boldsymbol{h}^k - \nabla_2 f(\boldsymbol{z}_0^k)\right\|^2 \le L^2 G_k + L^2 q\alpha^2 \left\|\boldsymbol{g}^k\right\|^2, \quad \text{whereas} \quad \left\|\boldsymbol{g}^k - \nabla_1 f(\boldsymbol{z}_0^k)\right\|^2 \le L^2 G_k. \quad (25)$$

*Proof.* Because of $L$-Lipschitz continuity of $\boldsymbol{h}_t^k(\cdot; \cdot)$,

$$
\begin{aligned}
\left\|\boldsymbol{h}^k - \nabla_2 f(\boldsymbol{z}_0^k)\right\|^2 &= \left\|\frac{1}{q} \sum_{t=1}^{q} \left[\boldsymbol{h}_t^k(\boldsymbol{x}_t^k; \boldsymbol{y}_{t-1}^k) - \boldsymbol{h}_t^k(\boldsymbol{x}_0^k; \boldsymbol{y}_0^k)\right]\right\|^2 \\
&\le \frac{1}{q} \sum_{t=1}^{q} \left\|\boldsymbol{h}_t^k(\boldsymbol{x}_t^k; \boldsymbol{y}_{t-1}^k) - \boldsymbol{h}_t^k(\boldsymbol{x}_0^k; \boldsymbol{y}_0^k)\right\|^2 \\
&\le \frac{L^2}{q} \sum_{t=1}^{q} \left\|\boldsymbol{z}_{t-1}^k - \boldsymbol{z}_0^k\right\|^2 + \frac{L^2}{q} \left\|\boldsymbol{x}_q^k - \boldsymbol{x}_0^k\right\|^2 = L^2 G_k + L^2 q\alpha^2 \left\|\boldsymbol{g}^k\right\|^2.
\end{aligned}
$$

The last ineqaulity holds because $\boldsymbol{x}_q^k = \boldsymbol{x}_0^{k+1}$. $\qquad\qquad\qquad\qquad\qquad\qquad\qquad\qquad\square$

### D.2 BOUNDING NOISE TERMS: A BIT DIFFERENT PROOF OF LEMMA 18

We notice that the same result as Lemma 18 holds not only for simultaneous updates but also alternating updates, even though it is not very straightforward. We need to reflect the changes from the previous subsection. That is, we have to be careful when we expand the term $\left\|\boldsymbol{y}_t^k - \boldsymbol{y}_0^k\right\|^2$ $(0 \le t \le q-1)$. Unlike the inequality (12) (in the original proof), we have

$$
\begin{aligned}
\left\|\boldsymbol{y}_t^k - \boldsymbol{y}_0^k\right\|^2 &= \beta^2 t^2 \left\|\frac{1}{t} \sum_{j=1}^{t} \boldsymbol{h}_j^k(\boldsymbol{x}_j^k; \boldsymbol{y}_{j-1}^k)\right\|^2 \\
&\le 3\beta^2 t^2 \left[\left\|\frac{1}{t} \sum_{j=1}^{t} \left[\boldsymbol{h}_j^k(\boldsymbol{x}_j^k; \boldsymbol{y}_{j-1}^k) - \boldsymbol{h}_j^k(\boldsymbol{z}_0^k)\right]\right\|^2 + \left\|\frac{1}{t} \sum_{j=1}^{t} \boldsymbol{h}_j^k(\boldsymbol{z}_0^k) - \nabla_2 f(\boldsymbol{z}_0^k)\right\|^2 + \left\|\nabla_2 f(\boldsymbol{z}_0^k)\right\|^2\right] \\
&\le 3\beta^2 t^2 \left[\frac{1}{t} \sum_{j=1}^{t} \left\|\boldsymbol{h}_j^k(\boldsymbol{x}_j^k; \boldsymbol{y}_{j-1}^k) - \boldsymbol{h}_j^k(\boldsymbol{z}_0^k)\right\|^2 + \left\|\frac{1}{t} \sum_{j=1}^{t} \boldsymbol{h}_j^k(\boldsymbol{z}_0^k) - \nabla_2 f(\boldsymbol{z}_0^k)\right\|^2 + \left\|\nabla_2 f(\boldsymbol{z}_0^k)\right\|^2\right] \\
&\le 3\beta^2 t^2 \left[\frac{L^2}{t} \left(\left\|\boldsymbol{x}_t^k - \boldsymbol{x}_0^k\right\|^2 + \sum_{j=1}^{t} \left\|\boldsymbol{z}_{j-1}^k - \boldsymbol{z}_0^k\right\|^2\right) + \left\|\frac{1}{t} \sum_{j=1}^{t} \boldsymbol{h}_j^k(\boldsymbol{z}_0^k) - \nabla_2 f(\boldsymbol{z}_0^k)\right\|^2 + \left\|\nabla_2 f(\boldsymbol{z}_0^k)\right\|^2\right] \\
&\le 3\beta^2 L^2 t \sum_{j=1}^{t} \left\|\boldsymbol{z}_j^k - \boldsymbol{z}_0^k\right\|^2 + 3\beta^2 t^2 \left[\left\|\frac{1}{t} \sum_{j=1}^{t} \boldsymbol{h}_j^k(\boldsymbol{z}_0^k) - \nabla_2 f(\boldsymbol{z}_0^k)\right\|^2 + \left\|\nabla_2 f(\boldsymbol{z}_0^k)\right\|^2\right] \\
&\le 3\beta^2 L^2 t \cdot q G_k + 3\beta^2 t^2 \left[\left\|\frac{1}{t} \sum_{j=1}^{t} \boldsymbol{h}_j^k(\boldsymbol{z}_0^k) - \nabla_2 f(\boldsymbol{z}_0^k)\right\|^2 + \left\|\nabla_2 f(\boldsymbol{z}_0^k)\right\|^2\right].
\end{aligned}
$$

The second and third inequality holds by Jensen's inequality, and the last inequality holds because $t \leq q - 1$. The resulting upper bound is identical to the inequality (13). Proving this inequality above suffices to show that the conclusion of Lemma 18 also holds for altSGDA-RR, because we eventually take an average along $0 \leq t \leq q - 1$ and the other steps in the proof do not utilize the "order" (either simultaneous or alternating) of updates.

### D.3 RECURRENCE INEQUALITIES FOR GENERAL SMOOTH NONCONVEX-PŁ OBJECTIVE

In the proof for simSGDA-RR, we applied Lemma 16, Lemma 18, and the "small-step-size" assumptions (three inequalities in (17)) to deduce Lemma 19. However, due to Lemma 24 that we obtained for altSGDA-RR, we need slightly different assumptions on step sizes rather than (17).

Fortunately, we notice that the Lemma 16 also holds for altSGDA-RR, with a modified version of $\boldsymbol{h}^k$. This is because the proof of the lemma does not utilize step-wise updates, while the discrepancy between simultaneous and alternating updates only appears in the step-wise updates. Thus, we have the same result as Lemma 19.

**Lemma 25.** *Suppose that Assumptions 1, 2, 3, and 4 hold. Modify the inequalities (17) (from Lemma 19) by*

$$\lambda - \{(\lambda + 1)(\kappa_2 + 1) + 1\} Lq\alpha - L^2 q\alpha\beta \geq 0, \quad \beta \leq \frac{1}{qL}, \quad \alpha^2 + \beta^2 \leq \frac{1}{3q(q-1)L^2}. \quad (26)$$

*(In fact, only the first one is different.) Then, the result of Lemma 19 still holds for mini-batch altSGDA-RR.*

*Proof.* We first apply Lemma 24 to the general bound resulted from Lemma 16:

$$V_\lambda(\boldsymbol{z}_0^{k+1}) - V_\lambda(\boldsymbol{z}_0^k)$$
$$\leq - \left(\frac{\lambda + 1}{2}\right) q\alpha \left\|\nabla\Phi(\boldsymbol{x}_0^k)\right\|^2 + (\lambda + 1)q\alpha \left\|\nabla\Phi(\boldsymbol{x}_0^k) - \nabla_1 f(\boldsymbol{z}_0^k)\right\|^2$$
$$+ \frac{q\alpha}{2}\left\|\nabla_1 f(\boldsymbol{z}_0^k)\right\|^2 - \frac{q\beta}{2}\left\|\nabla_2 f(\boldsymbol{z}_0^k)\right\|^2 + \frac{(2\lambda + 1)\alpha + \beta}{2}qL^2 G_k$$
$$- \left[\lambda - \{(\lambda + 1)(\kappa_2 + 1) + 1\} Lq\alpha - L^2 q\alpha\beta\right]\frac{q\alpha}{2}\left\|\boldsymbol{g}^k\right\|^2 - (1 - Lq\beta)\frac{q\beta}{2}\left\|\boldsymbol{h}^k\right\|^2. \quad (27)$$

Hence, the first two inequalities of (26) eliminate the last two terms on the right side of the inequality (27) above:

$$V_\lambda(\boldsymbol{z}_0^{k+1}) - V_\lambda(\boldsymbol{z}_0^k) \leq - \left(\frac{\lambda + 1}{2}\right) q\alpha \left\|\nabla\Phi(\boldsymbol{x}_0^k)\right\|^2 + (\lambda + 1)q\alpha \left\|\nabla\Phi(\boldsymbol{x}_0^k) - \nabla_1 f(\boldsymbol{z}_0^k)\right\|^2$$
$$+ \frac{q\alpha}{2}\left\|\nabla_1 f(\boldsymbol{z}_0^k)\right\|^2 - \frac{q\beta}{2}\left\|\nabla_2 f(\boldsymbol{z}_0^k)\right\|^2 + \frac{(2\lambda + 1)\alpha + \beta}{2}qL^2 G_k.$$

This is identical to the inequality (18) in the proof of Lemma 19. From this point on, the rest of the proof is exactly identical to Lemma 19. $\qquad\square$

Lemma 25 establishes that altSGDA-RR also satisfies a concise bound on the expected per-epoch change of $V_\lambda$, albeit under a slightly different set of assumptions (26) on step sizes. Using this result, we can prove the convergence rates for altSGDA-RR that are exactly the same as simSGDA-RR.

### D.4 SMALL STEP SIZE ASSUMPTIONS

It is left to show an altSGDA-RR counterpart for Lemma 20 which establishes the general recurrence inequality ($\star$). In fact, the same choice of step sizes as simSGDA-RR, namely

$$0 < \beta \leq \frac{1}{6L\sqrt{q^2 + \frac{q(q-1)}{n-1}A}} \quad \text{and} \quad \alpha = \frac{\beta}{r} \quad \text{where } r \geq 14\kappa_2^2,$$

actually meets the newly introduced conditions (26). Among the three inequalities, the only one that needs to be checked is

$$\lambda - \{(\lambda + 1)(\kappa_2 + 1) + 1\} Lq\alpha - L^2 q\alpha\beta > 0.$$

Note that, regardless of $A \geq 0$,

$$\beta \leq \frac{1}{6Lq} \quad \text{and} \quad \alpha \leq \frac{1}{6Lqr} \leq \frac{1}{84L\kappa_2^2 q}$$

In this case,

$$\lambda - \{(\lambda + 1)(\kappa_2 + 1) + 1\} Lq\alpha - L^2 q\alpha\beta$$
$$\geq 4 - (11\kappa_2 + L\beta)Lq\alpha \geq 4 - \left(11\kappa_2 + \frac{1}{6}\right) \cdot \frac{1}{84\kappa_2^2} > 0.$$

Therefore, there is no need to modify our choices of $\lambda$ and the step sizes $\alpha, \beta$ for the analysis of altSGDA-RR, and the rest of the proof for simSGDA-RR goes through.

# E    Proofs for lower bound of deterministic full-batch simGDA

In this appendix, we illustrate a comprehensive lower bound for full-batch GDA, which is specific to the choice of step size ratio (Theorem 3). Before we start the proof, we define a class of smooth strongly-convex-strongly concave functions.

**Definition 4.** *Let $\mathcal{F}(L, \mu_1, \mu_2)$ be the class of functions $f(\boldsymbol{x}; \boldsymbol{y})$ with two arguments $\boldsymbol{x}$ and $\boldsymbol{y}$ of any dimension, which is $L$-smooth, $\mu_1$-strongly-convex in $\boldsymbol{x}$, and $\mu_2$-strongly-concave in $\boldsymbol{y}$. Let $\kappa_1 = L/\mu_1 \geq 1$ and $\kappa_2 = L/\mu_2 \geq 1$ be condition numbers of the function class. Denote the (unique) saddle (or, global minimax) point by $\boldsymbol{z}^* = (\boldsymbol{x}^*; \boldsymbol{y}^*)$.*

We restate and prove the Theorem 3 for reader's convenience.

**Theorem 26** (Restatement of Theorem 3)**.** *Suppose $\kappa_1 \geq c$ and $\kappa_2 \geq c$ for some constant $c > 1$. Then, for each step size ratio $r > 0$, there exists a function $f \in \mathcal{F}(L, \mu_1, \mu_2)$ for which simGDA with any step sizes $\alpha$ and $\beta$ of ratio $r = \beta/\alpha$ requires*

$$K = \begin{cases} \Omega\left(\kappa_1 r \log(1/\varepsilon)\right), & \text{if } r \geq \kappa_2/c, \\ \Omega\left(\kappa_1 \kappa_2 \log(1/\varepsilon)\right), & \text{if } c/\kappa_1 \leq r \leq \kappa_2/c, \\ \Omega((\kappa_2/r) \log(1/\varepsilon)), & \text{if } 0 < r \leq c/\kappa_1 \end{cases}$$

*iterations to achieve either $\|\boldsymbol{z}_k - \boldsymbol{z}^*\|^2 \leq \varepsilon^2$ or $V_\lambda(\boldsymbol{z}_K) \leq \varepsilon^2$.*

*Proof.* The proof is done in case by case, constructing a worst-case function for each of 4 different regimes of step size ratio $r$: *(1)* $\mu_1/\mu_2 \leq r \leq \kappa_2/c$, *(2)* $c/\kappa_1 \leq r \leq \mu_1/\mu_2$, *(3)* $r \geq \kappa_2/c$, and *(4)* $0 < r \leq c/\kappa_1$. Readers might notice the similarities of the proofs for *(1)*$\leftrightarrow$*(2)* and *(3)*$\leftrightarrow$*(4)*.

Case 1. $(\mu_1/\mu_2 \leq r \leq \kappa_2/c)$. Consider

$$f^{(1)}(v, x; y) := \frac{\mu_1}{2}v^2 + \frac{r\mu_2}{2}x^2 - \frac{\mu_2}{2}y^2 + \ell xy,$$

where $\ell^2 = L^2 - r\mu_2^2 - L\mu_2|r - 1| \geq 0$. Applying Proposition 28, it can be shown that $f^{(1)} \in \mathcal{F}(L, \mu_1, \mu_2)$. Also, $\boldsymbol{z}^* = (0, 0; 0)$ is its unique saddle point. Note that, the GDA on $f^{(1)}$ can be written as

$$v_{t+1} = \left(1 - \frac{\beta\mu_1}{r}\right) v_t, \quad \begin{bmatrix} x_{t+1} \\ y_{t+1} \end{bmatrix} = \underbrace{\begin{bmatrix} 1 - \beta\mu_2 & -\beta\ell/r \\ \beta\ell & 1 - \beta\mu_2 \end{bmatrix}}_{\boldsymbol{A}} \begin{bmatrix} x_t \\ y_t \end{bmatrix} = \boldsymbol{A} \begin{bmatrix} x_t \\ y_t \end{bmatrix}.$$

Also, the eigenvalues $\tau$ of $\boldsymbol{A}$ is

$$\tau = 1 - \beta\mu_2 \pm \sqrt{(1 - \beta\mu_2)^2 - ((1 - \beta\mu_2)^2 + \beta^2\ell^2/r)}$$
$$= 1 - \beta\mu_2 \pm \frac{\beta\ell}{\sqrt{r}}\sqrt{-1}.$$

The spectral radius (*i.e.*, maximum absolute eigenvalue) is

$$\rho(\boldsymbol{A}) = \sqrt{(1 - \beta\mu_2)^2 + \beta^2\ell^2/r}.$$

Since the eigenvalues are complex conjugates of each other (the magnitudes are the same), both eigenvalues have magnitude $\rho(\boldsymbol{A})$. Then, by Proposition 27, $\rho(\boldsymbol{A}) < 1$ is necessary for convergence. To this end, we need $\beta > 0$ satisfying $\beta < 2\mu_2 r/(r\mu_2^2 + \ell^2)$.

To guarantee $\|(v_k, x_k; y_k) - (0, 0; 0)\|^2 \le \varepsilon^2$, we need a large enough $k$ to have $v_k^2 \le \mathcal{O}(\varepsilon^2)$. Such a $k$ is required to be at least $\Omega\left(\frac{r}{\beta\mu_1}\log(1/\varepsilon)\right)$. Now note that, since $\mu_1/\mu_2 \le r \le \kappa_2/c$ and $\kappa_2 \ge c$,

$$\frac{1}{\beta} > \frac{r\mu_2^2 + \ell^2}{2\mu_2 r} = \frac{L^2 - L\mu_2|r - 1|}{2\mu_2 r} = \frac{L^2}{2\mu_2 r}\left(1 - \frac{|r-1|}{\kappa_2}\right) \ge \frac{L^2}{2\mu_2 r}\left(1 - \frac{1}{c}\right).$$

The last inequality is true by minimizing $\left(1 - \frac{|r-1|}{\kappa_2}\right)$ for $r \in [\mu_1/\mu_2, \kappa_2/c]$. If $r \ge 1$, it has smaller value when $r$ is larger: by taking $r = \kappa_2/c$, we have $1 - \frac{\kappa_2/c - 1}{\kappa_2} \ge 1 - \frac{1}{c}$. Otherwise ($r < 1$), which is possible only when $\mu_1 < \mu_2$, the term has smaller value when $r$ is smaller: by taking $r = \mu_1/\mu_2$, we have $1 + \frac{\mu_1/\mu_2 - 1}{\kappa_2} = 1 + \frac{\mu_1 - \mu_2}{L} \ge 1 - \frac{1}{\kappa_2} \ge 1 - \frac{1}{c}$. Thus, we eventually need $\Omega\left(\frac{L^2}{\mu_1\mu_2}\log(1/\varepsilon)\right)$ iterations.

Case 2. ($c/\kappa_1 \le r \le \mu_1/\mu_2$). Consider

$$f^{(2)}(x; y, w) := \frac{\mu_1}{2}x^2 - \frac{\mu_1}{2r}y^2 + \tilde{\ell}xy - \frac{\mu_2}{2}w^2,$$

where $\tilde{\ell}^2 = L^2 - \mu_1^2/r - L\mu_1|1 - 1/r| \ge 0$. Applying Proposition 28, it can be shown that $f^{(2)} \in \mathcal{F}(L, \mu_1, \mu_2)$, and $\boldsymbol{z}^* = (0; 0, 0)$ is its unique saddle point. Note that, the GDA on $f^{(2)}$ can be written as

$$\begin{bmatrix} x_{t+1} \\ y_{t+1} \end{bmatrix} = \underbrace{\begin{bmatrix} 1 - \beta\mu_1/r & -\beta\ell/r \\ \beta\ell & 1 - \beta\mu_1/r \end{bmatrix}}_{\boldsymbol{B}}\begin{bmatrix} x_t \\ y_t \end{bmatrix} = \boldsymbol{B}\begin{bmatrix} x_t \\ y_t \end{bmatrix}, \quad w_{t+1} = (1 - \beta\mu_2)\,w_t.$$

Also, the eigenvalues $\tau$ of $\boldsymbol{B}$ is

$$\tau = 1 - \beta\mu_1/r \pm \sqrt{(1 - \beta\mu_1/r)^2 - ((1 - \beta\mu_1/r)^2 + \beta^2\ell^2/r)}$$
$$= 1 - \frac{\beta\mu_1}{r} \pm \frac{\beta\ell}{\sqrt{r}}\sqrt{-1}.$$

The spectral radius is
$$\rho(\boldsymbol{B}) = \sqrt{(1 - \beta\mu_1/r)^2 + \beta^2\ell^2/r}.$$

Since the eigenvalues are complex conjugates of each other (the magnitudes are the same), both eigenvalues have magnitude $\rho(\boldsymbol{B})$. Then, by Proposition 27, $\rho(\boldsymbol{B}) < 1$ is necessary for convergence. To this end, we need $\beta > 0$ satisfying $\beta < 2\mu_1/(\mu_1^2/r + \ell^2)$.

To guarantee $\|(x_k; y_k, w_k) - (0; 0, 0)\|^2 \le \varepsilon^2$, we need a large enough $k$ to have $w_k^2 \le \mathcal{O}(\varepsilon^2)$. Such a $k$ is required to be at least $\Omega\left(\frac{1}{\beta\mu_2}\log(1/\varepsilon)\right)$. Now note that, since $c/\kappa_1 \le r \le \mu_1/\mu_2$ and $\kappa_1 \ge c$,

$$\frac{1}{\beta} > \frac{\mu_1^2/r + \ell^2}{2\mu_1} = \frac{L^2 - L\mu_1|1 - 1/r|}{2\mu_1} = \frac{L^2}{2\mu_1}\left(1 - \frac{|1 - 1/r|}{\kappa_1}\right) \ge \frac{L^2}{2\mu_1}\left(1 - \frac{1}{c}\right).$$

The last inequality is true by minimizing $\left(1 - \frac{|1 - 1/r|}{\kappa_1}\right)$ for $r \in [c/\kappa_1, \mu_1/\mu_2]$. If $1 > 1/r$, which is possible only when $\mu_1 > \mu_2$, it has smaller value when $r$ is larger: by taking $r = \mu_1/\mu_2$, we have $1 - \frac{1 - \mu_2/\mu_1}{\kappa_1} = 1 - \frac{\mu_1 - \mu_2}{L} \ge 1 - \frac{1}{\kappa_1} \ge 1 - \frac{1}{c}$. Otherwise ($1 < 1/r$), the term has smaller value when $r$ is smaller: by taking $r = c/\kappa_1$, we have $1 + \frac{1 - \kappa_1/c}{\kappa_1} \ge 1 - \frac{1}{c}$. Thus, we eventually need $\Omega\left(\frac{L^2}{\mu_1\mu_2}\log(1/\varepsilon)\right)$ iterations.

__Case 3.__ $(r \geq \kappa_2/c)$. Consider $f^{(3)}(x;y) = \frac{\mu_1}{2}x^2 - \frac{L}{2}y^2$. Clearly, $f^{(3)} \in \mathcal{F}(L, \mu_1, L) \subset \mathcal{F}(L, \mu_1, \mu_2)$ and $\boldsymbol{z}^* = (0,0)$ is its unique saddle point. The GDA on $f^{(3)}$ can be written as

$$x_{k+1} = \left(1 - \frac{\beta\mu_1}{r}\right)x_k, \quad y_{k+1} = (1 - \beta L)\,y_k.$$

To guarantee $\|(x_k; y_k) - (0,0)\|^2 \leq \varepsilon^2$, we need a large enough $k$ to have $x_k^2 \leq \mathcal{O}(\varepsilon^2)$. Such a $k$ is required to be at least $\Omega\left(\frac{r}{\beta\mu_1}\log(1/\varepsilon)\right)$. Also, we need $\beta < 2/L$ to guarantee $y_k \to 0$ (*i.e.,* otherwise, it diverges). Combining these facts, we eventually need $\Omega\left(\frac{Lr}{\mu_1}\log(1/\varepsilon)\right)$ iterations.

__Case 4.__ $(0 < r \leq c/\kappa_1)$. Consider $f^{(4)}(x;y) = \frac{L}{2}x^2 - \frac{\mu_2}{2}y^2$. Clearly, $f^{(4)} \in \mathcal{F}(L, L, \mu_2) \subset \mathcal{F}(L, \mu_1, \mu_2)$ and $\boldsymbol{z}^* = (0,0)$ is its unique saddle point. The GDA on $f^{(4)}$ can be written as

$$x_{k+1} = \left(1 - \frac{\beta L}{r}\right)x_k, \quad y_{k+1} = (1 - \beta\mu_2)\,y_k.$$

To guarantee $\|(x_k; y_k) - (0,0)\|^2 \leq \varepsilon^2$, we need a large enough $k$ to have $y_k^2 \leq \mathcal{O}(\varepsilon^2)$. Such a $k$ is required to be at least $\Omega\left(\frac{1}{\beta\mu_2}\log(1/\varepsilon)\right)$. Also, we need $\beta < 2r/L$ to guarantee $x_k \to 0$ (*i.e.,* otherwise, it diverges). Combining these facts, we eventually need $\Omega\left(\frac{L}{r\mu_2}\log(1/\varepsilon)\right)$ iterations.

Lastly, we note that the lower iteration complexity bound in terms of the potential function $V_\lambda$ is equivalent to the complexity in terms of squared distance norm from the (unique) saddle point $\boldsymbol{z}^*$, up to constant factors. This is proved in Lemma 29 that we defer its proof. $\qquad\square$

Here are the postponed/omitted proofs from the proof above.

__Proposition 27.__ *For a square matrix $\boldsymbol{A} \in \mathbb{R}^{m \times m}$ and a sequence of $m$-dimensional vectors $(\boldsymbol{v}_k)$, the matrix iteration $\boldsymbol{v}_{k+1} = \boldsymbol{A}\boldsymbol{v}_k$ converges to $\boldsymbol{v}_k \to \boldsymbol{0}$ if and only if the spectral radius (i.e., maximum absolute eigenvalue) of $\rho(\boldsymbol{A})$ of $\boldsymbol{A}$ is less than 1. Furthermore, its convergence speed is characterized by $\mathcal{O}((\rho(\boldsymbol{A}) + \varepsilon)^k)$ for any (arbitrarily small) $\varepsilon > 0$.*

*Proof.* See Horn & Johnson (2012, Theorem 5.6.10-12). $\qquad\square$

__Proposition 28.__ *Let $\mu_1$, $\mu_2$, and $L$ be positive numbers such that $L \geq \max\{\mu_1, \mu_2\}$. Consider a quadratic function $f$ on $\mathbb{R} \times \mathbb{R}$ defined by*

$$f(x;y) = \frac{\mu_1}{2}x^2 - \frac{\mu_2}{2}y^2 + \ell xy, \quad \text{where } \ell^2 \leq L^2 - \mu_1\mu_2 - L|\mu_1 - \mu_2|.$$

*Then, $f \in \mathcal{F}(L, \mu_1, \mu_2)$, and its unique saddle point is $\boldsymbol{z}^* = (0,0)$.*

*For example, if $\mu_1 \geq \mu_2$, $\ell^2 = (L - \mu_1)(L + \mu_2)$ is enough to guarantee $L$-smoothness.*

*Proof.* The strong-convex-strong-concavity is trivially true. Note that the gradient and hessian of $f$ is

$$\nabla f(x;y) = \boldsymbol{H}[x\ y]^\top, \quad \boldsymbol{H} = \begin{bmatrix} \mu_1 & \ell \\ \ell & -\mu_2 \end{bmatrix}.$$

Since $\boldsymbol{H}$ is a non-singular matrix, $f$ has a unique stationary point at origin ($x = 0, y = 0$). By Proposition 11, it is also a unique saddle & global minimax point.

For any two distinct points $\boldsymbol{z}_1 = (x_1; y_1)$ and $\boldsymbol{z}_2 = (x_2; y_2)$ in $\mathbb{R} \times \mathbb{R}$,

$$\frac{\|\nabla f(\boldsymbol{z}_1) - \nabla f(\boldsymbol{z}_2)\|}{\|\boldsymbol{z}_1 - \boldsymbol{z}_2\|} = \frac{\|\boldsymbol{H}(\boldsymbol{z}_1 - \boldsymbol{z}_2)\|}{\|\boldsymbol{z}_1 - \boldsymbol{z}_2\|} \leq \|\boldsymbol{H}\|_2,$$

where $\|\boldsymbol{H}\|_2$ is spectral norm (*i.e.,* maximum singular value) of $\boldsymbol{H}$. We would like to show that $\|\boldsymbol{H}\|_2 \leq L$. To this end, it is enough to verify the following two inequalities:

$$\det(L^2\boldsymbol{I} - \boldsymbol{H}\boldsymbol{H}^\top) = L^4 - (\mu_1^2 + \mu_2^2 + 2\ell^2)L^2 + (\mu_1\mu_2 + \ell^2)^2 \geq 0,$$
$$\text{trace}(\boldsymbol{H}\boldsymbol{H}^\top)/2 = (\mu_1^2 + \mu_2^2)/2 + \ell^2 \leq L^2.$$

This is because the characteristic polynomial of $\boldsymbol{H}\boldsymbol{H}^\top$, or $\det(\omega\boldsymbol{I} - \boldsymbol{H}\boldsymbol{H}^\top)$, is a quadratic polynomial of $\omega$, and its maximum root should not be greater than $L^2$. Let $\ell^2 = L^2 - \mu_1\mu_2 + a$ for some $a \in \mathbb{R}$. Plugging this $\ell^2$ into both inequalities above, we get

$$a^2 - (\mu_1 - \mu_2)^2 L^2 \geq 0 \quad \text{and} \quad a \leq -(\mu_1 - \mu_2)^2/2,$$

respectively. One can check that $a = -L|\mu_1 - \mu_2|$ is the largest possible $a$ satisfying both inequalities above. This proves the proposition. $\qquad\square$

Subsequently, we show that if our convergence rate is exponential, then the iteration complexity in terms of $\|\boldsymbol{z} - \boldsymbol{z}^*\|^2$ is equivalent to that in terms of $V_\lambda(\boldsymbol{z}) = \lambda[\Phi(\boldsymbol{x}) - \Phi^*] + [\Phi(\boldsymbol{x}) - f(\boldsymbol{z})]$ for PŁ($\Phi$)-PŁ problem, up to constant factors. This also applies to the function class $\mathcal{F}(L, \mu_1, \mu_2)$ since it is a subclass of smooth PŁ($\Phi$)-PŁ functions (∵ Propositions 7 and 10).

**Lemma 29.** *Suppose $f(\boldsymbol{x};\boldsymbol{y})$ is an $L$-smooth function satisfying $\boldsymbol{y}$-side $\mu_2$-PŁ condition and primal $\mu_1$-PŁ condition (i.e., PŁ($\Phi$)-PŁ). Suppose $\boldsymbol{z}^* = (\boldsymbol{x}^*;\boldsymbol{y}^*)$ is a global minimax point of $f$. Then, it satisfies*

$$\frac{\lambda\mu_1\mu_2^2}{2(\lambda\mu_1\mu_2 + 2L^2)} \|\boldsymbol{z} - \boldsymbol{z}^*\|^2 \leq V_\lambda(\boldsymbol{z}) \leq \frac{(\lambda + 1)L^3}{\mu_2^2} \|\boldsymbol{z} - \boldsymbol{z}^*\|^2.$$

We remark that the second inequality also holds for general smooth nonconvex-PŁ problems.

*Proof.* Let $\kappa_1 = L/\mu_1$ and $\kappa_2 = L/\mu_2$ be condition numbers. By the conditions of $f$ (smoothness and PŁ conditions), for any $\boldsymbol{x}$ and $\boldsymbol{y}$,

$$\frac{\mu_1}{2} \|\boldsymbol{x} - \boldsymbol{x}^*\|^2 \overset{\text{Prop. 10}}{\leq} \Phi(\boldsymbol{x}) - \Phi^* \overset{\text{Prop. 9}}{\leq} \frac{L(\kappa_2 + 1)}{2} \|\boldsymbol{x} - \boldsymbol{x}^*\|^2,$$

$$\frac{\mu_2}{2} \|\boldsymbol{y} - \boldsymbol{y}^*(\boldsymbol{x})\|^2 \overset{\text{Ass. 4}}{\leq} \Phi(\boldsymbol{x}) - f(\boldsymbol{x};\boldsymbol{y}) \overset{\text{Ass. 1}}{\leq} \frac{L}{2} \|\boldsymbol{y} - \boldsymbol{y}^*(\boldsymbol{x})\|^2,$$

where $\boldsymbol{y}^*(\boldsymbol{x})$ is a projection of $\boldsymbol{y}$ to $\arg\max_{\boldsymbol{y}'} f(\boldsymbol{x};\boldsymbol{y}')$. In particular, $\boldsymbol{y}^*(\boldsymbol{x}^*) = \boldsymbol{y}^*$. Since $\boldsymbol{y}^*(\boldsymbol{x})$ is a function of $\boldsymbol{x}$ and can differ from $\boldsymbol{y}^*$, we need to bound the term $\|\boldsymbol{y} - \boldsymbol{y}^*(\boldsymbol{x})\|^2$ using $\|\boldsymbol{x} - \boldsymbol{x}^*\|^2$ and $\|\boldsymbol{y} - \boldsymbol{y}^*\|^2$. To upper-bound the term $\|\boldsymbol{y} - \boldsymbol{y}^*(\boldsymbol{x})\|^2$, note that,

$$\|\boldsymbol{y} - \boldsymbol{y}^*(\boldsymbol{x})\|^2 \leq (\|\boldsymbol{y} - \boldsymbol{y}^*(\boldsymbol{x}^*)\| + \|\boldsymbol{y}^*(\boldsymbol{x}) - \boldsymbol{y}^*(\boldsymbol{x}^*)\|)^2$$
$$\leq (\|\boldsymbol{y} - \boldsymbol{y}^*\| + \kappa_2 \|\boldsymbol{x} - \boldsymbol{x}^*\|)^2$$
$$\leq (1 + \kappa_2^2) \left(\|\boldsymbol{y} - \boldsymbol{y}^*\|^2 + \|\boldsymbol{x} - \boldsymbol{x}^*\|^2\right).$$

The first inequality holds by triangle inequality, the second inequality holds by Proposition 8, and the last inequality holds by Cauchy-Schwarz inequality.[12] To lower-bound in a similar way, note that for any constant $a > 0$,

$$\|\boldsymbol{y} - \boldsymbol{y}^*\|^2 \leq (\|\boldsymbol{y} - \boldsymbol{y}^*(\boldsymbol{x})\| + \|\boldsymbol{y}^*(\boldsymbol{x}) - \boldsymbol{y}^*(\boldsymbol{x}^*)\|)^2$$
$$\leq \left(\|\boldsymbol{y} - \boldsymbol{y}^*(\boldsymbol{x})\| + \frac{\kappa_2}{\sqrt{a}} \cdot \sqrt{a} \|\boldsymbol{x} - \boldsymbol{x}^*\|\right)^2$$
$$\leq \left(1 + \frac{\kappa_2^2}{a}\right) \left(\|\boldsymbol{y} - \boldsymbol{y}^*(\boldsymbol{x})\|^2 + a \|\boldsymbol{x} - \boldsymbol{x}^*\|^2\right).$$
$$\therefore \|\boldsymbol{y} - \boldsymbol{y}^*(\boldsymbol{x})\|^2 \geq \frac{1}{1 + \kappa_2^2/a} \|\boldsymbol{y} - \boldsymbol{y}^*\|^2 - a \|\boldsymbol{x} - \boldsymbol{x}^*\|^2.$$

---

[12] $(ax + by)^2 \leq (a^2 + b^2)(x^2 + y^2)$ for real numbers $a, b, x, y$.

Now we can prove the inequalities in the lemma. We first show the second one. Applying $\kappa_2 \geq 1$ multiple times,

$$
\begin{aligned}
V_\lambda(\boldsymbol{x}; \boldsymbol{y}) &= \lambda[\Phi(\boldsymbol{x}) - \Phi^*] + [\Phi(\boldsymbol{x}) - f(\boldsymbol{z})] \\
&\leq \frac{\lambda L(\kappa_2 + 1)}{2} \|\boldsymbol{x} - \boldsymbol{x}^*\|^2 + \frac{L}{2} \|\boldsymbol{y} - \boldsymbol{y}^*(\boldsymbol{x})\|^2 \\
&\leq \left( \frac{\lambda L(\kappa_2 + 1)}{2} + \frac{L(1 + \kappa_2^2)}{2} \right) \|\boldsymbol{x} - \boldsymbol{x}^*\|^2 + \frac{L(1 + \kappa_2^2)}{2} \|\boldsymbol{y} - \boldsymbol{y}^*\|^2 \\
&\leq (\lambda + 1)L\kappa_2^2 \left( \|\boldsymbol{x} - \boldsymbol{x}^*\|^2 + \|\boldsymbol{y} - \boldsymbol{y}^*\|^2 \right) = \frac{(\lambda + 1)L^3}{\mu_2^2} \|\boldsymbol{z} - \boldsymbol{z}^*\|^2.
\end{aligned}
$$

To show the first inequality of the lemma, let $a = \frac{\lambda \mu_1}{2\mu_2}$.

$$
\begin{aligned}
V_\lambda(\boldsymbol{x}; \boldsymbol{y}) &\geq \frac{\lambda \mu_1}{2} \|\boldsymbol{x} - \boldsymbol{x}^*\|^2 + \frac{\mu_2}{2} \|\boldsymbol{y} - \boldsymbol{y}^*(\boldsymbol{x})\|^2 \\
&\geq \left( \frac{\lambda \mu_1}{2} - \frac{\mu_2 a}{2} \right) \|\boldsymbol{x} - \boldsymbol{x}^*\|^2 + \frac{\mu_2}{2(1 + \kappa_2^2/a)} \|\boldsymbol{y} - \boldsymbol{y}^*\|^2 \\
&\geq \frac{\lambda \mu_1}{4} \|\boldsymbol{x} - \boldsymbol{x}^*\|^2 + \frac{\lambda \mu_1}{4(a + \kappa_2^2)} \|\boldsymbol{y} - \boldsymbol{y}^*\|^2 \\
&\geq \frac{\lambda \mu_1}{4(a + \kappa_2^2)} \left( \|\boldsymbol{x} - \boldsymbol{x}^*\|^2 + \|\boldsymbol{y} - \boldsymbol{y}^*\|^2 \right) = \frac{\lambda \mu_1 \mu_2^2}{2(\lambda \mu_1 \mu_2 + 2L^2)} \|\boldsymbol{z} - \boldsymbol{z}^*\|^2.
\end{aligned}
$$

This concludes the proof. $\qquad\qquad\square$

The equivalence of iteration complexities for achieving $\|\boldsymbol{z}_K - \boldsymbol{z}^*\|^2 \leq \varepsilon^2$ or $V_\lambda(\boldsymbol{z}_K) \leq \varepsilon^2$ is quite straightforward from this lemma, as long as the convergence speed is exponential. For example, suppose we have a upper convergence bound $\|\boldsymbol{z}_K - \boldsymbol{z}^*\|^2 \leq a \exp(-K/r)$ for some constants $a, r > 0$. This implies a upper iteration complexity bound $K = \mathcal{O}(r \log(1/\varepsilon))$ sufficient to achieve $\|\boldsymbol{z}_K - \boldsymbol{z}^*\|^2 \leq \varepsilon^2$. Then by Lemma 29, we also have $V_\lambda(\boldsymbol{z}_K)^2 \leq a' \exp(-K/r)$ where $a' = a(\lambda + 1)L^3/\mu_2^2$ is also a constant. This implies a lower iteration complexity bound $K = \mathcal{O}(r \log(1/\varepsilon))$ as well, sufficient to achieve $V_\lambda(\boldsymbol{z}_K)^2 \leq \varepsilon^2$. The other way of complexity translation operates with a similar logic.

## F    REMARK ON SMOOTHNESS ASSUMPTIONS AND LOWER BOUND OF WITH-REPLACEMENT SGD(A)

During the discussion phase of the conference, a reviewer raised a question about whether or not the *component smoothness* (Assumption 1) is more crucial than the without-replacement component sampling for faster convergence. However, we would like to claim that the component smoothness alone is not sufficient for improving the convergence rate for with-replacement SGD(A). To this end, we provide some formal results on lower convergence bounds. For simplicity, we use mini-batches of size 1 throughout this appendix.

Firstly, the theorem below provides a lower bound on with-replacement SGD for minimization problems. Readers can also verify that an analogous lower bound holds for SGD with unbiased and independently sampled gradient oracle for more general stochastic minimization problems. The proof will appear later in this appendix.

**Theorem 30.** *For any step size $\eta > 0$, there exists a real-valued strongly-convex function $f(\boldsymbol{x})$ defined on $\mathbb{R}^d$ with $f^* := \min_{\boldsymbol{x}} f(\boldsymbol{x})$, satisfying:*

1. *$f$ consists of $n > 1$ smooth component functions $f_i$: $f(\boldsymbol{x}) = \frac{1}{n} \sum_{i=1}^n f_i(\boldsymbol{x})$, where each component $f_i$ is smooth;*

2. *After running $T > 1$ iterations of with-replacement SGD (with mini-batch size 1) starting from $x_0 \in \mathbb{R}^d$, the last iterate $x_T$ satisfies $\mathbb{E}[f(\boldsymbol{x}_T) - f^*] \geq \Omega(1/T)$, where the expectation is taken with respect to the randomness of i.i.d. index choice at each iteration.*

Next, we show this theorem naturally induces a convergence lower bound for the minimax counterpart: *with-replacement SGDA*. Consider a (finite-sum) minimax problem $\min_x \max_y g(\boldsymbol{x}, \boldsymbol{y}) := f(\boldsymbol{x}) - f(\boldsymbol{y})$, where $f = \frac{1}{n} \sum_{i=1}^{n} f_i$ is a worst-case function in the proof of Theorem 30. Here, the minimax problem on $g$ can be solved by minimizing $f$. Moreover, since the primal function $\Phi(\boldsymbol{x}) := \max_y g(\boldsymbol{x}, \boldsymbol{y})$ associated with $g$ is in fact the same as $f(\boldsymbol{x}) - f^*$, the potential function $V_\lambda(\boldsymbol{x}, \boldsymbol{y}) := \lambda[\Phi(\boldsymbol{x}) - (\min_x \Phi(\boldsymbol{x}))] + [\Phi(\boldsymbol{x}) - g(\boldsymbol{x}, \boldsymbol{y})]$ becomes the same as $\lambda(f(\boldsymbol{x}) - f^*) + (f(\boldsymbol{y}) - f^*)$ for a constant $\lambda > 0$. Combining these facts, we can immediately obtain the following lower convergence bound of with-replacement SGDA.

**Corollary 1.** *There exists a strongly-convex-strongly-concave function* $g(\boldsymbol{x}, \boldsymbol{y}) := \frac{1}{n} \sum_{i=1}^{n} g_i(\boldsymbol{x}, \boldsymbol{y})$ *consisting of* $n$ *smooth component functions* $g_i$, *where the last iterate* $(\boldsymbol{x}_T, \boldsymbol{y}_T)$ *of with-replacement SGDA satisfies* $\mathbb{E}[V(\boldsymbol{x}_T, \boldsymbol{y}_T)] \geq \Omega(1/T)$.

Corollary 1 formally proves that with-replacement SGDA on strongly-convex-strongly-concave minimax problems with smooth components has a worst-case convergence rate $\Omega(1/T)$. This in fact matches the $\mathcal{O}(1/T)$ upper bound obtained for primal-PŁ-PŁ problems by Yang et al. (2020). Considering that strongly-convex-strongly-concave functions form a strict subset of primal-PŁ-PŁ functions, Corollary 1 establishes that adding component smoothness assumption does not provide further speed up for with-replacement SGDA.

In contrast, our theoretical result in Theorem 2 shows that SGDA-RR has a much faster convergence rate $\mathbb{E}[V_\lambda] \leq \tilde{\mathcal{O}}(\frac{1}{nK^2})$ for primal-PŁ-PŁ minimax problems, where $K$ is the number of epochs. One can check that our $\tilde{\mathcal{O}}(\frac{1}{nK^2})$ bound is faster than the tight convergence rate $\Theta(1/T)$ of with-replacement SGDA by simply plugging in $T = nK$. In light of Corollary 1 we proved, we can now claim that the improvement can be solely attributed to RR.

Although we do not provide a lower bound for more general nonconvex-PŁ problems here, we believe the more challenging case of nonconvex-PŁ lower bound is a topic for another separate paper. Nonetheless, we conjecture that the speed up by SGDA-RR in nonconvex-PŁ settings is also due to the effect of RR, not component smoothness.

From now on, we provide the postponed proof of Theorem 30.

*Proof of Theorem 30.* We construct worst-case functions with quadratic functions on $\mathbb{R}$, which are clearly $L$-smooth for a fixed constant $L > 0$. Then, it is easy to extend the logic to the functions with domains of higher dimensions. Let $x_0 \in \mathbb{R}$ be the initial iterate.

**Case 1** ($\frac{1}{LT} \leq \eta \leq (\frac{2}{L} - \frac{1}{LT})$). Note that the condition on the step size, $\frac{1}{LT} \leq \eta \leq (\frac{2}{L} - \frac{1}{LT})$, is equivalent to an inequality $(1 - \eta L)^2 \leq (1 - 1/T)^2$.

We first assume $n$ is an even number. We will encounter the case with an odd $n > 1$ a bit later. Consider $f(x) = \frac{L}{2} x^2$ consisting of even number of components $f_i$'s defined by

$$f_i(x) = \begin{cases} \frac{L}{2} x^2 + \nu x, & (i \leq \frac{n}{2}), \\ \frac{L}{2} x^2 - \nu x, & (i \geq \frac{n}{2} + 1), \end{cases}$$

for some number $\nu \in \mathbb{R}$. At each iteration $t \geq 1$, we choose a component index $i(t) \overset{\text{i.i.d.}}{\sim} \text{Unif}([n])$ (with-replacement sampling). Then we can write the chosen component function at iteration $t$ as $f_i(t) = \frac{L}{2} x^2 - s_t \nu x$ for some i.i.d. random variable $s_t \sim \text{Unif}(\{\pm 1\})$. Accordingly, an SGD step can be written as
$$x_t = x_{t-1} - \eta \nabla f_{i(t)}(x_{t-1}) = (1 - \eta L) x_{t-1} + \eta s_t \nu.$$
By applying telescopic sum, we have

$$x_T = (1 - \eta L)^T x_0 + \eta \nu \sum_{t=1}^{T} (1 - \eta L)^{(T-t)} \cdot s_t.$$

Taking squares and expectations (with respect to the random variables $s_1, \dots, s_T$) to both sides, we have

$$\mathbb{E}[x_T^2] = (1 - \eta L)^{2T} x_0^2 + \eta^2 \nu^2 \sum_{t=1}^{T} (1 - \eta L)^{2(T-t)},$$

by applying the fact that $s_t$'s are zero-mean independent random variables with absolute values 1:

$$\mathbb{E}[s_t \cdot s_{t'}] = \begin{cases} 0, & t \neq t' \quad (\because \text{ independent}), \\ 1, & t = t' \quad (\because s_t^2 = 1). \end{cases}$$

We calculate the sum above as follows: since $(1 - \eta L)^2 \leq (1 - 1/T)^2$ and $(1 - 1/T)^T \leq e^{-1}$,

$$\sum_{t=1}^{T} (1 - \eta L)^{2(T-t)} = \frac{1 - (1 - \eta L)^{2T}}{1 - (1 - \eta L)^2} \geq \frac{1 - (1 - \frac{1}{T})^{2T}}{2\eta L(1 - \frac{\eta L}{2})} \geq \frac{1 - e^{-2}}{2\eta L}.$$

With this inequality, and since $(1 - \eta L)^{2T} x_0^2 \geq 0$, we can lower-bound the expectation $\mathbb{E}[x_T^2]$:

$$\mathbb{E}[x_T^2] \geq \eta^2 \nu^2 \cdot \frac{1 - e^{-2}}{2\eta L} = \frac{(1 - e^{-2})\nu^2}{2L}\eta \geq \frac{(1 - e^{-2})\nu^2}{2L^2 T}.$$

Since $f$ has a minimum $f^* = 0$ at $x = 0$, we eventually have

$$\mathbb{E}[f(x_T) - f^*] = \frac{L}{2}\mathbb{E}[x_T^2] \geq \frac{(1 - e^{-2})\nu^2}{4LT} = \Omega\left(\frac{\nu^2}{LT}\right).$$

Now we consider the case when the number of components $n > 1$ is odd. Consider $f_n(x) \equiv 0$ and let the remaining $n - 1$ components be the same as the case above (with an even number of components). Note that the zero-component $f_n$ does not affect the trajectory of SGD (*i.e.*, the points visited by SGD) and the optimality of $f$ ($f^* = 0$ at $x = 0$), while the whole objective function becomes $f(x) = \frac{n-1}{n} \cdot \frac{L}{2}x^2$. Thus, it can be easily shown that the $\Omega\left(\frac{\nu^2}{LT}\right)$ lower bound also holds.

**Case 2** ($0 < \eta < \frac{1}{LT}$ or $\eta > \left(\frac{2}{L} - \frac{1}{LT}\right)$). From the condition on the step size, we have $(1 - \eta L)^2 > (1 - 1/T)^2$. Consider $f_i(x) = \frac{L}{2}x^2$ for every $i \in [n]$: every components are the same. In this case, we show that the last iterate of SGD is bounded below by a constant with respect to $T > 1$.

At each iteration $t \geq 1$, we obtain $x_t = (1 - \eta L)x_{t-1}$ by a step of SGD. Then, applying $T \geq 2$,

$$x_T^2 = (1 - \eta L)^{2T} \cdot x_0^2 > \left(1 - \frac{1}{T}\right)^{2T} x_0^2 \geq \left(1 - \frac{1}{2}\right)^4 x_0^2 = \frac{x_0^2}{16}.$$

Since $f(x) = \frac{1}{n}\sum_{i=1}^{n} f_i(\boldsymbol{x}) = \frac{L}{2}x^2$ has a minimum $f^* = 0$ at $x = 0$, we have

$$f(x_T) - f^* > \frac{Lx_0^2}{32} = \Omega(1) \cdot Lx_0^2.$$

$\square$

# G  EXPERIMENTS: QUADRATIC GAMES

In this appendix, we provide a more detailed illustration of our numerical evaluations on quadratic games introduced in Section 6. Recall that the objective function $f$ and its component functions $f_i$ are given in Equation (5) as

$$f(\boldsymbol{x}; \boldsymbol{y}) = \frac{1}{2}\boldsymbol{x}^\top \boldsymbol{A}\boldsymbol{x} + \boldsymbol{x}^\top \boldsymbol{B}\boldsymbol{y} - \frac{1}{2}\boldsymbol{y}^\top \boldsymbol{C}\boldsymbol{y},$$

$$f_i(\boldsymbol{x}; \boldsymbol{y}) = \frac{1}{2}\boldsymbol{x}^\top \boldsymbol{A}_i\boldsymbol{x} + \boldsymbol{x}^\top \boldsymbol{B}_i\boldsymbol{y} - \frac{1}{2}\boldsymbol{y}^\top \boldsymbol{C}_i\boldsymbol{y} + \boldsymbol{u}_i^\top \boldsymbol{x} - \boldsymbol{v}_i^\top \boldsymbol{y}.$$

We choose the same dimensions for the variables $\boldsymbol{x} \in \mathbb{R}^{d_x}$ and $\boldsymbol{y} \in \mathbb{R}^{d_y}$: we set $d_x = d_y = d$.

## G.1  PARAMETER CHOICES

To sample the matrix $\boldsymbol{C} = \frac{1}{n}\sum_{i=1}^{n} \boldsymbol{C}_i \in \mathbb{R}^d$ satisfying that $\mu_C \boldsymbol{I}_d \preceq \boldsymbol{C}$ and $\|\boldsymbol{C}_i\|_2 \leq L_C$, we first randomly generate an orthogonal matrix $\boldsymbol{Q}_C \in \mathbb{R}^{d \times d}$ (*i.e.*, $\boldsymbol{Q}_C \boldsymbol{Q}_C^\top = \boldsymbol{I}_d$), by taking advantage of

the QR-decomposition of a random matrix. Then, we generate the eigenvalues of $C_i$'s as follows. We sample the entries of $n$ vectors $\boldsymbol{\lambda}_i^C \in \mathbb{R}^d$ ($i \in [n]$) uniformly from the interval $[\mu_C, L_C]$. We add some level of perturbations to some entries of each $\boldsymbol{\lambda}_i^C$; we replace some entries to the numbers in an interval $[-L_C, \mu_C]$, keeping the entries of the vector $\frac{1}{n}\sum_{i=1}^n \boldsymbol{\lambda}_i^C$ in the interval $[\mu_C, L_C]$. Finally, we define $C_i = Q_C \Lambda_i^C Q_C^\top$ where $\Lambda_i^C = \mathrm{diag}(\boldsymbol{\lambda}_i^C)$. Because of the perturbation step, some $C_i$'s are not positive definite, thereby some components $f_i$'s become non-(strongly-)concave in $\boldsymbol{y}$.

Next, we sample the matrix $B_i$'s. There are no requirements for $B$ but $\|B_i\|_2 \le L_B$; $B_i$'s are even not necessarily symmetric when $d_x \ne d_y$. Thus, we first generate the orthogonal matrices $U_i^B$ and $V_i^B$ by taking advantage of the singular value decomposition of random matrices. Then, we generate the singular values of $B_i$'s by sampling the entries of $n$ vectors $\boldsymbol{\sigma}_i^B$ uniformly from the interval $[0, L_C]$. After that, we define $B_i = U_i^B \Sigma_i^B V_i^B$ where $\Sigma_i^C = \mathrm{diag}(\boldsymbol{\sigma}_i^C)$. We typically want to take a larger $L_B$ than $L_C$ to strengthen the interaction term $\boldsymbol{x}^\top B \boldsymbol{y}$.

Recall that the primal function $\Phi$ associated with $f$ is explicitly written as

$$\Phi(\boldsymbol{x}) = \max_{\boldsymbol{y} \in \mathbb{R}^d} f(\boldsymbol{x}; \boldsymbol{y}) = \frac{1}{2}\boldsymbol{x}^\top \left(A + BC^{-1}B^\top\right)\boldsymbol{x} := \frac{1}{2}\boldsymbol{x}^\top M \boldsymbol{x}. \tag{28}$$

Note that the inverse of $C$ can be efficiently computed as $C^{-1} = Q_C (\Lambda^C)^{-1} Q_C^\top$.

Before generating the matrices $A_i$'s, we first generate $M_i$'s satisfying that $\frac{1}{n}\sum_{i=1}^n M_i = M$ and the nonzero eigenvalues of positive *semi*definite $M$ are in the interval $[\mu_M, L_M]$. The process of sampling $M_i$'s is almost identical to how to sample $C_i$'s. One notable difference is, $M_i$'s and $M$ are forced to have $r(< d)$ zero eigenvalues: this makes $M$ a positive semidefinite (but not strictly positive definite) matrix of rank $d - r$. Moreover, we get the $\mu_M$-PŁ($\Phi$) condition in $\boldsymbol{x}$ as follows:

**Proposition 31.** *Consider a positive semidefinite matrix $M \in \mathbb{R}^d$. If the smallest nonzero eigenvalue of $M$ is $\mu$, then $\Phi(\boldsymbol{x}) := \frac{1}{2}\boldsymbol{x}^\top M \boldsymbol{x}$ is $\mu$-PŁ in $\boldsymbol{x}$. Also, $\Phi^* = \min_{\boldsymbol{x}} \Phi(\boldsymbol{x}) = 0$.*

*Proof.* Apply the eigendecomposition of $M$: $M = Q\Lambda Q^\top$. Let $\overline{M} = \Lambda^{1/2}Q^\top$. Then, we have $\Phi(\boldsymbol{x}) = \frac{1}{2}\|\overline{M}\boldsymbol{x}\|^2$, which implies that $\Phi(\boldsymbol{x}) \ge 0$ ($\forall \boldsymbol{x}$) and in fact $\Phi^* = 0$. Note that $\frac{1}{2}\|\boldsymbol{x}\|^2$ is 1-strongly convex. Also, the minimum nonzero singular value of $\overline{M}$ is $\sqrt{\mu}$ ($\because M = \overline{M}^\top \overline{M}$). Therefore, by the proof of Proposition 12, $\Phi(\boldsymbol{x})$ is a $\mu$-PŁ function of $\boldsymbol{x}$. Lastly, we note that $\Phi(\boldsymbol{x})$ is not strongly convex in general, especially when $M$ is a rank-deficient matrix. $\square$

Typically, the spectral norm $\|M\|_2$ is known to be bounded above by $\|A\|_2 + L_B^2/\mu_C$ in *worst-case* (Nouiehed et al., 2019; Li et al., 2022). However, since we sample $M$ without knowing the exact form of $A_i$'s while we want to control the spectral norm $\|A_i\|_2$ not too large (for smoothness of $f_i$), we (empirically) decide to choose rather smaller $L_M$: simply, we choose $L_M = L_B$.

Now we let $A_i = M_i - BC^{-1}B^\top$ and $A = \frac{1}{n}\sum_{i=1}^n A_i$ to satisfy Equation (28). We emphasize that $A$ may have negative eigenvalues; the objective is nonconvex in $\boldsymbol{x}$ in general. We have checked this is true across the experimental settings. Also, we let $L := \max\{\|A\|_2, L_B, L_C\}$ for further parameter selection. (In fact, because of our choice of parameter values, $L$ was always equal to $L_B$ in our experiments.)

Furthermore, we generate the vectors $\boldsymbol{u}_i$'s and $\boldsymbol{v}_i$'s satisfying $\sum_{i=1}^n \boldsymbol{u}_i = \boldsymbol{0} = \sum_{i=1}^n \boldsymbol{v}_i$. The entries of these vectors are uniformly sampled from an interval $[-\Delta, \Delta]$, thereby the average of entries is centered to zero. In addition, to verify our theory, we choose the step-sizes of the form $\beta = c_1 \cdot b/nL$ and $\alpha = c_0 \cdot \beta/\kappa_2^2$ for some constants $c_0$ and $c_1$ and batch size $b$.

Lastly, we specify the values of parameters described above: $n = 100$, $d = 25$, $\mu_M = \mu_C$, and $L_C = 1 < L_M = L_B$. The constants $c_0$ and $c_1$ are tuned among $10^{\{-2, -1.5, \pm 1, \pm 0.5, 0\}}$. In the following subsections, we investigate the effects of the change of

 (i) $\Delta \in \{10, \boldsymbol{20}, 40\}$, determining the discrepancy between components,

 (ii) condition number $\kappa_2 \in \{5, \boldsymbol{10}, 20\}$, determined by $L_B$ and $\mu_C$, and

 (iii) batch size $b \in \{\boldsymbol{1}, 25, 50, 100\}$,

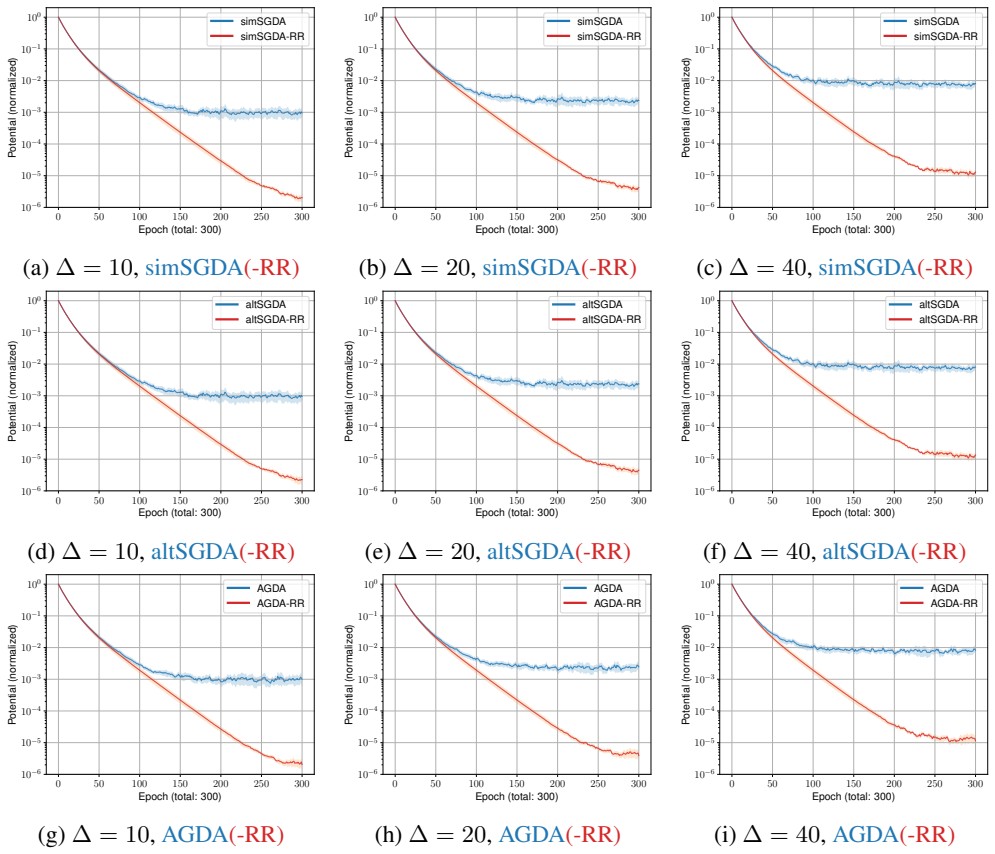

(a) $\Delta = 10$, simSGDA(-RR)    (b) $\Delta = 20$, simSGDA(-RR)    (c) $\Delta = 40$, simSGDA(-RR)

(d) $\Delta = 10$, altSGDA(-RR)    (e) $\Delta = 20$, altSGDA(-RR)    (f) $\Delta = 40$, altSGDA(-RR)

(g) $\Delta = 10$, AGDA(-RR)    (h) $\Delta = 20$, AGDA(-RR)    (i) $\Delta = 40$, AGDA(-RR)

Figure 2: Comparisons by changing the value of $\Delta \in \{10, 20, 40\}$. Solid lines: average across 10 different runs. Shaded regions: 95% confidence intervals ($\pm 1.96$ std). The vertical axes are on a *logarithmic scale*.

from the plots of the values potential function $V_\lambda(\boldsymbol{x}; \boldsymbol{y}) = (1 + \lambda)\Phi(\boldsymbol{x}) - f(\boldsymbol{x}; \boldsymbol{y})$ over *epochs*.[13] (Numbers in bold font above are the default values of parameters.)

### G.2 COMPARISON: THE EFFECT OF COMPONENT DISCREPANCY

Notice that the discrepancy between component functions gets larger as $\Delta$ grows. Technically, one can check that the gradient variance (that we controlled in Assumption 2) is proportional to the norms of the vectors $\boldsymbol{u}_i$ and $\boldsymbol{v}_i$. Moreover, we have already discussed that the gap between convergence speeds of SGDA and SGDA-RR becomes larger especially when the gradient variance is large.

Now, we present the results of numerical experiments by varying the values of $\Delta$ to 10, 20, and 40, while fixing $L_B = 4$, $\mu_C = 0.4$, $b = 1$, and other experiment parameters. As shown in Figure 2, we can observe that the difference between the random-reshuffling algorithm and the uniform-sampling algorithm gets larger as $\Delta$ increases.

### G.3 COMPARISON: THE EFFECT OF CONDITION NUMBER

Here, we present the results of experiments by varying the values of $\kappa_2$ to 5, 10, and 20, while fixing $\Delta = 20$, $b = 1$, and other experiment parameters. To this end, we applied the parameter settings for $L_B$ and $\mu_C$ as $(L_B, \mu_C) = (2.5, 0.5), (4, 0.4), (5, 0.25)$, respectively.

---

[13]During and after the discussion phase, we performed some more experiments. As we tried to plot all the results over *iterations*, the size of the figures in `pdf` format became too large. Consequently, in this appendix, we only plot the results over epochs to reduce the file size of the figures.

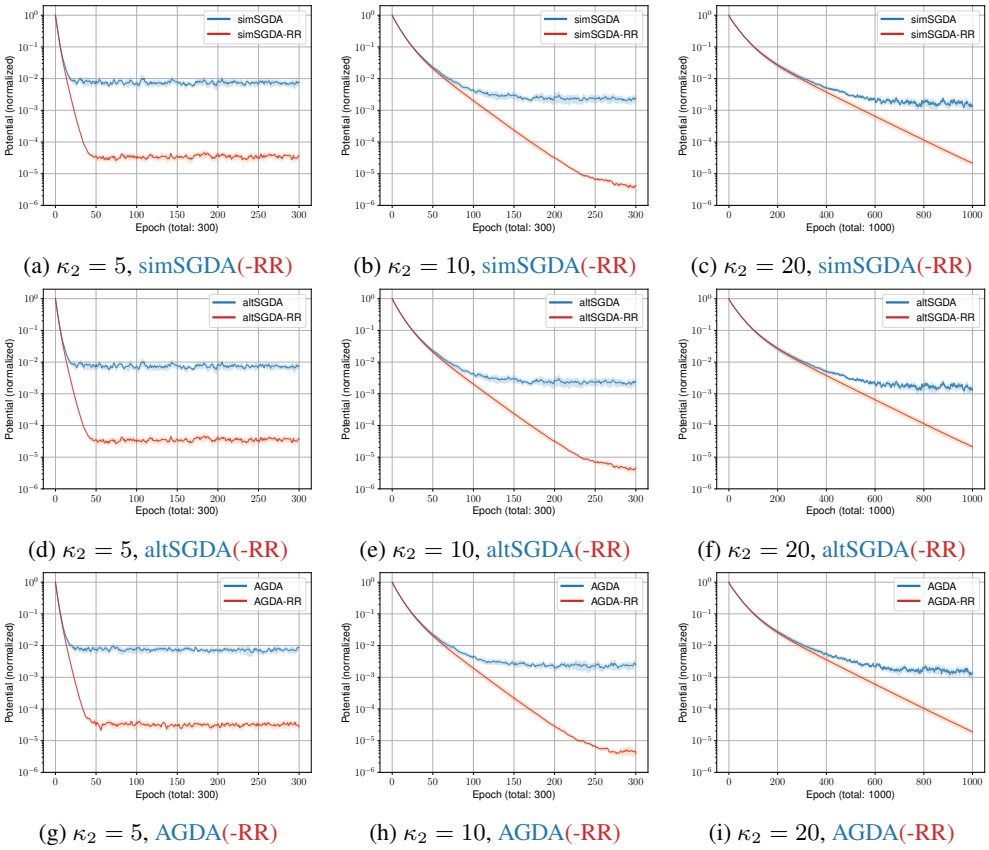

(a) $\kappa_2 = 5$, simSGDA(-RR)     (b) $\kappa_2 = 10$, simSGDA(-RR)     (c) $\kappa_2 = 20$, simSGDA(-RR)

(d) $\kappa_2 = 5$, altSGDA(-RR)     (e) $\kappa_2 = 10$, altSGDA(-RR)     (f) $\kappa_2 = 20$, altSGDA(-RR)

(g) $\kappa_2 = 5$, AGDA(-RR)     (h) $\kappa_2 = 10$, AGDA(-RR)     (i) $\kappa_2 = 20$, AGDA(-RR)

Figure 3: Comparisons by changing the value of $\kappa_2 = L/\mu_C \in \{5, 10, 20\}$. Solid lines: average across 10 different runs. Shaded regions: 95% confidence intervals ($\pm 1.96$ std). The vertical axes are on a *logarithmic scale*. Note: we run **1000 epochs for $\kappa_2 = 20$** (see the rightmost column), whereas we run **300 epochs for the other** $\kappa_2 \in \{5, 10\}$ (see the leftmost & middle columns).

The results are shown in Figure 3. We observe that more epochs are required for convergence when $\kappa_2$ increases, regardless of the type of algorithm. One may think that the performance gap between RR-based/non-RR-based algorithms is small when $\kappa_2$ is huge. However, when we run the algorithm for an extended number of epochs, we observe a significant gap in convergence speeds.

### G.4 COMPARISON: THE EFFECT OF BATCH SIZE

The last comparison is about the effect of batch size $b \in \{1, 25, 50, 100\}$. Recall that we linearly scale the step sizes as the batch size changes. However, since the number of epochs is fixed, the number of iterations decreases as $b$ gets larger.

As the readers can notice, the convergence behavior of SGDA (resp., SGDA-RR) and AGDA (resp., AGDA-RR) are similar in our construction of quadratic games. Thus, in this subsection, we only compare simSGDA and its variants. Rather, we introduce two more methods of component choice other than with-replacement uniform sampling and random reshuffling:

- WORB(WithOut-Replacement mini-Batching): every mini-batch is without-replacement & uniformly-randomly sampled, while any pair of mini-batches in an epoch may have some indices in common; the same as *b-minibatch sampling* (Loizou et al., 2021).

- NS(No Shuffle): accessing $1, ..., n$ in its predefined order to construct mini-batches; without-replacement but deterministic. Remark: for *minimization* problems, SGD with NS is usually referred to as *incremental gradient* (IG) algorithm (Mishchenko et al., 2020).

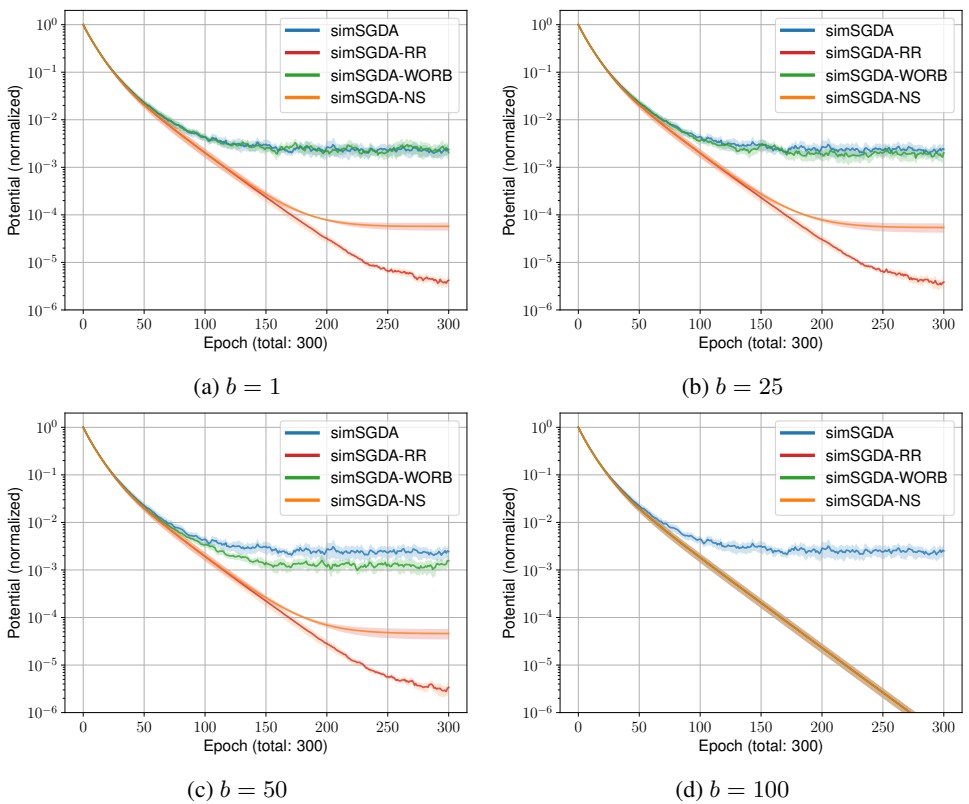

Figure 4: Comparisons of simSGDA(-RR,-WORB,-NS) as changing $b \in \{1, 25, 50, 100\}$. Solid lines: average across 10 different runs. Shaded regions: 95% confidence intervals ($\pm 1.96$ std). The vertical axes are on a *logarithmic scale*.

These two methods are somewhat related to without-replacement component sampling, whereas they are both different from RR which uniformly randomly samples a permutation of $[n]$ every epoch. We call simSGDA using mini-batches sampled by WORB and NS as *simSGDA-WORB* and *simSGDA-NS*, respectively. Remarks: If $b = 1$, simSGDA-WORB becomes the same algorithm as vanilla simSGDA. Also, since we choose $n = 100$, if $b = n = 100$, all three algorithms simSGDA-RR/-WORB/-NS become the same as deterministic & full-batch (simultaneous) GDA.

The results are shown in Figure 4. One can notice that the potential plots of simSGDA, simSGDA-RR, and simSGDA-NS are respectively the same even if we change the batch size ($b < 100$). Also, if $b > 1$, simSGDA-WORB has better performance than vanilla simSGDA. These imply that without-replacement mini-batches benefit the convergence speed to some extent in our quadratic game. However, the result of experiments also implies that both (i) without-replacement *per epoch* (*i.e.*, shuffling) and (ii) randomization are indeed essential for fast convergence in our quadratic game experiments. In particular, WORB requires a very large batch size but still has a much slower convergence rate than RR (see Figure 4c which is the case of using half of the total components at each iteration).

## H  OMITTED COMPARISON WITH RELATED WORKS

### H.1  COMPARISON WITH XIE ET AL. (2021)

To specialize Xie et al. (2021, Theorem 3) to the single-machine setup and discuss their results in terms of our notation, we need to replace their symbols

$$(T, S, K, \sigma_1^2, \sigma_2^2, G_1^2, G_2^2, L_{12}, L_f, \mu, L_\Phi, \mathcal{L}_0, \eta_t, \gamma_t)$$

with the following symbols from our notation

$$(K, 1, n, 0, 0, B, B, L, L, \mu_2, L(\kappa_2 + 1), V_\lambda(\boldsymbol{z}_0^1), \alpha, \beta),$$

and also put $A = 0$ (their analysis only applies uniformly bounded component variance per machine). Then we can *naively* translate the bound of Xie et al. (2021, Theorem 3) to our language as

$$\min_{k \in [K]} \mathbb{E}\left[\|\Phi(\boldsymbol{x})\|^2\right] \stackrel{(?)}{\leq} \mathcal{O}\left(\frac{\kappa_2 LV_\lambda(\boldsymbol{z}_0^1)}{K} + \kappa_2^2 \left(\frac{L^2 BV_\lambda(\boldsymbol{z}_0^1)^2}{K^2}\right)^{1/3}\right).$$

To the best of our knowledge, however, we believe there may be a mistake in the proof of Xie et al. (2021, Appendix C.4). From the inequalities on the last page of their paper, we notice that the term $\frac{40L_{12}^2\mathcal{L}_0}{\mu^2\gamma KT}$ might be missing in a step, where $\gamma$ is chosen to be the minimum of several terms including $\frac{1}{87L_f K}$. Thus, as far as we can tell, it seems inevitable that this omitted term would lead to an additional term $\frac{3480L_f L_{12}^2\mathcal{L}_0}{\mu^2 T}$ in the final bound. By combining this to their bound and re-translating it, we eventually have

$$\min_{k \in [K]} \mathbb{E}\left[\|\Phi(\boldsymbol{x})\|^2\right] \leq \mathcal{O}\left(\frac{\kappa_2^2 LV_\lambda(\boldsymbol{z}_0^1)}{K} + \kappa_2^2 \left(\frac{L^2 BV_\lambda(\boldsymbol{z}_0^1)^2}{K^2}\right)^{1/3}\right),$$

since their $L_{12}^2/\mu^2$ translates to our $\kappa_2^2$. Therefore, their result actually shows the same dependency on condition number $\kappa_2$ as our Theorem 1. Nevertheless, comparing the terms related to the component-wise variance $B$, ours is better. In the second term in the bound above does not shrink even when the number of iterations (per machine & per communication) grows. In our case (Theorem 1), however, the dominant term (in $K$) can be briefly written as $\mathcal{O}\left(\left(\frac{B}{nK^2}\right)^{1/3}\right)$ which can diminish with large $n$, *i.e.*, the number of iterations per epoch.