# OpenReview forum: "SGDA with shuffling: faster convergence for nonconvex-PŁ minimax optimization"
_ICLR.cc/2023/Conference — ICLR 2023 poster_

### Official Review · Reviewer_CLz9 · 2022-10-16

**Confidence:** 4
**Clarity, Quality, Novelty And Reproducibility:** The submission provides good clarity,…
**Correctness:** 4
**Technical Novelty And Significance:** 3
**Empirical Novelty And Significance:** Not applicable
**Recommendation:** 6

**Strength And Weaknesses:**

Strength:

1. The analysis of sampling without replacement for minimax optimization is indeed rare while it is the practical implementation in many cases. Therefore, this work fills a gap here.

2. The complexity results and lower bound results are all solid contributions. They are state-of-the-art results for the considered problems in a sense that they match their minimization problem counterparts in the literature.

Weakness:

1. It lacks some empirical studies to verify the theory. I would not doubt that the strategy of sampling without replacement can show its effectiveness in many cases, because there are extensive studies of it in minimization problems already. However, some empirical studies are still expected here in order to make the argument whole and convincing.

2. It needs to be justified that what applications can satisfy a nonconvex-PL landscape or a primal-PL-PL landscape. Would those problems be of enough interest to the community?




**Summary Of The Paper:**

This paper studies the SGDA algorithm with shuffling and sampling without replacement for non convex-PL minimax optimization problem. The complexity results show significant improvement over exiting analysis that use sampling with replacement.

**Summary Of The Review:**

I vote to accept this submission based on its significance and novelty. However, there are indeed some flaws as I mentioned above.

---

> ### Author Response · Authors · 2022-11-15
> **Response to Reviewer CLz9**
>
> We thank the reviewer for the valuable comments. We provide our response to each of the reviewer's concerns as follows.
>
>
> 1. "It lacks some empirical studies to verify the theory."
>
> - To address the issue raised by the reviewer, we have executed numerical experiments to support our theoretical findings. In the revision of our paper, we replaced Section 6 (proof sketch) with a section on empirical studies. Also, the reviewer can see Appendix G of the updated manuscript to see a more detailed description of our experiment design.
>
>
> 2."It needs to be justified that what applications can satisfy a nonconvex-PL landscape or a primal-PL-PL landscape. Would those problems be of enough interest to the community?"
>
> - We first introduce some examples of PL problems. One of the most simple examples is the function $g(Ax+b)$, where g is a strongly convex (SC) function (Karimi et al., 2016); e.g., $||Ax+b||^2$ when $g(x)=||x||^2$. Because of this, the undetermined linear regression is a prime example of a convex (but not strongly convex) and PL problem. In fact, many (non-strongly-)convex problems can be regarded as PL problems at least on a subset of the domain. Of course, there are more studies on the relevance between (deep) neural networks and the PL condition. For instance, the square loss on top of a deep linear neural network has been well-known to be nonconvex but satisfies the PL conditions on a large subset of the domain (Charles and Papailiopoulos, 2018). Recent work shows that a general over-parametrized neural network satisfies a generalized PL condition on most of the parameter space (Liu et al., 2022). These examples imply that we could find nonconvex-nonconcave minimax problems (at least locally) satisfying PL condition(s) in several applications. As an example, Liu et al. (2020) analyze AUC maximization (which is a popular example of a minimax problem) using deep neural networks by assuming a certain form of the primal-PL condition. For more simple examples, please refer to the papers by Nouiehed et al. (2019) and Yang et al. (2020, 2022).
> - As a side note, we would like to emphasize that the PL condition is a nontrivial and nonconvex extension of strong convexity (SC). This implies that previous theoretical results on SC-SC, primal-SC-SC, and even nonconvex-SC settings are naturally subsumed by the nonconvex-PL setting that we considered in our paper. We would like to note that one of the strengths of our work is that the analysis was conducted in a more general setting than the existing results.
>
>
> Thank you again for your feedback. Please let us know if you have any other comments.
>
> Best regards,
>
> Authors.
>
> ***
>
> References
>
> Charles, Zachary, and Dimitris Papailiopoulos. "Stability and generalization of learning algorithms that converge to global optima." International Conference on Machine Learning. PMLR, 2018.
> Karimi, Hamed, Julie Nutini, and Mark Schmidt. "Linear convergence of gradient and proximal-gradient methods under the Polyak-{\L}ojasiewicz condition." Joint European conference on machine learning and knowledge discovery in databases. Springer, Cham, 2016.
> Liu, Chaoyue, Libin Zhu, and Mikhail Belkin. "Loss landscapes and optimization in over-parameterized non-linear systems and neural networks." Applied and Computational Harmonic Analysis 59 (2022): 85-116.
> Nouiehed, Maher, et al. "Solving a class of non-convex min-max games using iterative first order methods." Advances in Neural Information Processing Systems 32 (2019).
> Yang, Junchi, Negar Kiyavash, and Niao He. "Global convergence and variance reduction for a class of nonconvex-nonconcave minimax problems." Advances in Neural Information Processing Systems 33. 2020.
> Yang, Junchi, et al. "Faster single-loop algorithms for minimax optimization without strong concavity." AISTATS 2022.

---

### Official Review · Reviewer_9Nrg · 2022-10-25

**Confidence:** 4
**Correctness:** 4
**Technical Novelty And Significance:** 2
**Empirical Novelty And Significance:** Not applicable
**Recommendation:** 6

**Clarity, Quality, Novelty And Reproducibility:**

The main problem considered in the paper is relatively new. The main technique is already well-studied, and authors extended the study into more general cases, which reveals novelty.

The paper is well-written and easy to follow.

**Strength And Weaknesses:**

Strength:
1. The writting of the paper is pretty good, the flow is pretty clear, I appreciate it.
2. The work extends the SGDA with reshuffling into the nonconvex minimax regime.

Weakness:
1. The comparison with exisitng works (to show the outperformance of proposed algorithms) made me a little confusing (details below).
2. Some closely related works are missing in the discussion.

**Summary Of The Paper:**

In this paper, authors studied two-time-scale SGDA (both simultaneous and alternative fashions) with random reshuffling, applied on nonconvex-PL and primal-PL minimax problems. They provided the gradient complexity results of proposed algorithms, which are claimed to outperform over existing literature. Furthermore authors studied the lower bound of SGD with two-time-scale on SC-SC problem, which verifies the tightness of complexity results in the deterministic primal-PL case.

**Summary Of The Review:**

1. I am confused on the comparison in Section 4. Your paper only considered the finite-sum case, while you compare with the SGDA result in Lin et al. (2020), and their paper considers the purely stochastic case $\mathbf{E}_{\xi}[f(x,y;\xi)]$. In my opinion, I may compare Theorem 1 with existing finite-sum algorithm complexity and corresponding lower bound results, e.g., SREDA paper (let's restrict to nonconvex-strongly-concave case only, because Lin et al. (2020) only discussed this case). And with that I may found that the $O(\epsilon^{-3})$ result is not sharp enough compared to SREDA (which is $O(\sqrt{n}\epsilon^{-2})$), is any reason on it, do I miss anything here?

- SREDA: *Luo, Luo, et al. "Stochastic recursive gradient descent ascent for stochastic nonconvex-strongly-concave minimax problems." NeurIPS 2020*.

2. Also at the same time, the comparison with Lin et al. (2020) is not that fair in my opinion. You assumed component smoothness (Assumption 1), while Lin et al. (2020) only requires the smoothness of the whole objective function. Also some lower bound works shows that $O(\epsilon^{-4})$ should be the fundamental limit in the case of Lin et al. (2020). With that, it is less convincing for me that the improvement that authors claimed stem from random reshuffling (or without replacement), I may argue that the improvement (if any) originates from the additional component smoothness assumption.

3. Also when in complexity result summary (e.g., Section 1.1), I found the complexity results always omit the dependence $n$, while Theorem 1 and 2 still have dependence on $n$. But in some other random reshuffling literature (e.g., Mishchenko et al., 2020), their results summary often come with the dependence on $n$, is there any difference between your work and existing literature which comes with the $n$ dependence?

4. I found that there is another work on nonconvex-PL problems, while authors do not discuss it.

- Yang, Junchi, et al. "Faster single-loop algorithms for minimax optimization without strong concavity." AISTATS 2022.

5. A minor question is that, in several existing finite-sum algorithm literature (e.g., SREDA above), they do not need the bounded variance assumption (or its variant), but here you need Assumption 2, should I regard it as a disadvantage of the random reshuffling?

I do not have enough time to check the lengthy proof regarding such short review time frame, but I will try to check it after the review deadline.

All in all, now I am concerned on evaluating the significance of the results compared to literature, I am skeptical on that the improvements on complexities has nothing to do with RR. I hope to have more insights from authors. Please definitely correct me if I misunderstand anything here. Thank you very much for the efforts.

---

> ### Author Response · Authors · 2022-11-15
> **Response to Reviewer 9Nrg**
>
> We are grateful for the reviewer's time and effort invested in reviewing our paper, as well as the insightful comments. Below, we address the comments by their numberings.
>
> 1. **On the comparison in Section 4, considering Lin et al. (2020) and Luo et al. (2020)**.
>
> - First of all, we want to explain why we chose the result from Lin et al. (2020) for the comparison. We consider the finite-sum problem as a subclass of purely-stochastic problems. This is possible because the sampled component index at each iteration can be regarded as a random variable. The analysis by Lin et al. (2020) takes advantage of the assumption of ‘independently sampled and unbiased’ gradient oracles. In the finite-sum setting, the with-replacement sampling of components naturally satisfies this assumption. Therefore, the analysis by Lin et al. (2020) naturally implies the convergence rate of with-replacement (simultaneous) SGDA for the finite-sum minimax problem. In our revision, we clarified this point at the beginning of Section 4.1.
> - On the other hand, we would like to argue that our results are not directly comparable with SREDA (Luo et al., 2020). In the finite-sum version of SREDA, the noiseless full-batch gradient of the objective function is periodically evaluated to apply the variance reduction technique. The algorithm's ability to access full-batch gradients is the key difference between SREDA and SGDA. In light of the fast convergence rates of other variance reduction algorithms, we strongly believe that it is the main reason for $O(\epsilon^{-2})$ complexity of SREDA pointed out by the reviewer.
> - We would like to emphasize the main purpose of our comparison in Section 4.1: we wanted to highlight the impact of the component sampling method (with-replacement v.s. random reshuffling) on the convergence speed of SGDA using ‘small mini-batches,’ i.e., without having access to the full objective gradient. Such stochastic settings with small mini-batches are of great relevance to practice.
> - As a side remark, if we are allowed to use all the components at each iteration, we could also utilize the full-batch gradients without any sampling noise. In that case, the vanilla full-batch GDA can also achieve the aforementioned $O(\epsilon^{-2})$ rate for smooth nonconvex-strongly-concave case, as already studied by Lin et al. (2020) and discussed in Section 4.3 of our paper. (We are not claiming that SREDA is as fast as vanilla GDA: the finite-sum-variant SREDA has better complexity in condition number $\kappa$ and/or the number of components $n$.)
> - Nonetheless, we agree that the variance reduction technique (Johnson and Zhang, 2013; Palaniappan and Bach, 2016; Reddi et al., 2016; Luo et al., 2020; Yang et al., 2020a, 2020b) is an interesting field of study, apart from the component shuffling methods. Notably, there is a paper by Malinovsky et al. (2021) that studies a combination of random reshuffling and variance reduction on (strongly-)convex minimization problem. The presence of such a result highlights that variance reduction and without-replacement sampling can be considered orthogonal.
>
>
> 2. **Lower complexity bound and the smoothness conditions.**
>
> - We should first explain that $\Omega(\epsilon^{-4})$ lower bound does not exactly apply to our analysis on random reshuffling. This is mainly because the lower bound assumes independent and unbiased stochastic gradient oracles above all, whereas this assumption is no longer true for algorithms using without-replacement sampling (Nagaraj et al., 2019).
>
> - To be more specific, Li et al. (2021) provided a $\Omega(\epsilon^{-4})$ lower complexity bound for nonconvex-strongly-concave minimax optimization using stochastic unbiased oracles. Also, Arjevani et al. (2022) proved $\Omega(\epsilon^{-4})$ lower bound for nonconvex stochastic minimization with unbiased oracles and smoothness of the whole objective; Yang et al. (2022) cited the paper to claim that their $O(\epsilon^{-4})$ complexity for (with-replacement, alternating) SGDA is optimal for the nonconvex-PL problem. Again, we note that both prior works on the lower bound assume that the stochastic gradient oracle must be unbiased and independently sampled. On the contrary, using the random reshuffling technique, the component gradients sampled at intermediate iterations of each epoch are ‘biased’ estimators of the full-batch gradient because the iterates in each epoch are not independent of each other. This crucial difference is also mentioned in the introduction of the paper by Nagaraj et al. (2019), and this is also one of the main reasons that make the analysis of without-replacement algorithms (including SGDA-RR) challenging.
>
> (continued to the next reply)

---

> > ### Author Response · Authors · 2022-11-15
> > **Response to Review 9Nrg (2)**
> >
> > (following the previous reply)
> >
> > - For this reason, we would like to emphasize that the faster convergence speed of SGDA-RR bypassing the known lower bounds is not a contradiction. Our key observation that random reshuffling leads to faster convergence also agrees with existing literature on without-replacement SGD for minimization (Haochen and Sra, 2019; Ahn et al., 2020; Mishchenko et al., 2020; Nguyen et al., 2021).
> >
> > - The reviewer points out that the convergence speed gain by SGDA-RR may in fact come from component smoothness. Although we cannot disprove the reviewer's claim, it is unclear to us whether we can immediately obtain faster convergence of ‘with-replacement’ SGDA just by additionally assuming component smoothness. Before and during the rebuttal period, we carefully went through the proof by Lin et al. (2020) and other several papers on with-replacement minimax algorithms to discover room for speed-up when the component smoothness is leveraged. Unfortunately, we could not find any room for improvement, mainly because the existing analyses depend heavily on the unbiasedness of gradient oracles and there is not much benefit that component smoothness can add to them. Further research is required to fully settle down the question of whether we could improve with-replacement SGDA by additionally assuming component smoothness. Besides, it might also be interesting to discuss in depth whether we can prove a faster convergence rate for without-replacement SGD(A) without the assumption.
> >
> > - Lastly, we would like to point out that component smoothness has been widely used in other existing results in the literature, especially on without-replacement algorithms (Haochen and Sra, 2019; Ahn et al., 2020; Mishchenko et al., 2020; Malinovsky et al., 2021; Nguyen et al., 2021) and other variance reduction techniques for finite-sum problems (Johnson and Zhang, 2013; Palaniappan and Bach, 2016; Reddi et al., 2016; Luo et al., 2020; Yang et al., 2020a, 2020b). Hence, our assumption on component smoothness is not a particularly strong one.
> >
> > 3. **Dependence on $n$.**
> >
> > - Thank you for pointing out the dependence on $n$. We agree that clearly stating this dependence is important. In our update, we now changed the expression of our results from epoch complexities to gradient complexities to ease the comparison between with-replacement and without-replacement SGDA(s). The modification is reflected in the following sections: Section 1.1 ("Summary of our contributions"), 3.2 ("Main theorems ..."), and 4.1 ("Comparison with stochastic with-replacement setting").
> >
> > 4. **Another work on nonconvex-PL problems (Yang et al., 2022)**
> >
> > - Thank you for bringing up the closely related paper (Yang et al., 2022). In fact, we became aware of the paper soon after the submission deadline, so we added the comparison to our local version a while ago but had to wait until the rebuttal period to reflect that update. In Section 4.1 of our revision, we added a comparison between our results and theirs. To summarize the added content, Yang et al. (2022) obtain the gradient complexity $O(\kappa_2^4\epsilon^{-4})$ for alternating SGDA to find $\epsilon$-stationary point of primal function ($\mathbb{E}\|\nabla\Phi(x_t)\|^2\le \epsilon^2$). Thus, we claim that our $O(\sqrt{n}\kappa_2^3\epsilon^{-3})$ rate is an improvement.
> >
> > 5. **Bounded variance assumption.**
> >
> > - As far as we know, it seems hard to say that the bounded variance assumption is only needed for the analysis of random reshuffling. This is because bounded gradient variance is a typical assumption in optimization literature even in the with-replacement and purely-stochastic settings (Ghadimi and Lan, 2013; Stitch, 2019; Li et al., 2021; Arjevani et al., 2022). As you correctly point out, several variance reduction methods (e.g., Reddi et al., 2016; Yang et al., 2020a, 2020b) do not require bounded variance assumption, while other stochastic methods do require it. We view it as a strength of the variance reduction methods, not as a weakness of many other stochastic methods including SGDA-RR.
> >
> >
> > Again, we appreciate the reviewer for the detailed comments and questions. We hope this response clarifies the reviewer's concerns. Please let us know if you have any remaining questions/comments.
> >
> > Best regards,
> >
> > Authors

---

> > > ### Author Response · Authors · 2022-11-15
> > > **References for the Author's last comment(s)**
> > >
> > > Reference
> > >
> > > Arjevani, Yossi, et al. "Lower bounds for non-convex stochastic optimization." Mathematical Programming (2022): 1-50.
> > > Ahn, Kwangjun, Chulhee Yun, and Suvrit Sra. "SGD with shuffling: optimal rates without component convexity and large epoch requirements." Advances in Neural Information Processing Systems 33. 2020.
> > > Ghadimi, Saeed, and Guanghui Lan. "Stochastic first-and zeroth-order methods for nonconvex stochastic programming." SIAM Journal on Optimization 23.4 (2013): 2341-2368.
> > > Haochen, Jeff, and Suvrit Sra. "Random shuffling beats sgd after finite epochs." International Conference on Machine Learning. PMLR, 2019.
> > > Johnson, Rie, and Tong Zhang. "Accelerating stochastic gradient descent using predictive variance reduction." Advances in Neural Information Processing Systems 26. 2013.
> > > Li, Haochuan, et al. "Complexity lower bounds for nonconvex-strongly-concave min-max optimization." Advances in Neural Information Processing Systems 34 (2021): 1792-1804.
> > > Lin, Tianyi, Chi Jin, and Michael Jordan. "On gradient descent ascent for nonconvex-concave minimax problems." International Conference on Machine Learning. PMLR, 2020.
> > > Luo, Luo, et al. "Stochastic recursive gradient descent ascent for stochastic nonconvex-strongly-concave minimax problems." Advances in Neural Information Processing Systems 33. 2020.
> > > Mishchenko, Konstantin, Ahmed Khaled, and Peter Richtárik. "Random reshuffling: Simple analysis with vast improvements." Advances in Neural Information Processing Systems 33 (2020): 17309-17320.
> > > Malinovsky, Grigory, Alibek Sailanbayev, and Peter Richtárik. "Random reshuffling with variance reduction: New analysis and better rates." arXiv preprint arXiv:2104.09342 (2021).
> > > Nagaraj, Dheeraj, Prateek Jain, and Praneeth Netrapalli. "Sgd without replacement: Sharper rates for general smooth convex functions." International Conference on Machine Learning. PMLR, 2019.
> > > Nguyen, Lam M., et al. "A unified convergence analysis for shuffling-type gradient methods." The Journal of Machine Learning Research 22.1 (2021): 9397-9440.
> > > Palaniappan, Balamurugan, and Francis Bach. "Stochastic variance reduction methods for saddle-point problems." Advances in Neural Information Processing Systems 29. 2016.
> > > Reddi, Sashank J., et al. "Stochastic variance reduction for nonconvex optimization." International Conference on Machine Learning. PMLR, 2016.
> > > Stich, Sebastian U. "Unified optimal analysis of the (stochastic) gradient method." arXiv preprint arXiv:1907.04232 (2019).
> > > Yang, Junchi, Negar Kiyavash, and Niao He. "Global convergence and variance reduction for a class of nonconvex-nonconcave minimax problems." Advances in Neural Information Processing Systems 33. 2020a.
> > > Yang, Junchi, et al. "A catalyst framework for minimax optimization." Advances in Neural Information Processing Systems 33. 2020b.
> > > Yang, Junchi, et al. "Faster single-loop algorithms for minimax optimization without strong concavity." AISTATS 2022.

---

> ### Comment · Reviewer_9Nrg · 2022-12-08
> **Thank you and response**
>
> I appreciate authors' efforts to clarify my confusions. First I want to list here the main arguments by authors to make sure that I understand it correctly:
> 1. Why compare to Lin et al. (2020) instead of SREDA: Here SGDA have no access to the full objective gradient, while VR type algorithms have these access, and authors believe this is the reason of the fundamental difference between $\mathcal{O}(\epsilon^{-2})$ and $\mathcal{O}(\epsilon^{-3})$.
> 2. The improvement comes from RR+component smoothness: Now it is still unclear which is the key part to improve the complexity. The lower bound for $\mathcal{O}(\epsilon^{-4})$ does not apply here due to the independence assumption of oracles.
> 3. Missing dependence on $n$: solved
> 4. Missing related works: solved
> 5. Bounded variance assumption: Authors suggest to regard it as an advantage of VR-type method, from a optimistic perspective.
>
> ---
>
> To be honest, it makes me still a bit hesitated on increasing the evaluation.
> 1. Although authors tried and claimed that the generic SGDA possibly cannot be improved with component smoothness (CS) alone, and we both agree CS is common in literature, but we both should agree that it is still requiring a formal proof. The improvement needs both RR and CS to take effect, which in my honest opinion will weaken the significance of the algorithm. Of course I admit I can take a more optimistic viewpoint to see that there is a open problem to further check whether CS is fundamental.
> 2. To some extent, I may argue that, now RR only works on finite-sum problems, which should correspond to offline problems, all the data are stored there, and it seems to be unrealistic/unnecessary to restrict ourselves to have access to only one sample in each iteration, at least we should be able to work with some small batches (as authors mentioned in Appendix A).
> 3. And with the above discussion, it seems to reveal that, besides the full gradient access, RR methods are systematically worse than VR methods (is that correct?). I want to make it clear that the above two points are my questions for the RR strategy itself, rather than the paper which reflects authors' great efforts.
> 4. A maybe minor point reflecting P1, is there any analysis for algorithms in finite-sum nonconvex-PL (or strongly-concave) problems, while does not require full gradient (e.g. single-call SGDA/SEG specifically for finite-sum problems)? I expect if exists, these should be more ideal candidates to be compared with your RR algorithm (with respect to Lin's work)?
>
> I acknowledge and appreciate authors' efforts in the paper and discussion here, while the issues remaining here make me lingering on increasing the evaluation. Now I will keep my score here, and I will raise my issues and the hesitation to AC and other reviewers. Thank you again for the helpful reply.

---

> > ### Author Response · Authors · 2022-12-09
> > **Additional response to Reviewer 9Nrg (1)**
> >
> > We truly thank the reviewer for their attention to our work and for continuing the discussion of quality. We would like to highlight that **we have a mathematical proof for our claim in the response to the first question**. We plan to add the proof to the camera-ready version if our paper gets accepted.
> >
> > **1. About component smoothness assumption**
> >
> > To answer the reviewer's doubt about whether or not the component smoothness is crucial for faster convergence, we provide some formal results on lower bounds of with-replacement SGD(A) (for simplicity, we use mini-batches of size 1). We defer the proof to a separate reply. The first theorem below provides a lower bound on with-replacement SGD for minimization problems.
> >
> > **Theorem A.** *For any step size $\eta>0$, there exists a function $f(x):= \frac{1}{n}\sum_{i=1}^n f_i (x)$ consisting of $n>1$ component functions $f_i$ satisfying:*
> >
> > 1. *Each component $f_i$ is smooth and strongly-convex;*
> > 2. *After running $T>1$ iterations of with-replacement SGD (with the mini-batch size 1) starting from $x_0$, the last iterate $x_T$ satisfies $\mathbb{E}[f(x_T)-f^\ast] \ge \Omega(1/T)$, where $f$ has a minimum $f^\ast$ and the expectation is taken with respect to the randomness of i.i.d. index choice at each iteration.*
> >
> > Next, we show this theorem naturally induces a convergence lower bound for the minimax counterpart: with-replacement SGDA.
> > Consider a (finite-sum) minimax problem $\min_x \max_y g(x,y) := f(x)-f(y)$, where $f=\frac{1}{n}\sum_{i=1}^n f_i$ is a worst-case function in the proof of Theorem A. Here, the minimax problem on $g$ can be solved by minimizing $f$. Moreover, since the primal function $\Phi(x):= \max_y g(x,y)$ is in fact the same as $f(x)-f^\ast$, the potential function $V_\lambda (x,y) := \lambda[\Phi(x)-(\min_x \Phi(x))] + [\Phi(x)-g(x,y)]$ becomes the same as $\lambda(f(x)-f^\ast) + (f(y)-f^\ast)$ for some constant $\lambda>0$. Combining these facts, we can easily obtain the following lower convergence bound of *with-replacement SGDA*.
> >
> > **Corollary B.** There exists a function $g(x,y):=\frac{1}{n}\sum_{i=1}^n g_i(x,y)$ consisting of $n$ component functions $g_i$ which are all smooth and strongly-convex-strongly-concave, for which the last iterate $(x_T, y_T)$ of with-replacement SGDA after $T>1$ iterations satisfies $\mathbb{E}[V_\lambda (x_T, y_T)]\ge \Omega(1/T)$.
> >
> > Corollary B formally proves that with-replacement SGDA on component-wise smooth and strongly-convex-strongly-concave minimax problems has a worst-case convergence rate $\Omega(1/T)$. This in fact matches the $\mathcal{O}(1/T)$ upper bound obtained for primal-PL-PL problems by Yang et al. (2020). Considering that component-wise smooth strongly-convex-strongly-concave functions form a strict subset of primal-PL-PL functions, Corollary B establishes that adding component smoothness assumption does not provide further speed up for with-replacement SGDA. In other words, component smoothness alone is not sufficient for improving the existing upper bounds for with-replacement SGDA.
> >
> > In contrast, our theoretical result (especially Theorem 2 of our paper) shows that SGDA with random reshuffling (SGDA-RR) has a much faster convergence rate $\mathbb{E} [V_\lambda] \le \tilde{\mathcal{O}}(\frac{1}{nK^2})$ for primal-PL-PL minimax problems, where $K$ is the number of epochs. One can check that our $\tilde{\mathcal{O}}(\frac{1}{nK^2})$ bound is faster than the tight convergence rate $\Theta(\frac{1}{T})$ of with-replacement SGDA by simply plugging in $T = nK$. In light of Corollary B we proved, we can now claim that the improvement can be solely attributed to RR.
> >
> > Although we do not provide a lower bound for more general nonconvex-PL problems here, we believe the more challenging case of nonconvex-PL lower bound is a topic for another separate paper. Nonetheless, we conjecture that the speed up by SGDA-RR in nonconvex-PL settings is also due to the effect of RR, not component smoothness. We hope that this lower bound does a better job of convincing the reviewer.
> >
> >
> > **2. “RR only works on finite-sum problems... unrealistic/unnecessary to restrict ourselves..."**
> >
> > We agree with the reviewer that in offline settings where the full dataset is available, it is much more common to use *mini-batches* to run stochastic optimization methods. We would like to remind the reviewer that our paper does handle mini-batch versions of SGDA-RR in Appendix A; in Theorems 4 and 5, we prove the convergence rate of the algorithms for the entire spectrum of mini-batch sizes $b$, ranging from 1 to $n$. Indeed, our main text only presents the results for the case $b = 1$, but this is just to simplify the exposition.
> >
> > (continued to the next reply)

---

> > > ### Author Response · Authors · 2022-12-09
> > > **Additional response to Reviewer 9Nrg (2)**
> > >
> > > (following the previous reply)
> > >
> > > **3. "RR methods are systematically worse than VR methods? ..."**
> > >
> > > It is true that the variance reduction (VR) methods achieve better convergence rates in several settings compared to stochastic methods. To the best of our knowledge, however, VR methods are not very popular in modern deep learning applications and it is more common to apply SGD(A) and their variants. We aim to explore what happens when we combine a sampling scheme, which is predominant in practice but theoretically less understood, with an algorithm that is widely used by practitioners; thus, our work is highly relevant to practice. Lastly, we would like to reiterate that RR and VR can be considered separate techniques and do not always conflict.
> > >
> > > **4. "Is there any finite-sum analysis that does not require full gradient?**
> > >
> > > So far as we are aware, for nonconvex-PL or nonconvex-strongly-concave problems and the algorithms not using a full-batch gradient, we could not find any other analysis *solely dedicated to finite-sum* problems. In our opinion, this is because such an analysis can be easily deduced from "general stochastic" problems with unbiased and independent oracles, as we mentioned before. Notably, we now believe that the comparison of with- vs without-replacement rates is now fair because we have elaborated a tight convergence rate of with-replacement SGDA even under component smoothness in our Corollary B above.
> > >
> > > ***
> > >
> > > Again, we appreciate the reviewer's additional comments and questions. We hope that our response will address your remaining concerns.
> > >
> > > Best regards,
> > > Authors
> > >
> > > ***
> > >
> > > References
> > >
> > > - Junchi Yang, Negar Kiyavash, and Niao He. Global convergence and variance reduction for a class of nonconvex-nonconcave minimax problems. Advances in Neural Information Processing Systems, 2020.

---

> > > > ### Author Response · Authors · 2022-12-09
> > > > **Additional response to Reviewer 9Nrg (3): Detailed proof for lower bound**
> > > >
> > > > **Theorem A (re-statement).** *For any step size $\eta>0$, there exists a function $f(x):= \frac{1}{n}\sum_{i=1}^n f_i (x)$ consisting of $n>1$ component functions $f_i$ satisfying:*
> > > >
> > > > 1. *Each component $f_i$ is smooth and strongly-convex;*
> > > > 2. *After running $T>1$ iterations of with-replacement SGD (with the mini-batch size 1) starting from $x_0$, the last iterate $x_T$ satisfies $\mathbb{E}[f(x_T)-f^\ast] \ge \Omega(1/T)$, where $f$ has a minimum $f^\ast$ and the expectation is taken with respect to the randomness of i.i.d. index choice at each iteration.*
> > > >
> > > > **Proof of Theorem A)** We construct worst-case functions with quadratic functions on $\mathbb{R}$, which are clearly $\mu$-smooth and $\mu$-strongly convex for a constant $\mu$. Fix $\mu>0$ and Let $x_0\in \mathbb{R}$ be the initial iterate.
> > > >
> > > > - **Case 1** ($0<\eta<\frac{1}{\mu T}$). Consider $f_i(x) = \frac{\mu}{2} x^2$ for every $i\in [n]$: every component is the same. At each iteration $t\ge 1$, we obtain $x_t = (1-\eta\mu)x_{t-1}$ by a step of SGD. Then, since $\eta\mu < 1/T\le 1/2$, $$x_T^2 =
> > > > \left(1-\eta\mu\right)^{2T} x_0^2 > \left( 1-\frac{1}{T} \right)^{2T} x_0^2 \ge \left( 1-\frac{1}{2} \right)^4 x_0^2 = \frac{x_0^2}{16}.$$
> > > > Since $f(x)=\frac{1}{n}\sum_{i=1}^n f_i(x)=\frac{\mu}{2} x^2$ has a minimum $f^\ast=0$ at $x=0$, we have $f(x_T)-f^\ast>\frac{\mu}{32}x_0^2 = \Omega(\mu)\cdot x_0^2$.
> > > >
> > > > - **Case 2-a)** ($\frac{1}{\mu T}\le \eta\le\left( \frac{2}{\mu}-\frac{1}{\mu T} \right)$, $n$ even). Consider $f(x)=\frac{\mu}{2} x^2$ consisting of even number of components $f_i$'s defined by $$f_i(x) =
> > > > \begin{cases}
> > > >     \frac{\mu}{2} x^2 + \nu x, & (i\le \frac{n}{2}),\\\\
> > > >     \frac{\mu}{2} x^2 - \nu x, & (i\ge \frac{n}{2}+1),
> > > > \end{cases}$$
> > > > for some number $\nu\in\mathbb{E}$.
> > > > At each iteration $t\ge 1$, we choose a component index $i(t)\stackrel{\mathrm{i.i.d.}}{\sim} \mathrm{Unif}([n])$ (with-replacement sampling). Then we can write the chosen component function at iteration $t$ as $f_i(t) = \frac{\mu}{2}x^2 - s_t \nu x$ for some i.i.d. random variable $s_t \sim \mathrm{Unif}({\pm 1})$. Accordingly, an SGD step can be written as $$
> > > > x_t = x_{t-1} - \eta \nabla f_{i(t)} (x_{t-1}) = (1-\eta \mu) x_{t-1} + \eta s_t \nu.$$
> > > > By applying telescopic sum, we have$$
> > > > x_T = (1-\eta \mu)^T x_0 + \eta \nu \sum_{t=1}^T (1-\eta\mu)^{(T-t)} \cdot s_{t}.$$
> > > > Taking squares and expectations (with respect to the random variables $s_1, \ldots, s_T$) to both sides, we have $$\mathbb{E}[x_T^2] = (1-\eta \mu)^{2T}x_0^2 + \eta^2\nu^2 \sum_{t=1}^T (1-\eta\mu)^{2(T-t)}.$$
> > > > We applied here the fact that $s_t$'s are zero-mean independent random variables with absolute values 1: $$\mathbb{E}[s_t\cdot s_{t'}] =
> > > > \begin{cases}
> > > >     0, & t\ne t'\quad (\because \text{independent}),\\\\
> > > >     1, & t=t' \quad (\because s_t^2=1).
> > > > \end{cases}$$
> > > > We calculate the sum above as follows: since $(1-\eta\mu)^2 \le (1-1/T)^2$ and $(1-1/T)^T\le e^{-1}$, $$
> > > > \sum_{t=1}^T (1-\eta\mu)^{2(T-t)} = \frac{1-(1-\eta\mu)^{2T}}{1-(1-\eta\mu)^2}\ge \frac{1-(1-\frac{1}{T})^{2T}}{2\eta\mu(1-\frac{\eta\mu}{2})}\ge \frac{1-e^{-2}}{2\eta\mu}.$$
> > > > With this inequality, and since $(1-\eta \mu)^{2T}x_0^2\ge 0$,  we can lower-bound the expectation $\mathbb{E}[x_T^2]$: $$
> > > > \mathbb{E}[x_T^2] \ge \eta^2 \nu^2 \cdot \frac{1-e^{-2}}{2\eta\mu} = \frac{(1-e^{-2})\nu^2}{2\mu}\eta\ge \frac{(1-e^{-2})\nu^2}{2\mu^2 T}.$$
> > > > Since $f$ has a minimum $f^\ast=0$ at $x=0$, we eventually have $$
> > > > \mathbb{E}[f(x_T)-f^\ast]=\frac{\mu}{2}\mathbb{E}[x_T^2]\ge \frac{(1-e^{-2})\nu^2}{4\mu T}=\Omega \left( \frac{\nu^2}{\mu T} \right).$$
> > > >
> > > > - **Case 2-b)** ($\frac{1}{\mu T}\le \eta\le\left( \frac{2}{\mu}-\frac{1}{\mu T} \right)$, $n>1~\text{odd}$). Consider $f_n(x)=0$ and let the remaining $n-1$ components be the same as the previous case (with an even number of components). Thereby, $f(x) = \left(\frac{n-1}{n}\right)\frac{\mu}{2}x^2$. The zero-component $f_n=0$ does not affect both the trajectory of SGD (*i.e.*, the points visited by SGD) and the optimality of $f$ ($f^\ast=0$ at $x=0$). Thus, the $\Omega \left( \frac{\nu^2}{\mu T} \right)$ bound also holds.
> > > >
> > > > - **Case 3** ($\eta>\left(\frac{2}{\mu}-\frac{1}{\mu T} \right)$). Consider the same functions in **Case 1**: $f_i(x) = \frac{\mu}{2} x^2$ for every $i\in [n]$. By running SGD for $T$ iterations, we have $x_T^2 = \left(1-\eta\mu\right)^{2T} x_0^2$. However, in this case, $1-\eta\mu$ is a negative number because$$
> > > > 1-\eta\mu < 1-\left( 2-\frac{1}{T} \right) = -1 + \frac{1}{T}<0.$$
> > > > Thus, applying $T\ge 2$, we have the following inequality.$$
> > > > x_T^2 = \left(1-\eta\mu\right)^{2T} x_0^2 > \left( -1+\frac{1}{T} \right)^{2T} x_0^2=\left( 1-\frac{1}{T} \right)^{2T} x_0^2\ge \left( 1-\frac{1}{2} \right)^4 x_0^2 =\frac{x_0^2}{16}$$
> > > > Hence, we have the same result as **Case 1**. $\square$

---

> > > > > ### Comment · Reviewer_9Nrg · 2022-12-12
> > > > > **Thank you**
> > > > >
> > > > > I appreciate authors' efforts for improving the paper during this period, I am satisfied with the lower bound result for with-replacement SGDA on SC-SC case.
> > > > >
> > > > > Lastly I want to reiterate that, I totally agree with authors that "...this is because such an analysis can be easily deduced from "general stochastic" problems with unbiased and independent oracles, as we mentioned before...", while my point here is that such deduction may be vacuous (not tight in the dependence of $\epsilon$), because I often find complexities in finite-sum works come with form $O(\text{poly}(n)\text{poly}(\epsilon))$, rather than $O(\text{poly}(\epsilon))$ in general stochastic problems, that's the reason I keep asking this and expect a comparison between your work and results "purely" in finite-sum literature.
> > > > >
> > > > > All in all, I am satisfied with the additional result, please definitely add this one to the final version (also the conjecture/open problem in nonconvex-PL case), I am happy to turn my position to 6. Thank you for the efforts!

---

### Official Review · Reviewer_8wv3 · 2022-10-27

**Confidence:** 3
**Correctness:** 4
**Technical Novelty And Significance:** 3
**Empirical Novelty And Significance:** Not applicable
**Recommendation:** 8

**Clarity, Quality, Novelty And Reproducibility:**

The clarity is good and the paper is easy for readers to understand.


**Strength And Weaknesses:**

This paper is clearly written and well organized. The theoretical analysis is solid and the results are strong enough to show the advantage of the studied method over its competitors. A minor weakness is a comparison with recent work on similar topics.


**Summary Of The Paper:**

This paper studies the convergence of stochastic gradient descent ascend (SGDA) for nonconvex-PL minimax problems, where the data points are randomly shuffled and sampled without replacement during the training. The authors provided strong theoretical convergence of SGDA and confirm the empirical observation that random shuffling converges faster than sampling with replacement. They also provided a lower bound for strongly-convex-strongly-convex mini-max optimization problems.


**Summary Of The Review:**

In general, this paper is a strong work with a good presentation and solid theoretical results. I only have the following minor comments that could potentially improve the current manuscript.

Is $\nabla f_i(z)$ in Assumption 1 the concatenation of the gradients $\nabla_1$ and $\nabla_2$? I did not find a definition for it.

I am not sure why it is called a two-time-scale algorithm since the order of learning rates $\alpha$ and $\beta$ are both in the same order with respect to the time parameter $T$. It is just two different updating frequencies that differ in a constant order.

In Section 4, you mentioned that the results can be easily extended to the mini-batch setting. Can you elaborate more on this? Since the data points in the batch are not i.i.d. (if you first shuffle the data and then read a batch in a sequential way), does this bring benefit in the convergence rat?

It would be nice to have some empirical comparison between the wise-alternating SGDA-RR in this paper and the AGDA-RR in Das et al. (2022) which is proved to be optimal in the two-sided PL setting. This is also to verify the practical advantage of SGDA-RR claimed in this paper for different epoch sizes $n$.

---

> ### Author Response · Authors · 2022-11-15
> **Response to Reviewer 8wv3**
>
> We sincerely appreciate the reviewer for their effort in reviewing our paper. Thank you also for the positive evaluation. Below, we reply to the questions raised by the reviewer.
>
>
> 1. "Is $\nabla f_i(z)$ in Assumption 1 the concatenation of the gradients $\nabla_1$ and $\nabla_2$?"
>
> - Yes, exactly. The full gradient (for both arguments $x$ and $y$) is a concatenation of the argument-wise (partial) gradients $\nabla_1$ and $\nabla_2$. We clarified this point in Section 2.1 of our revision. We apologize for the confusion.
>
>
> 2. "I am not sure why it is called a two-time-scale algorithm since the order of learning rates $\alpha$ and $\beta$ are both in the same order with respect to the time parameter $T$."
>
> - We agree that the term "two-time-scale" is commonly used when one step size is much smaller than the other step size(s) in terms of time parameter, i.e., $\alpha(T) = o(\beta(T))$ (Borkar, 1997; Heusel et al., 2017). On the other hand, some recent results use the term "two-time-scale" in a broader sense, highlighting that the step sizes $\alpha$ and $\beta$ are possibly not the same constants (e.g., Lin et al., 2020; Yang et al., 2020, 2022).
>
> - In some easy-to-analyze settings such as smooth strongly-monotone problems, the existing analyses of GDA and extragradient (EG) commonly choose $\alpha=\beta$ (i.e., one-time-scale), which allows them to exploit strong monotonicity in a natural way. For the wider classes of problems (nonconvex-PL and primal-PL-PL) that we consider in this paper, however, it is necessary that we deviate from the one-time-scale approach, since the choice of $\alpha = \beta$ leads to divergence in some instances of primal-PL-PL functions (e.g., Theorem 4.1 of Li et al. 2022). Our Theorems 1 and 2 hence use step sizes of ratio $\beta/\alpha=r\gtrsim \kappa_2^2=(L/\mu_2)^2$, and we adopt the term two-time-scale to emphasize this choice. Also, please notice that when our minimax problem is ill-conditioned in terms of $y$ (i.e., $\kappa_2$ is huge), then $\alpha$ we chose becomes very small compared to $\beta$, which further motivates our choice of the term "two-time-scale."
>
>
> 3. On benefits of mini-batch extension for without-replacement sampling.
>
> - As we mentioned in our main results (Section 3.2), there is a full description of 'Mini-batch SGDA-RR' (Algorithm 2) in Appendix A, a natural extension of SGDA-RR (Algorithm 1). The algorithm indeed shuffles the component functions and accesses them in a minibatch fashion according to the shuffled order, as the reviewer correctly guessed.
>
> - If we choose a subset of $b$ vectors from the given $n$ vectors without replacement, the variance due to the sampling must be $\frac{n-b}{b(n-1)}$ multiplied by the total sample variance $\tau^2$: as $b$ gets larger, the variance gets lower. This property of without-replacement sampling can be deduced from probability theory, and it is formally stated in Proposition 14 in Appendix B.2. Proposition 14 is in fact a simple extension of Lemma 1 by Mishchenko et al. (2020) to general mini-batch sizes. By applying Proposition 14, our convergence results on the fully-stochastic case (with mini-batches of size $b=1$) can be readily extended to the case of general mini-batch size, all the way up to the full-batch case $b=n$.
>
> - In Appendix A, we presented the convergence rates of Mini-batch SGDA-RR in Theorem 4 (nonconvex-PL) and Theorem 5 (primal-PL-PL). There are three noteworthy features you could find from these theorems.
>   - We use $b \in \{1,...,n\}$ to denote the size of a mini-batch. If we put $b=1$, Theorems 4 and 5 reduce to Theorems 1 and 2, respectively; if $b=n$, Theorems 4 and 5 reduce to (full-batch) GDA.
>   - Here we use a step size $\beta=O(b/nL)$ instead of $O(1/nL)$ used in Theorem 1 and 2. Loosely speaking, we scale up the step sizes by the mini-batch size $b$.
>   - For each upper bound of convergence rate, there is a term proportional to $B\cdot \left(\frac{n-b}{n-1}\right)$. Here, $B$ is a constant for bounding the variance of component gradients (please see Assumption 2). Therefore, the term decreases as $b$ increases, thereby we obtain faster convergence upper bound as $b$ increases.
>
>
> 4. Experimental comparison between SGDA-RR and AGDA-RR (Das et al. 2022).
>
> - Thank you for your suggestion about experiments. In our updated manuscript, we replaced the current section on the proof sketch (Section 6) with a section on empirical comparison between our SGDA-RR, AGDA-RR by Das et al. (2022), and vanilla (with-replacement) SGDA.
>
> We again thank the reviewer again for the comments and feedback. Please let us know if there are any other questions.
>
> Best regards,
>
> Authors

---

> > ### Author Response · Authors · 2022-11-15
> > **References for the Author's last comment**
> >
> > References
> >
> > Borkar, Vivek S. "Stochastic approximation with two time scales." Systems & Control Letters, Volume 29, Issue 5, 1997.
> > Heusel, Martin, et al. "Gans trained by a two time-scale update rule converge to a local nash equilibrium." Advances in neural information processing systems 30. 2017.
> > Lin, Tianyi, Chi Jin, and Michael Jordan. "On gradient descent ascent for nonconvex-concave minimax problems." International Conference on Machine Learning. PMLR, 2020.
> > Mishchenko, Konstantin, Ahmed Khaled, and Peter Richtárik. "Random reshuffling: Simple analysis with vast improvements." Advances in Neural Information Processing Systems 33 (2020): 17309-17320.
> > Yang, Junchi, Negar Kiyavash, and Niao He. "Global convergence and variance reduction for a class of nonconvex-nonconcave minimax problems." Advances in Neural Information Processing Systems 33. 2020.
> > Yang, Junchi, et al. "Faster single-loop algorithms for minimax optimization without strong concavity." AISTATS 2022.

---

### Official Review · Reviewer_s6TB · 2022-10-28

**Confidence:** 4
**Clarity, Quality, Novelty And Reproducibility:** Please see the above review for furth…
**Correctness:** 4
**Technical Novelty And Significance:** 4
**Empirical Novelty And Significance:** Not applicable
**Recommendation:** 8

**Strength And Weaknesses:**

The paper is well-written and the main contributions are clear. To the best of my knowledge, this is one of the first papers that provides an analysis of random reshuffling methods for solving min-max problems. Das et. al 2022 as correctly pointed out by the authors is the most closely related work to this paper but it focuses on either different classes of problems or slightly different algorithms.

I enjoy reading this paper. I went through the proofs and the results seem correct. The related work from optimization literature is also very well presented.

I gave a score of 6 rather than 8 for one main reason:
I believe that the paper needs experimental evaluation. That is provide plots where the theoretical results are verified. The comparison of the proposed SGDA-RR compared to the classical SGDA is heavily needed to convince the reader that this is a valuable algorithm that works also in practice.

Even if the experiments do not match exactly the theoretical results, the reader needs to know what to expect. The existing theorems suggest very small step sizes which means that in practice the proposed RR might not work as expected compared to the vanilla uniform sampling SGDA.

In terms of space, the current section "6. Proof Sketch" could easily move to the Appendix and get replaced by Numerical experiments section.

**Summary Of The Paper:**

The paper focuses on SGDA with random reshuffling (SGDA-RR) for solving finite-sum min-max optimization problems. In particular, it studies simultaneous and alternative SGDA-RR for two different classes of problems nonconvex-PL and primal-PL-PL. The proposed analysis extends to the mini-batch regime and as a result, main theorems can capture the deterministic GDA as a special case. A lower bound of the two time-scale GDA is also presented.


**Summary Of The Review:**

The paper has a solid theoretical contribution and it is very well-written.

I gave a score of 6 rather than 8 for one main reason:
I believe that the paper needs experimental evaluation. That is provide plots where the theoretical results are verified. The comparison of the proposed SGDA-RR compared to the classical SGDA is heavily needed to convince the reader that this is a valuable algorithm that works also in practice.

--------Update------------

Thank you to the authors for providing further clarification on the raised points. I have read the other reviews and the rebuttal and browsed through the paper again.

I increase my score from 6 to 8.

---

> ### Author Response · Authors · 2022-11-15
> **Response to Reviewer s6TB**
>
> Thank you very much for your kind review and feedback. We are glad that you enjoyed reading our paper and that you went the extra mile to read through our proofs.
>
> As per the reviewer's suggestion, we performed numerical experiments on a finite-sum minimax problem to compare SGDA-RR with other algorithms (e.g., vanilla with-replacement SGDA) and to corroborate our theoretical results. Moreover, we replaced the current section "6. Proof Sketch" with "6. Experiments" in our revision. Please see Appendix G of the revised manuscript for a more detailed description of the experiments.
>
> Again, we thank the reviewer for the valuable feedback and the positive evaluation. Please let us know if you have any other comments/questions. Also, we would appreciate it if you could consider reassessing the paper based on our updates.
>
> Best regards,
>
> Authors

---

### Author Response · Authors · 2022-11-15
**Revision of Paper**

Dear reviewers and AC(s),

We express our deepest gratitude for your valuable comments and constructive feedback.

We would like to announce that we made a revision to our submission. For your convenience, we marked newly added/modified sentences, paragraphs, and sections in *green*.

Some noteworthy changes in this revision include:
- As per most reviewers' suggestions, we performed a numerical evaluation on finite-sum quadratic games. We compared six algorithms in total: simSGDA-RR, altSGDA-RR, AGDA-RR (Das et al., 2022), and the with-replacement counterparts of these three algorithms. Moreover, we replaced the content of Section 6 ("Proof Sketch", previously) with experimental results (now, "6. Experiments"). Instead, we moved the proof sketch section to Appendix C.1.
- To ease the comparison between SGDA-RR and vanilla SGDA (with-replacement), we changed the expression of our results from "epoch" complexities to gradient complexities. The modification is reflected in the following sections: Section 1.1 ("Summary of our contributions"), 3.2 ("Main theorems ..."), and 4.1 ("Comparison with stochastic with-replacement setting").
- At the beginning of Section 4.1, we added an explanation that justifies our comparison with prior works about purely stochastic setting. The prior works utilize independently sampled & unbiased gradient oracles for their analysis and this reduces to with-replacement sampling in finite-sum case.
- In Section 4.1, we also added a comparison between our findings (simSGDA-/altSGDA-RR) and the analysis of Yang et al. (2022, altSGDA) on the nonconvex-PL problem,
- We clarified in Section 2.1 (Notation) that the full gradient (for both arguments $x$ and $y$) is a concatenation of the argument-wise (partial) gradients $\nabla_1$ and $\nabla_2$.

While our experimental setup may look somewhat limited as of now, please take into account the fact that the two-week discussion period is too short to make for extensive empirical evaluations. If the paper gets accepted, we plan to add some additional experimental results to the camera-ready version.

We would greatly appreciate it if you could take another look at our revised manuscript and consider updating your scores if needed.

Sincerely,

Authors

---

### Decision · Program_Chairs · 2023-01-20

**Decision:**

Accept: poster

**Justification For Why Not Higher Score:**

I would be ok with having a higher score; it is a good paper. The reason I am not recommending a higher score is that I reserve spotlight/oral recommendations for more creative/original papers. This feels more like technical progress on a rather obvious question (given the existing literature). Which is good, but not in the category of being overly impressive. It did not feel like the paper had to derive fundamentally new techniques to obtain these results, though it is possible I am mistaken.

**Justification For Why Not Lower Score:**

There is a solid contribution so I think the paper should definitely be accepted.

**Metareview: Summary, Strengths And Weaknesses:**

The paper studies the convergence of stochastic gradient descent ascent with random reshuffling (SGDA-RR) in two main finite-sum min-max optimization settings: where only the 'max' part satisfies the Polyak-Lojasiewicz (PL) condition and where both 'min' and 'max' satisfies the PL condition. The component functions in the min-max objective are all assumed to be smooth. Random reshuffling is a commonly used sampling technique where the data points/component functions are selected in a cyclic manner according to a randomly selected permutation, or, equivalently, where the component functions are sampled without replacement. This stands in contrast with easier to analyze sampling with replacement, more commonly examined from a theoretical perspective. Sampling *without* replacement is common in practice and often outperforms sampling with replacement, however, it is much less understood from a theoretical perspective.

The paper proves complexity bounds for SGDA-RR in the two studied settings and provides theoretical justification for the effectiveness of random reshuffling. It further advances the theory of methods with random reshuffling, which is important in its own right. Interestingly, the results crucially rely on taking asymmetrical steps in SGDA updates (i.e., the steps for the min and max part have different lengths). The paper further provides a lower bound on the convergence of SGDA in strongly convex-strongly concave settings which matches the proved convergence bound of SGDA-RR under sufficiently asymmetrical steps.

The initial version of the paper did not provide experimental results, which was a cause of reservation on behalf of some of the reviews. The authors added experiments in the rebuttal period, which led to the more positive view of the paper. The experiments further validate the theoretical results from the paper. From my point of view, the paper should have been accepted with or without the experiments.



**Note From Pc:**

if the above contains the word "oral" or "spotlight" please see: "oral" presentation means -> notable-top-5% and "spotlight" means -> notable-top-25%. As stated in our emails, we are disassociating presentation type from AC recommendations

**Summary Of Ac-Reviewer Meeting:**

N/A